# Adam Optimization with Adaptive Batch Selection

**Gyu Yeol Kim**
Seoul National University
Seoul, South Korea
gyuyeolkim@snu.ac.kr

**Min-hwan Oh**
Seoul National University
Seoul, South Korea
minoh@snu.ac.kr

## Abstract

Adam is a widely used optimizer in neural network training due to its adaptive learning rate. However, because different data samples influence model updates to varying degrees, treating them equally can lead to inefficient convergence. To address this, a prior work proposed adapting the sampling distribution using a bandit framework to select samples adaptively. While promising, the bandit-based variant of Adam suffers from limited theoretical guarantees. In this paper, we introduce *Adam with Combinatorial Bandit Sampling* (AdamCB), which integrates combinatorial bandit techniques into Adam to resolve these issues. AdamCB is able to fully utilize feedback from multiple samples at once, enhancing both theoretical guarantees and practical performance. Our regret analysis shows that AdamCB achieves faster convergence than Adam-based methods including the previous bandit-based variant. Numerical experiments demonstrate that AdamCB consistently outperforms existing methods.

## 1 Introduction

Adam (Kingma & Ba, 2015) is one of the most widely used optimizers for training neural networks, primarily due to its ability to adapt learning rates. The standard version of Adam and its numerous variants treat each training sample equally by employing uniform sampling over the dataset. In practice, however, different data samples can influence model updates to varying degrees. As a result, simply performing full dataset sweeps or sampling data with equal weighting may lead to inefficient convergence and, consequently, unnecessary computational overhead when aiming to satisfy a given convergence criterion.

To address these challenges, Liu et al. (2020) introduced a dynamic approach called AdamBS, which adapts the sampling distribution during training using a multi-armed bandit (MAB) framework. In this method, each training sample is treated as an arm in the MAB, allowing more important samples to be selected with higher probability and having a greater influence on model updates. This approach was intended to improve both the adaptability and efficiency of the optimization process, presenting a promising direction for further advancements.

However, despite its potential benefits, critical issues remain: Specifically, the convergence issue previously identified in the original analysis of Adam (initially reported by Reddi et al. 2018 and later resolved by Zhang et al. 2022) also affects its bandit-based variant, AdamBS, as newly discovered in our work. Consequently, the existing theoretical guarantees regarding the efficiency and effectiveness of AdamBS are invalid (see Section 2.5.3). As a result, to the best of our knowledge, there is no existing Adam-based method that can adaptively sample while providing rigorous performance guarantees. This raises a critical question: *is it possible to design an algorithm that adaptively adjusts the sampling distribution while ensuring both provable guarantees and practical performance improvements?*

In this paper, we propose a new optimization method, *Adam with Combinatorial Bandit Sampling* (AdamCB), which addresses the limitation in both the analysis and implementation of AdamBS by incorporating a combinatorial bandit approach into the sample selection process. In this approach, batch selection is formulated as a combinatorial action, where multiple arms (samples) are selected simultaneously. This combinatorial bandit framework can take advantage of feedback from multiple samples at once, significantly enhancing the adaptivity of the optimizer. For the first time, we provide provable performance guarantees for adaptive batch selection in Adam-based methods, leading to

faster convergence and demonstrating both theoretical and practical improvements over existing approaches. Our main contributions are summarized as follows:

- We propose *Adam with Combinatorial Bandit Sampling* (`AdamCB`), a novel optimization algorithm that integrates the `Adam` method with a combinatorial bandit approach for sample selection. To the best of our knowledge, `AdamCB` is not only the first algorithm to successfully combine *combinatorial* bandit techniques with the `Adam` framework, but also the first to correctly adapt any bandit techniques to `Adam`, significantly enhancing its adaptability.

- We provide a rigorous regret analysis of the proposed `AdamCB` algorithm, demonstrating that it achieves a sharper regret bound compared to both the original `Adam` (which uses uniform sampling) and its bandit-based variant, `AdamBS` (Liu et al., 2020). Additionally, we correct the theoretical errors in the analysis of `AdamBS` and present a revised regret bound (see Table 1 for comparisons).

- We perform empirical evaluations across multiple datasets and models, showing that `AdamCB` consistently outperforms existing `Adam`-based optimization methods in terms of both convergence rate and practical performance. Our results establish `AdamCB` as the first `Adam`-based algorithm to offer both provable convergence guarantees and practical efficiency for bandit-based `Adam` optimization methods.

## 2 Preliminaries

### 2.1 Notations

We denote by $[n]$ the set $\{1, 2, \ldots n\}$ for a positive integer $n$. For a vector $x \in \mathbb{R}^d$, we denote by $\|x\|$ the vector's Euclidean norm. For two positive sequences $\{a_n\}_{n=1}^{\infty}$ and $\{b_n\}_{n=1}^{\infty}$, $a_n = \mathcal{O}(b_n)$ implies that there exists an absolute constant $C > 0$ such that $a_n \leq C b_n$ holds for all $n \geq 1$. Similarly, $a_n = o(b_n)$ indicates that $\lim_{n \to \infty} \frac{a_n}{b_n} = 0$.

### 2.2 Expected Risk and Empirical Risk

**Expected Risk.** In many machine learning problems, the primary goal is to develop a model with robust generalization performance. By generalization, we mean that while models are trained on a finite sample of data points, we aim for them to perform well on the entire population of data. To achieve this, we focus on minimizing a quantity known as the *expected risk*. The expected risk is the average loss across the entire population data distribution, reflecting the model's anticipated error if it had access to the complete set of possible data samples. Formally, the expected risk is defined as:

$$\mathbb{E}_{(x,y)\sim P}\left[\ell(\theta; x, y)\right] := \int \ell(\theta; x, y) dP(x, y) \tag{1}$$

where $\theta \in \mathbb{R}^d$ is the model parameter, $\ell(\theta; x, y)$ is the loss function that measures the error of the model on a single data sample $(x, y)$, and $P$ is the true distribution of the data. The gold standard goal is to find the $\theta$ that minimizes the expected risk in Eq.(1), ensuring that the model generalizes well to all data drawn from $P$.

**Empirical Risk.** In practice, however, the true distribution $P$ is typically unknown. Instead, we only work with a finite dataset $\mathcal{D}$ consisting of $n$ samples, which is denoted as $\mathcal{D} := \{(x_i, y_i)\}_{i=1}^{n}$. To approximate the expected risk, we use the empirical distribution $\hat{P}$ derived from the dataset $\mathcal{D}$. For this empirical distribution $\hat{P}$ to be a reliable approximation, we assume that the dataset $\mathcal{D}$ is representative of the true distribution $P$. This requires that each sample in the dataset $\mathcal{D}$ is equally likely and independently drawn from the true distribution $P$ (i.e., the samples $(x_i, y_i)$ are i.i.d. according to $P$). The empirical distribution $\hat{P}$ can be expressed as:

$$\hat{P}(x, y; \mathcal{D}) = \frac{1}{n} \sum_{i=1}^{n} \delta(x = x_i, y = y_i) \tag{2}$$

where $\delta$ is the Dirac-delta function. With the empirical distribution at hand, the *empirical risk* is the average loss over the given finite dataset $\mathcal{D}$. The empirical risk serves as an estimate of the expected

risk and is formally defined as:

$$\mathbb{E}_{(x,y)\sim\hat{P}}[\ell(\theta;x,y)] := \int \ell(\theta;x,y)d\hat{P}(x,y;\mathcal{D}) = \frac{1}{n}\sum_{i=1}^{n}\ell(\theta;x_i,y_i). \tag{3}$$

However, if the dataset is non-uniformly distributed, some samples may be over-represented or under-represented, leading to a biased estimate of the expected risk. To address this issue, one can use *importance sampling* (Katharopoulos & Fleuret, 2018), which adjusts the sample weights to ensure the empirical risk remains an unbiased estimate of the expected risk.

## 2.3 OBJECTIVE FUNCTION AND MINI-BATCHES

**Objective Function.** In the context of optimizing machine learning models, the objective function $f(\theta;\mathcal{D})$ is often the empirical risk shown in Eq.(3). Given a dataset $\mathcal{D} = \{(x_i,y_i)\}_{i=1}^{n}$, the objective function $f(\theta;\mathcal{D})$ is defined as, $f(\theta;\mathcal{D}) := \frac{1}{n}\sum_{i=1}^{n}\ell(\theta;x_i,y_i)$. As studied in the relevant literature of `Adam` optimization (Duchi et al., 2011; Tieleman & Hinton, 2012; Kingma & Ba, 2015; Dozat, 2016; Reddi et al., 2018), we focus on the problem setting where $f$ is convex (i.e., $\ell$ is convex). Then, the goal of the optimization problem is to find a parameter $\theta^* \in \mathbb{R}^d$ that minimizes the objective function $f(\theta;\mathcal{D})$. This problem is known as *empirical risk minimization*:

$$\theta^* \in \arg\min_{\theta\in\mathbb{R}^d} f(\theta;\mathcal{D}).$$

The gradient of the objective function $f$ with respect to $\theta$ is denoted by $g := \nabla_\theta f(\theta;\mathcal{D}) = \frac{1}{n}\sum_{i=1}^{n}g_i$, where $g_i := \nabla_\theta \ell(\theta;x_i,y_i)$ is the gradient of the loss based on the $i$-th data sample in $\mathcal{D}$.

**Mini-Batches.** When the full dataset $\mathcal{D} = \{(x_i,y_i)\}_{i=1}^{n}$ is very large (i.e., large $n$), computing the gradient over the entire dataset for each optimization iteration becomes computationally expensive. To address this, mini-batches—smaller subsets of the full dataset—are commonly used to reduce computational overhead per iteration. Consider the sequence of mini-batches $\mathcal{D}_1, \mathcal{D}_2, \ldots, \mathcal{D}_T \subseteq \mathcal{D}$ used for training, with corresponding objective functions $f_t(\theta) := f(\theta,\mathcal{D}_t)$ for each $t \in \{1,\ldots,T\}$. Let $K$ be the size of the mini-batch $\mathcal{D}_t$ for all $t$, then $\mathcal{D}_t := \{(x_{J_t^1},y_{J_t^1}),(x_{J_t^2},y_{J_t^2}),\ldots,(x_{J_t^K},y_{J_t^K})\}$, where $J_t := \{J_t^1, J_t^2, \ldots, J_t^K\} \subseteq [n]$ is the set of indices of the samples in the mini-batch $\mathcal{D}_t$. The objective function $f_t(\theta)$ for the mini-batch $\mathcal{D}_t$ is defined as the expected risk over this mini-batch:

$$f_t(\theta) = f(\theta;\mathcal{D}_t) := \int \ell(\theta;x,y)d\hat{P}(x,y;\mathcal{D}_t) \tag{4}$$

where $\hat{P}(x,y;\mathcal{D}_t)$ is the empirical distribution derived from the mini-batch $\mathcal{D}_t$. The gradient of the objective function $f_t$ with respect to $\theta$ is denoted as $g_t := \nabla_\theta f_t$.

Note that the sequence of mini-batches $\{\mathcal{D}_t\}_{t=1}^{T}$ can be selected adaptively. *Adaptive selection* involves choosing mini-batches based on results observed during previous optimization steps, potentially adjusting the importance assigned to specific samples. The empirical distribution $\hat{P}(x,y;\mathcal{D}_t)$ is significantly influenced by the method used to select the mini-batch $\mathcal{D}_t$ from the full dataset $\mathcal{D}$.

## 2.4 REGRET MINIMIZATION

**Cumulative Regret.** An online optimization method can be analyzed within the framework of regret minimization. Consider an online optimization algorithm $\pi$ that generates a sequence of model parameters $\theta_1, \ldots, \theta_T$ over $T$ iterations. The performance of $\pi$ can be compared to the optimal parameter $\theta^* \in \arg\min_{\theta\in\mathbb{R}^d} f(\theta;\mathcal{D})$, which minimizes the objective function over the full dataset $\mathcal{D}$. The cumulative regret after $T$ iterations is defined as:

$$\mathcal{R}^\pi(T) := \mathbb{E}\left[\sum_{t=1}^{T}f(\theta_t;\mathcal{D}) - T \cdot \min_{\theta\in\mathbb{R}^d}f(\theta;\mathcal{D})\right] \tag{5}$$

where the expectation is taken with respect to any stochasticity in data sampling and parameter estimation. For the optimization algorithm $\pi$ to converge to optimality, we require the cumulative regret $\mathcal{R}^\pi(T)$ to grow slower than the number of iterations $T$, specifically $\mathcal{R}^\pi(T) = o(T)$.

**Online Regret.** An alternative notion of regret is the online regret, defined over a sequence of mini-batch datasets $\{\mathcal{D}_t\}_{t=1}^T$, or equivalently, over the sequence of functions $\{f_t\}_{t=1}^T$. Specifically, the online regret of the optimization algorithm $\pi$ after $T$ iterations is given by:

$$\mathcal{R}_{\text{online}}^\pi(T) := \mathbb{E}\left[\sum_{t=1}^T f_t(\theta_t) - \min_{\theta \in \mathbb{R}^d} \sum_{t=1}^T f_t(\theta)\right]$$

where the expectation is again taken over any stochasticity in the optimization process. It is important to note that the primary focus should not solely be on minimizing the online regret. An algorithm might select $\mathcal{D}_t \subset \mathcal{D}$ in a way that allows $\pi$ to perform well on $\{\mathcal{D}_t\}_{t=1}^T$, but it may perform poorly on the full dataset $\mathcal{D}$. Therefore, our ultimate goal remains minimizing the cumulative regret $\mathcal{R}^\pi(T)$. Later, in the proof of Theorem 1, we demonstrate how minimizing the cumulative regret $\mathcal{R}^\pi(T)$ in Eq.(5) relates to minimizing the online regret $\mathcal{R}_{\text{online}}^\pi(T)$ with respect to the sequence $\{f_t\}_{t=1}^T$.

## 2.5 Related Work: Adam and Technical Issues in Convergence Guarantees

### 2.5.1 Adam Optimizer

Adam (Kingma & Ba, 2015) is a widely used first-order gradient-based optimization method that computes adaptive learning rates for each parameter by using both the first and second moment estimates of the gradients. In each iteration $t$, Adam maintains the accumulated gradients $m_t \leftarrow \beta_{1,t} m_{t-1} + (1 - \beta_{1,t}) g_t$ and the accumulated squared gradients $v_t \leftarrow \beta_2 v_{t-1} + (1 - \beta_2) g_t^2$, where $g_t$ is the gradient at iteration $t$ and $g_t^2$ represents the element-wise square of gradient $g_t$. The hyper-parameters $\beta_1, \beta_2 \in [0, 1)$ control the decay rates of $m_t$ and $v_t$, respectively. Since these moment estimates are initially biased towards zero, the estimates are corrected as $\hat{m}_t \leftarrow m_t/(1 - \beta_1^t)$ and $\hat{v}_t \leftarrow v_t/(1 - \beta_2^t)$. The Adam algorithm then updates the parameters using $\theta_t \leftarrow \theta_{t-1} - \alpha_t \frac{\hat{m}_t}{\sqrt{\hat{v}_t} + \epsilon}$, where $\epsilon$ is a small positive constant added to prevent division by zero. The key characteristic of Adam lies in its use of exponential moving average for both the gradient estimates (first-order) and the element-wise squares of gradients (second-order). This approach has shown empirical effectiveness for optimizing deep neural networks and has led to many follow-up works, such as Reddi et al. (2018), Loshchilov & Hutter (2019), Chen et al. (2020), Alacaoglu et al. (2020), and Chen et al. (2023).

### 2.5.2 Convergence of Adam

After Adam was first introduced, there was considerable debate regarding its convergence properties. In particular, Reddi et al. (2018) provided a counterexample demonstrating that Adam might fail to converge under certain conditions (see Section 3 of Reddi et al. 2018). In response, numerous variants of adaptive gradient methods have been proposed, such as AMSGrad (Reddi et al., 2018), AdamW (Loshchilov & Hutter, 2019), and AdaBelief (Zhuang et al., 2020), to address this issue and ensure convergence.

However, recent studies (Zhang et al., 2022; Défossez et al., 2022) indicate that the standard Adam algorithm itself can achieve convergence with appropriate hyperparameter choices, thereby resolving its earlier theoretical concerns. These recent works provide alternative convergence proofs for the original Adam algorithm without requiring modifications to its update rules, contingent upon hyperparameter conditions being satisfied.

### 2.5.3 Technical Issues in Adam with Bandit Sampling (Liu et al., 2020)

The work most closely related to ours is by Liu et al. (2020), who proposed AdamBS, an extension of the original Adam algorithm and proof framework incorporating a bandit sampling approach. However, the initial convergence issue present in the original proof of Adam, discussed in Section 2.5.2, also affects AdamBS, thus invalidating the convergence guarantee provided by Liu et al. (2020). Moreover, even if the alternative convergence proofs of Adam are adapted to the AdamBS framework, several other critical shortcomings persist. We summarize these remaining issues as follows:

- **AdamBS unfortunately fails to provide guarantees on convergence** despite its claims, both on the regret bound and on the effectiveness of the adaptive sample selection via the bandit approach. Specifically, the claimed regret bound in Theorem 1 of Liu et al. (2020) is

incorrect. In particular, Eq.(7) on Page 3 of the supplemental material of Liu et al. (2020) contains an error in the formula expansion.[1] This technical error is critical to their claims regarding the convergence rate of `AdamBS` and its dependence on the mini-batch size $K$

- **Their problem setting is also limited and impractical**, even if the analysis were corrected. The analysis assumes that feature vectors follow a *doubly heavy-tailed* distribution, a strong and restrictive condition that may not hold in practical scenarios. Importantly, no analysis is provided for bounded or sub-Gaussian (light-tailed) distributions, which are more commonly encountered in real-world applications.

- Despite their claim regarding mini-batch selection of size $K$, their algorithm design **allows the same sample to be selected multiple times within a single mini-batch**. This occurs because the bandit algorithm they employ is based on single-action selection rather than a combinatorial bandit approach. As a result, their method may repeatedly sample the same data points within a mini-batch. Moreover, due to this limitation, their method fails to achieve performance gains with increasing mini-batch size $K$, contradicting their claim.

- **Numerical evaluations (in Section 5) demonstrate poor performance of `AdamBS`.** Our numerical experiments across various models and datasets reveal that `AdamBS` exhibits poor and inconsistent performance. Additionally, an independent evaluation by a separate group has also reported inconsistent results for `AdamBS` (Bansal et al., 2022).

## 3 Proposed Algorithm: AdamCB

### 3.1 AdamCB Algorithm

---

**Algorithm 1:** Adam with Combinatorial Bandit Sampling (`AdamCB`)

---

**Input:** learning rate $\{\alpha_t\}_{t=1}^T$, decay rates $\{\beta_{1,t}\}_{t=1}^T$, $\beta_2$, batch size $K$, exploration parameter $\gamma \in [0, 1)$

**Initialize:** model parameters $\theta_0$, first moment estimate $m_0 \leftarrow 0$, second moment estimate $v_0 \leftarrow 0, \hat{v}_0 \leftarrow 0$, sample weights $w_{i,0} \leftarrow 1$ for all $i \in [n]$

1 **for** $t = 1$ **to** $T$ **do**
2      $J_t, p_t, S_{\text{null},t} \leftarrow$ `Batch-Selection`$(w_{t-1}, K, \gamma)$ (Algorithm 2)
3      Compute unbiased gradient estimate $g_t$ with respect to $J_t$ using Eq.(7)
4      $m_t \leftarrow \beta_{1,t} m_{t-1} + (1 - \beta_{1,t}) g_t$
5      $v_t \leftarrow \beta_2 v_{t-1} + (1 - \beta_2) g_t^2$
6      $\hat{v}_1 \leftarrow v_1, \hat{v}_t \leftarrow \max\left\{\frac{(1-\beta_{1,t})^2}{(1-\beta_{1,t-1})^2} \hat{v}_{t-1}, v_t\right\}$ if $t \geq 2$
7      $\theta_{t+1} \leftarrow \theta_t - \alpha_t \frac{m_t}{\sqrt{\hat{v}_t} + \epsilon}$
8      $w_t \leftarrow$ `Weight-Update`$(w_{t-1}, p_t, J_t, \{g_{j,t}\}_{j \in J_t}, S_{\text{null},t}, \gamma)$ (Algorithm 3)

---

We present our proposed algorithm, *Adam with Combinatorial Bandit Sampling* (`AdamCB`), which is described in Algorithm 1. The algorithm begins by initializing the sample weights $w_0 := \{w_{1,0}, w_{2,0}, \ldots, w_{n,0}\}$ uniformly, assigning an equal weight of 1 to each of $n$ training samples. At each iteration $t \in [T]$, the current sample weights $w_{t-1} = \{w_{1,t-1}, w_{2,t-1}, \ldots, w_{n,t-1}\}$ are used to determine the sample selection probabilities $p_t := \{p_{1,t}, p_{2,t}, \ldots, p_{n,t}\}$, where these probabilities are controlled with the exploration parameter $\gamma$ (Line 2). A subset of samples, denoted by $\mathcal{D}_t \subseteq \mathcal{D}$, is chosen based on these probabilities. The set of indices for samples chosen in the mini-batch $\mathcal{D}_t$ is denoted by $J_t := \{J_t^1, J_t^2, \ldots, J_t^K\} \subseteq [n]$. Using this mini-batch $\mathcal{D}_t$, an unbiased gradient estimate $g_t$ is computed (Line 3). The algorithm then updates moments estimates $m_t$, $v_t$, and $\hat{v}_t$ following the `Adam`-based update rules (Lines 4–6). The model parameters $\theta_t$ are subsequently updated based on these moment estimates (Line 7). Finally, the weights $w_{t-1}$ are adjusted to reflect the importance of each sample, improving the batch selection process in future iterations (Line 8).

---

[1] Liu et al. (2020) apply Jensen's inequality to handle the expectation of the squared norm of the sum of gradient estimates. However, the convexity assumption required for Jensen's inequality does not hold in this context, rendering this step in their proof invalid.

The following sections describe the detailed process for deriving the sample probabilities $p_t$ and selecting the mini-batch $\mathcal{D}_t = \{(x_j, y_j)\}_{j \in J_t}$ from the sample weights $w_{t-1}$ utilizing our proposed combinatorial bandit sampling.

## 3.2 BATCH SELECTION: COMBINATORIAL BANDIT SAMPLING

In our approach, we incorporate a bandit framework where each sample is treated as an arm. Since multiple samples must be selected for a mini-batch, we extend the selection process to handle multiple arms. There are two primary methods for sampling multiple arms: *with* replacement or *without* replacement. The previous method, `AdamBS` (Liu et al., 2020), samples multiple arms *with* replacement. In contrast, our proposed method, `AdamCB`, employs a combinatorial bandit algorithm to sample multiple arms *without* replacement, achieved by `Batch-Selection` (Algorithm 2).

---

**Algorithm 2:** `Batch-Selection`

**Input:** Sample weights $w_{t-1}$, batch size $K$, exploration parameter $\gamma \in [0, 1)$
1  Set $C \leftarrow (1/K - \gamma/n)/(1 - \gamma)$
2  **if** $\max_{i \in [n]} w_{i,t-1} \geq C \sum_{i=1}^{n} w_{i,t-1}$ **then**
3       Let $\bar{w}_{t-1}$ be a sorted list of $\{w_{i,t-1}\}_{i=1}^{n}$ in descending order
4       Set $S \leftarrow \sum_{i=1}^{n} \bar{w}_{i,t-1}$
5       **for** $i = 1$ **to** $n$ **do**
6           Compute $\tau \leftarrow C \cdot S/(1 - i \cdot C)$
7           **if** $\bar{w}_{i,t-1} < \tau$ **then** break, **else** update $S \leftarrow S - \bar{w}_{i,t-1}$
8       Set $S_{\text{null},t} \leftarrow \{i : w_{i,t-1} \geq \tau\}$ and $w_{i,t-1} = \tau$ for $i \in S_{\text{null},t}$
9  **else**
10      Set $S_{\text{null},t} \leftarrow \emptyset$
11 Set $p_{i,t} \leftarrow K \left( (1-\gamma) \frac{w_{i,t-1}}{\sum_{j=1}^{n} w_{j,t-1}} + \frac{\gamma}{n} \right)$ for all $i \in [n]$
12 Set $J_t \leftarrow \texttt{DepRound}(K, (p_{1,t}, p_{2,t}, \ldots, p_{n,t}))$ (Algorithm 6)
13 **return** $J_t, p_t, S_{null,t}$

---

**Weight Adjustment (Lines 2–10).** Unlike single-arm selection bandit approach like `AdamBS`, where $\sum_{i=1}^{n} p_{i,t} = 1$, because only one sample is selected at a time, `AdamCB` must select $K$ samples simultaneously for a mini-batch. Therefore, it is natural to scale the sum of the probabilities to $K$, reflecting the expected number of samples selected in each round.[2] Allowing the sum of probabilities to equal $K$ can lead to individual probabilities $p_{i,t}$ exceeding 1, especially when certain samples are assigned significantly higher weights due to their importance (or gradient magnitude). To ensure valid probabilities and prevent any sample from being overrepresented, `AdamCB` introduces a threshold $\tau$. If a weight $w_{i,t-1}$ exceeds $\tau$, the index $i$ is added to a null set $S_{\text{null},t}$, effectively removing it from active consideration for selection. The probabilities of the remaining samples are adjusted to redistribute the excess weight while ensuring the sum of probabilities remains $K$.

**Probability Computation (Line 11).** After adjusting the weights, the probabilities $p_t$ for selecting each sample are computed using the adjusted weights $w_{t-1}$ and the exploration parameter $\gamma$. This computation balances the need to *exploit* samples with higher weights (more likely to provide useful gradients) and *explore* other samples. The inclusion of $K$ in the scaling ensures that the sum of probabilities matches the batch size: $\sum_{i=1}^{n} p_{i,t} = K$.

**Mini-batch Selection (Line 12).** The final selection of $K$ distinct samples for the mini-batch is performed using `DepRound` (Algorithm 6), originally proposed by Gandhi et al. (2006) and later adapted by Uchiya et al. (2010). `DepRound` efficiently selects $K$ distinct samples from a set of $n$ samples, ensuring that each sample $i$ is selected with probability $p_{i,t}$. The algorithm has a computational complexity of $\mathcal{O}(n)$, which is significantly more efficient than a naive approach requiring consideration of all possible combinations with a complexity of at least $\binom{n}{K}$.

---

[2]If the sum of probabilities were constrained to 1, the algorithm would need to perform additional rescaling or sampling adjustments. Instead, directly setting $\sum_{i=1}^{n} p_{i,t} = K$ aligns the probability distribution with the batch-level selection requirements.

### 3.3 COMPUTING UNBIASED GRADIENT ESTIMATES

Given the mini-batch data $\mathcal{D}_t = \{(x_j, y_j)\}_{j \in J_t}$ from Algorithm 2, and since $p_t$ is a probability over the full dataset $\mathcal{D}$, and $\mathcal{D}_t$ is sampled according to $p_t$, we employ an importance sampling technique to compute the empirical distribution $\hat{P}$ for $\mathcal{D}_t$:

$$\hat{P}(x, y; \mathcal{D}_t) := \frac{1}{K} \sum_{j \in J_t} \frac{\delta(x = x_j, y = y_j)}{n p_{j,t}} \tag{6}$$

where $\delta$ is the Dirac-delta function. This formulation ensures that the empirical distribution $\hat{P}$ for the mini-batch $\mathcal{D}_t$ closely approximates the original empirical distribution $\hat{P}(x, y; \mathcal{D})$ defined over the full dataset $\mathcal{D}$, as expressed in Eq.(2). According to the empirical distribution $\hat{P}(x, y; \mathcal{D}_t)$ in Eq.(6), the online objective function $f_t$ corresponding to the mini-batch $\mathcal{D}_t$ (as defined in Eq.(4)) can be computed as

$$f_t(\theta) = f(\theta; \mathcal{D}_t) = \int \ell(\theta; x, y) d\hat{P}(x, y; \mathcal{D}_t) = \frac{1}{K} \sum_{j \in J_t} \frac{\ell(\theta; x_j, y_j)}{n p_{j,t}}.$$

This implies that the gradient $g_t = \nabla_\theta f_t(\theta)$ obtained from the mini-batch $\mathcal{D}_t$ at iteration $t$ is computed as follows:

$$g_t = \nabla_\theta f_t(\theta) = \frac{1}{K} \sum_{j \in J_t} \frac{\nabla_\theta \ell(\theta; x_j, y_j)}{n p_{j,t}} = \frac{1}{K} \sum_{j \in J_t} \frac{g_{j,t}}{n p_{j,t}} \tag{7}$$

Here, we denote the gradients for each individual sample in the mini-batch $\mathcal{D}_t$ as $\{g_{j,t}\}_{j \in J_t}$, where $J_t$ is the set of indices for $\mathcal{D}_t$. In stochastic optimization methods like `SGD` and `Adam`, it is crucial to use an unbiased gradient estimate when updating the moment vectors. We can easily show that $g_t$ is an unbiased estimate of the true gradient $g$ over the entire dataset by taking the expectation over $p_t$, i.e, $\mathbb{E}_{p_t}[g_t] = g$. The unbiased gradient estimate $g_t$ in Eq.(7) is then used to update the first moment estimate $m_t$ and the second moment estimate $v_t$ in each iteration of the algorithm.

### 3.4 UPDATE OF SAMPLE WEIGHTS

The final step in each iteration of Algorithm 1 involves updating the sample weights $w_t$. Treating the optimization problem as an adversarial semi-bandit, our partial feedback consists only of the gradients $\{g_{j,t}\}_{j \in J_t}$. The loss $\ell_{i,t}$ occurred when the $i$-th arm is pulled is computed based on the norm of the gradient $\|g_{i,t}\|$. Specifically, the loss $\ell_{i,t}$ is always non-negative and inversely related to $\|g_{i,t}\|$. This implies that samples with smaller gradient norms are assigned lower weights, while samples with larger gradient norms are more likely to be selected in future iterations.

---

**Algorithm 3: `Weight-Update`**

**Input:** $w_{t-1}, p_t, J_t, \{g_{j,t}\}_{j \in J_t}, S_{\text{null},t}, \gamma \in [0, 1)$

1 **for** $j = 1$ **to** $n$ **do**

2     Compute loss $\ell_{j,t} = \frac{p_{\min}^2}{L^2} \left( -\frac{\|g_{j,t}\|^2}{(p_{j,t})^2} + \frac{L^2}{p_{\min}^2} \right)$ if $j \in J_t$; otherwise $\ell_{j,t} = 0$

3     **if** $j \notin S_{null,t}$ **then**

4         $w_{j,t} \leftarrow w_{j,t-1} \exp\left( -K\gamma \ell_{j,t}/n \right)$

5 **return** $w_t$

---

## 4 REGRET ANALYSIS

In this section, we present a regret analysis for our proposed algorithm, `AdamCB`. We begin by introducing the standard assumptions commonly used in the analysis of optimization algorithms.

**Assumption 1** (Bounded gradient). *There exists $L > 0$ such that $\|g_{i,t}\| \leq L$ for all $i \in [n]$ and $t \in [T]$.*

**Assumption 2** (Bounded parameter). *There exists $D > 0$ such that $\|\theta_s - \theta_t\| \leq D$ for any $s, t \in [T]$.*

**Discussion of Assumptions.** Both Assumptions 1 and 2 are the standard assumptions in the relevant literature that studies the regret bounds of `Adam`-based optimization (Kingma & Ba, 2015; Reddi et al., 2018; Luo et al., 2019; Liu et al., 2020; Chen et al., 2020). A closely related work (Liu et al., 2020) relies on the additional stronger assumption of a doubly heavy-tailed feature distribution. In contrast, the regret bound for `AdamCB` is derived using only these two standard assumptions.

## 4.1 REGRET BOUND OF ADAMCB

**Theorem 1** (Regret bound of `AdamCB`). *Suppose Assumptions 1-2 hold, and we run `AdamCB` for a total $T$ iterations with $\alpha_t = \frac{\alpha}{\sqrt{t}}$ and with $\beta_{1,t} := \beta_1 \lambda^{t-1}$, $\lambda \in (0,1)$. Then, the cumulative regret of `AdamCB` (Algorithm 1) with batch size $K$ is upper-bounded by*

$$\mathcal{O}\left(d\sqrt{T} + \frac{\sqrt{d}}{n^{3/4}}\left(\frac{T}{K}\ln\frac{n}{K}\right)^{1/4}\right).$$

**Discussion of Theorem 1.** Theorem 1 establishes that the cumulative regret bound of `AdamCB` is sub-linear in $T$, i.e., $\mathcal{R}^\pi(T) = o(T)$. Hence, `AdamCB` is guaranteed to converge to the optimal solution. The first term in the regret bound, $d\sqrt{T}$, which is commonly shared by the results in all `Adam`-based methods (Kingma & Ba, 2015; Reddi et al., 2018; Liu et al., 2020). The second term, $(\sqrt{d}/n^{3/4})\cdot((T/K)\ln{(n/K)})^{1/4}$, illustrates the impact of the number of samples $n$ as well as the batch size $K$ on regret. As the number of samples $n$ increases, this term decreases, suggesting that having more data generally helps in reducing regret (hence converging faster to optimality). Similarly, increasing the batch size $K$ also decreases this term, reflecting that larger mini-batches can reduce the variance in gradient estimates, thus improving the performance.

## 4.2 PROOF SKETCH OF THEOREM 1

In this section, we present the proof sketch of the regret bound in Theorem 1. The proof start by decomposing the cumulative regret $\mathcal{R}^\pi(T)$ into three parts: the cumulative online regret $\mathcal{R}^\pi_{\text{online}}(T)$ and auxiliary terms (A) and (B), as shown below:

$$\mathcal{R}^\pi(T) = \mathcal{R}^\pi_{\text{online}}(T) + \underbrace{\mathbb{E}\left[\sum_{t=1}^T \left(f(\theta_t; \mathcal{D}) - f(\theta_t; \mathcal{D}_t)\right)\right]}_{\text{(A)}} + \underbrace{\mathbb{E}\left[\min_{\theta \in \mathbb{R}^d}\sum_{t=1}^T f(\theta; \mathcal{D}_t) - T \cdot \min_{\theta \in \mathbb{R}^d} f(\theta; \mathcal{D})\right]}_{\text{(B)}} \quad (8)$$

We now prove the following two key lemmas to bound the online regret $\mathcal{R}^\pi_{\text{online}}(T)$.

**Lemma 1.** *Suppose Assumptions 1-2 hold. `AdamCB` (Algorithm 1) with a mini-batch of size $K$, which is formed dynamically by distribution $p_t$, achieves the following upper-bound for the cumulative online regret $\mathcal{R}^\pi_{online}(T)$ over $T$ iterations,*

$$\mathcal{R}^\pi_{online}(T) \leq \rho_1 d\sqrt{T} + \sqrt{d}\rho_2\sqrt{\frac{1}{n^2 K}\sum_{t=1}^T \mathbb{E}_{p_t}\left[\sum_{j \in J_t}\frac{\|g_{j,t}\|^2}{(p_{j,t})^2}\right]} + \rho_3$$

*where $\rho_1$, $\rho_2$, and $\rho_3$ are constants (See Appendix B.2).*

Lemma 1 provides an upper bound for the cumulative online regret over $T$ iterations. This lemma shows that $p_t$ affects the upper bound of $\mathcal{R}^\pi_{\text{online}}(T)$. Hence, we wish to choose $p_t$ that could lead to minimizing the upper bound. The following key lemma shows that it can be achieved by a combinatorial semi-bandit approach, adapted from `EXP3` (Auer et al., 2002).

**Lemma 2.** *Suppose Assumptions 1-2 hold. If we set $\gamma = \min\left\{1, \sqrt{\frac{n\ln{(n/K)}}{(e-1)TK}}\right\}$, the batch selection (Algorithm 2) and the weight update rule (Algorithm 3) following `AdamCB` (Algorithm 1) implies*

$$\sum_{t=1}^T \mathbb{E}_{p_t}\left[\sum_{j \in J_t}\frac{\|g_{j,t}\|^2}{(p_{j,t})^2}\right] - \min_{p_t}\sum_{t=1}^T \mathbb{E}_{p_t}\left[\sum_{j \in J_t}\frac{\|g_{j,t}\|^2}{(p_{j,t})^2}\right] = \mathcal{O}\left(\sqrt{KnT\ln\frac{n}{K}}\right)$$

Table 1: Comparison of Regret Bounds

| Optimizer | Regret Bound |
|---|---|
| AdamX (Tran et al., 2019) (variant of Adam[†]) | $\mathcal{O}\big(d\sqrt{T} + \frac{\sqrt{d}}{n^{1/2}}\sqrt{T}\big)$ |
| AdamBS (Liu et al., 2020) (corrected[‡]) | $\mathcal{O}\big(d\sqrt{T} + \frac{\sqrt{d}}{n^{3/4}}\,(T\ln n)^{\frac{1}{4}}\big)$ |
| AdamCB (**Ours**) | $\mathcal{O}\big(d\sqrt{T} + \frac{\sqrt{d}}{n^{3/4}}\,\big(\frac{T}{K}\ln\frac{n}{K}\big)^{\frac{1}{4}}\big)$ |

[†] While the convergence of Adam (Kingma & Ba, 2015) has been correctly established by Zhang et al. (2022), the results in Zhang et al. (2022) do not provide regret guarantees. For fair comparisons of regret, we use a slight modification of Adam, AdamX (Tran et al., 2019), for which correct regret analysis is feasible.
[‡] Similarly, for AdamBS(Liu et al., 2020), since the existing proof of regret guarantee contains technical errors, we present a corrected proof, resulting in a new regret bound (Theorem 3).

Lemma 2 bounds the difference between the expected cumulative loss of the chosen mini-batch and the optimal mini-batch, showing sub-linear growth in $T$ with dependence on the batch size $K$. Combining Lemma 1 and Lemma 2, we can bound the cumulative online regret $\mathcal{R}^\pi_{\text{online}}(T)$, which also grows sub-linearly in $T$. Proofs of Lemma 1 and Lemma 2 are in Appendix B.2 and Appendix B.3, respectively. The discrepancy terms (A) and (B) in Eq.(8) capture the difference between the full dataset $\mathcal{D}$ and the mini-batches $\{\mathcal{D}_t\}_{t=1}^T$, and are also bounded sub-linearly in $T$ (See Lemma 11 in Appendix B.4). Since the cumulative regret $\mathcal{R}^\pi(T)$ is decomposed into the online regret $\mathcal{R}^\pi_{\text{online}}(T)$ with additional sub-linear terms, we obtain the cumulative regret bound for AdamCB.

### 4.3 Comparisons with Adam and AdamBS

Our main goal is to demonstrate that the convergence rate of AdamCB (Algorithm 1) is provably more efficient than those of the existing Adam-based methods including ones that employ *uniform sampling* and AdamBS (Liu et al., 2020) that utilizes *(non-combinatorial) bandit sampling*. Since the claimed regret bounds in the original Adam and AdamBS are invalid, we provide new regret bounds for both a variant of Adam, called AdamX (Tran et al., 2019) and AdamBS in Theorems 2 and 3, respectively, which may be of independent interest.

For comparison purposes, we additionally introduce the following assumption:

**Assumption 3** (Bounded variance of gradient). *There exists $\sigma > 0$ such that $\mathrm{Var}(\|g_{i,t}\|) \leq \sigma^2$ for all $i \in [n]$ and $t \in [T]$*

Assumption 3 is commonly used in the previous literature (Reddi et al., 2016; Nguyen et al., 2018; Zou et al., 2019; Patel et al., 2022). It is important to note that Assumption 3 is not required for the analysis of our algorithm in Theorem 1. Rather, we employ the assumption to fairly compare with corrected results for the existing Adam-based methods (Tran et al., 2019; Liu et al., 2020).

Under Assumptions 1, 2, and 3, the regret bound for the Adam variant (AdamX) *using uniform sampling* is given by $\mathcal{O}(d\sqrt{T} + n^{-1/2}\sqrt{dT})$ (Theorem 2 in Appendix C), while the regret bound for a corrected version of AdamBS *using (non-combinatorial) bandit sampling* is $\mathcal{O}\big(d\sqrt{T} + n^{-3/4}\sqrt{d}(T\ln n)^{1/4}\big)$ (Theorem 3 in Appendix D) when Assumptions 1 and 2 hold. The comparisons of regret bounds are outlined in Table 1.

**Faster Convergence of AdamCB.** The second term in the regret bound of AdamX exhibits a dependence on $n^{-1/2}$, which is the rate of regret decrease as the dataset size increases. However, this reduction in regret occurs at a slower rate compared to bandit-based sampling methods. Both AdamBS (corrected) and AdamCB achieve an improved $n^{-3/4}$ dependency, resulting in a faster convergence. When comparing the two bandit-based sampling methods, AdamCB surpasses AdamBS (corrected) in terms of convergence, particularly by the factor of the batch size $K$. That is, as far as regret performance is concerned, AdamBS does not benefit from multiple samples in batch while our AdamCB enjoys faster convergence. Hence, AdamCB is not only the first algorithm with correct performance guarantees for AdamX with adaptive batch selection, but to our best knowledge, also the method with the fastest convergence guarantees in terms of regret performance.

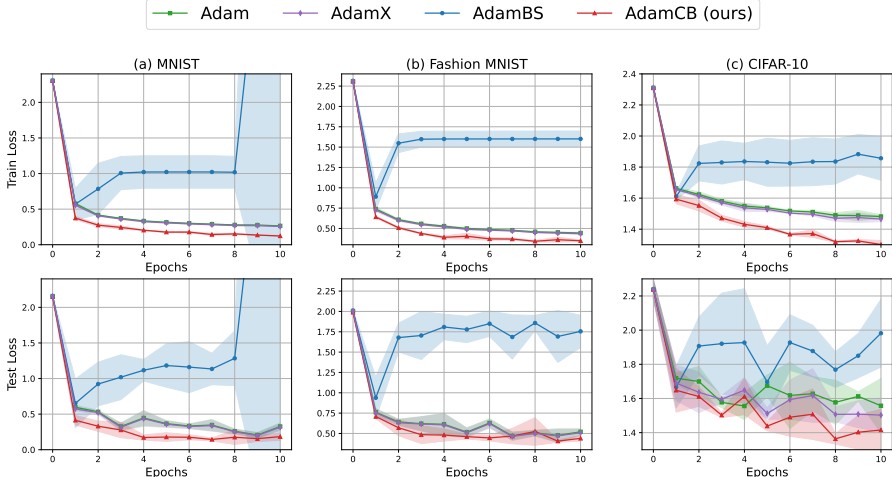

Figure 1: Performances with MLP model on MNIST, Fashion MNIST, and CIFAR10

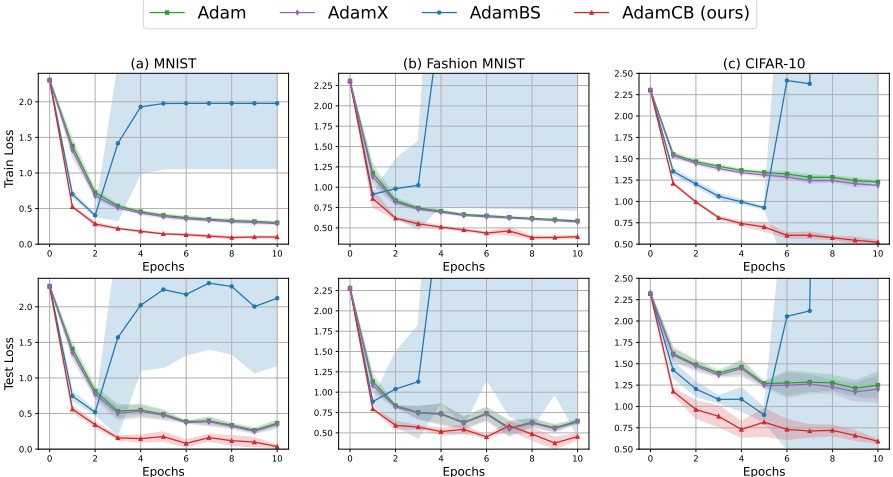

Figure 2: Performances with CNN model on MNIST, Fashion MNIST, and CIFAR10

## 5 NUMERICAL EXPERIMENTS

**Experimental Setup.** To evaluate our proposed algorithm, `AdamCB`, we conduct experiments using deep neural networks, including multilayer perceptrons (MLP) and convolutional neural networks (CNN), on three benchmark datasets: MNIST, Fashion MNIST, and CIFAR10. Comparisons are made with `Adam`, `AdamX` and `AdamBS`, with all experiments implemented in PyTorch. Performance is assessed by plotting training and test losses over epochs, with training loss calculated on the full dataset and test loss calculated on the held-out validation data set. Results represent the average of five runs with different random seeds, including standard deviations. All methods use the same hyperparameters: $\beta_1 = 0.9$, $\beta_2 = 0.999$, $\gamma = 0.4$, $K = 128$, and $\alpha = 0.001$. Additional experimental details are provided in Appendix F.

**Results.** Figures 1 and 2 show that `AdamCB` consistently outperforms `Adam`, `AdamX` and `AdamBS`, demonstrating faster reductions in both training and test losses across all datasets. These results suggest that combinatorial bandit sampling is more effective than uniform sampling for performance optimization. Attempts to replicate the results of `AdamBS` from Liu et al. (2020) revealed inconsistent outcomes, with significant fluctuations in losses, indicating potential instability and divergence. In contrast, `AdamCB` exhibits consistent convergence across all datasets, highlighting its superior performance and practical efficiency compared to `Adam`, `AdamX` and `AdamBS`. Additional experimental results in Appendix F further reinforce the superior performance of `AdamCB`.

REPRODUCIBILITY STATEMENT

For each theoretical result, we present the complete set of assumptions in the main paper and the detailed proofs of the main results are provided in the appendix, along with experimental details and additional experiments in Appendix F to reproduce the main experimental results.

ACKNOWLEDGEMENTS

The authors would like to thank Se Young Chun and Byung Hyun Lee for bringing the previous work (Liu et al., 2020) to our attention. This work was supported by the National Research Foundation of Korea (NRF) grant funded by the Korea government (MSIT) (No. RS-2022-NR071853 and RS-2023-00222663) and by AI-Bio Research Grant through Seoul National University.

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

# Appendix

## A AUXILIARY LEMMAS

**Definition 1.** *A function $f : \mathbb{R}^d \to \mathbb{R}$ is **convex** if for all $u, v \in \mathbb{R}^d$, and all $\lambda \in [0, 1]$,*
$$\lambda f(u) + (1 - \lambda)f(v) \geq f(\lambda u + (1 - \lambda)v)$$

**Lemma 3.** *If a function $f : \mathbb{R}^d \to \mathbb{R}$ is convex, then for all $u, v \in \mathbb{R}^d$,*
$$f(v) \geq f(u) + \nabla f(u)^\top (v - u)$$
*where $(-)^\top$ denotes the transpose of $(-)$.*

**Lemma 4** (Cauchy-Schwarz inequality). *For all $n \geq 1$, $a_i, b_i \in \mathbb{R}$, $(1 \leq i \leq n)$,*
$$\left( \sum_{i=1}^n a_i b_i \right)^2 \leq \left( \sum_{i=1}^n a_i^2 \right) \left( \sum_{i=1}^n b_i^2 \right)$$

**Lemma 5** (Taylor series). *For $\alpha \in \mathbb{R}$, and $0 \leq \alpha \leq 1$,*
$$\sum_{t \geq 1} \alpha^t = \frac{1}{1 - \alpha} \qquad and \qquad \sum_{t \geq 1} t \alpha^{t-1} = \frac{1}{(1 - \alpha)^2}$$

**Lemma 6** (Upper bound for the harmonic series). *For $N \in \mathbb{N}$,*
$$\sum_{n=1}^N \frac{1}{n} \leq \ln N + 1 \qquad and \qquad \sum_{n=1}^N \frac{1}{\sqrt{n}} \leq 2\sqrt{N}$$

**Lemma 7.** *For all $n \in \mathbb{N}$, and $a_i, b_i \in \mathbb{R}$ such that $a_i \geq 0$ and $b_i > 0$ for all $i \in [n]$,*
$$\frac{\sum_{i=1}^n a_i}{\sum_{j=1}^n b_j} \leq \sum_{i=1}^n \frac{a_i}{b_i}$$

## B PROOF FOR ADAMCB REGRET BOUND

In this section, we provide proofs of key lemmas, Lemma 1 and Lemma 2. They are needed to prove Theorem 1, which shows the regret bound for `AdamCB`. In the last of this section, we present the proof for Theorem 1.

### B.1 AUXILIARY LEMMAS FOR LEMMA 1

We first present auxiliary lemmas and proofs for Lemma 1. Our proofs basically follow arguments as in Tran et al. (2019). For the sake of completeness, all lemmas from Tran et al. (2019) are restated with our problem setting.

**Lemma 8.** *For all $t \geq 1$, we have*
$$\hat{v}_t = \max \left\{ \frac{(1 - \beta_{1,t})^2}{(1 - \beta_{1,s})^2} v_s \text{ for all } 1 \leq s \leq t \right\}, \tag{9}$$
*where $\hat{v}_t$ is in `AdamCB` (Algorithm 1).*

*Proof.* Prove by induction on $t$. Recall that by the update rule on $\hat{v}_t$, we have $\hat{v}_1 \leftarrow v_1$, $\hat{v}_t \leftarrow \max \left\{ \frac{(1 - \beta_{1,t})^2}{(1 - \beta_{1,t-1})^2} \hat{v}_{t-1}, v_t \right\}$ if $t \geq 2$. Thus,
$$\hat{v}_2 = \max \left\{ \frac{(1 - \beta_{1,2})^2}{(1 - \beta_{1,1})^2} \hat{v}_1, v_2 \right\}$$
$$= \max \left\{ \frac{(1 - \beta_{1,2})^2}{(1 - \beta_{1,1})^2} v_1, v_2 \right\}$$
$$= \max \left\{ \frac{(1 - \beta_{1,2})^2}{(1 - \beta_{1,s})^2} v_s, 1 \leq s \leq 2 \right\}$$

which we proved for the case when $t = 2$ in Eq.(9). Now, assume that

$$\hat{v}_{t-1} = \max\left\{ \frac{(1 - \beta_{1,t-1})^2}{(1 - \beta_{1,s})^2} v_s \text{ for all } 1 \leq s \leq t - 1 \right\},$$

and Eq.(9) holds for all $1 \leq j \leq t - 1$. By the update rule on $\hat{v}_t$,

$$
\begin{aligned}
\hat{v}_t &= \max\left\{ \frac{(1 - \beta_{1,t})^2}{(1 - \beta_{1,t-1})^2} \hat{v}_{t-1}, v_t \right\} \\
&= \max\left\{ \frac{(1 - \beta_{1,t})^2}{(1 - \beta_{1,t-1})^2} \left( \max\left\{ \frac{(1 - \beta_{1,t-1})^2}{(1 - \beta_{1,s})^2} v_s \text{ for all } 1 \leq s \leq t - 1 \right\} \right), v_t \right\} \\
&= \max\left\{ \max\left\{ \frac{(1 - \beta_{1,t})^2}{(1 - \beta_{1,t-1})^2} \frac{(1 - \beta_{1,t-1})^2}{(1 - \beta_{1,s})^2} v_s \text{ for all } 1 \leq s \leq t - 1 \right\}, \frac{(1 - \beta_{1,t})^2}{(1 - \beta_{1,t-1})^2} v_t \right\} \\
&= \max\left\{ \max\left\{ \frac{(1 - \beta_{1,t})^2}{(1 - \beta_{1,s})^2} v_s \text{ for all } 1 \leq s \leq t - 1 \right\}, \frac{(1 - \beta_{1,t})^2}{(1 - \beta_{1,t-1})^2} v_t \right\} \\
&= \max\left\{ \frac{(1 - \beta_{1,t})^2}{(1 - \beta_{1,s})^2} v_s \text{ for all } 1 \leq s \leq t \right\}
\end{aligned}
$$

which ends the proof. $\qquad\square$

**Lemma 9.** *For all $t \geq 1$, we have*

$$\sqrt{\hat{v}_t} \leq \frac{L}{\gamma(1 - \beta_1)}$$

*where $\hat{v}_t$ is in* **AdamCB** *(Algorithm 1).*

*Proof.* By Lemma 8,

$$\hat{v}_t = \max\left\{ \frac{(1 - \beta_{1,t})^2}{(1 - \beta_{1,s})^2} v_s \text{ for all } 1 \leq s \leq t \right\}$$

Therefore, there is some $1 \leq s \leq t$ such that $\hat{v}_t = \frac{(1-\beta_{1,t})^2}{(1-\beta_{1,s})^2} v_s$. Recall that by the update rule on $v_t$, we have $v_t \leftarrow \beta_2 v_{t-1} + (1 - \beta_2) g_t^2$. This implies

$$v_t = (1 - \beta_2) \sum_{k=1}^{t} \beta_2^{t-k} g_k^2$$

Hence,

$$
\begin{aligned}
\sqrt{\hat{v}_t} &= \sqrt{\frac{(1 - \beta_{1,t})^2}{(1 - \beta_{1,s})^2} v_s} \\
&= \sqrt{1 - \beta_2} \left( \frac{1 - \beta_{1,t}}{1 - \beta_{1,s}} \right) \sqrt{\sum_{k=1}^{s} \beta_2^{s-k} g_k^2} \\
&\leq \sqrt{1 - \beta_2} \left( \frac{1 - \beta_{1,t}}{1 - \beta_{1,s}} \right) \sqrt{\sum_{k=1}^{s} \beta_2^{s-k} (\max_{1 \leq r \leq s} \|g_r\|)^2}
\end{aligned}
$$

Recall the unbiased gradient estimate $g_t$ in Eq.(7),

$$g_t = \frac{1}{K} \sum_{j \in J_t} \frac{g_{j,t}}{n p_{j,t}}$$

By the triangle inequality property of norms and the fact that $p_{i,t} \geq \gamma/n$ and $\|g_{i,t}\| \leq L$ for all $i \in [n]$ and $t \in [T]$ from Assumption 1, the unbiased gradient estimate is bounded by $L/\gamma$, i.e,

$\|g_t\| \leq L/\gamma$. Therefore,

$$\sqrt{\hat{v}_t} \leq (L/\gamma)\sqrt{1-\beta_2} \left( \frac{1-\beta_{1,t}}{1-\beta_{1,s}} \right) \sqrt{\sum_{k=1}^{s} \beta_2^{s-k}}$$

$$\leq (L/\gamma)\sqrt{1-\beta_2} \left( \frac{1-\beta_{1,t}}{1-\beta_{1,s}} \right) \frac{1}{\sqrt{1-\beta_2}}$$

$$= (L/\gamma) \left( \frac{1-\beta_{1,t}}{1-\beta_{1,s}} \right)$$

$$\leq \frac{L}{\gamma(1-\beta_1)}$$

which ends the proof. $\qquad\square$

**Lemma 10.** *For the parameter settings and conditions assumed in Lemma 1, we have*

$$\sum_{t=1}^{T} \frac{m_{t,u}^2}{\sqrt{t\hat{v}_{t,u}}} \leq \frac{\sqrt{\ln T + 1}}{(1-\beta_1)\sqrt{1-\beta_2}(1-\eta)} \|g_{1:T,u}\|$$

*Proof.* Recall that by the update rule on $m_t, v_t$, we have $m_t \leftarrow \beta_{1,t} m_{t-1} + (1-\beta_{1,t})g_t$ and $v_t \leftarrow \beta_2 v_{t-1} + (1-\beta_2)g_t^2$. This implies

$$m_t = \sum_{k=1}^{t}(1-\beta_{1,k}) \left( \prod_{r=k+1}^{t} \beta_{1,r} \right) g_k, \quad v_t = (1-\beta_2)\sum_{k=1}^{t} \beta_2^{t-k} g_k^2$$

Since for all $t \geq 1$, $\hat{v}_{t,u} \geq v_{t,u}$ by Lemma 8, we have

$$\frac{m_{t,u}^2}{\sqrt{t\hat{v}_{t,u}}} \leq \frac{m_{t,u}^2}{\sqrt{tv_{t,u}}}$$

$$= \frac{\left[ \sum_{k=1}^{t}(1-\beta_{1,k}) \left( \prod_{r=k+1}^{t} \beta_{1,r} \right) g_{k,u} \right]^2}{\sqrt{(1-\beta_2)t \sum_{k=1}^{t} \beta_2^{t-k} g_{k,u}^2}}$$

$$\leq \frac{\left( \sum_{k=1}^{t}(1-\beta_{1,k})^2 \left( \prod_{r=k+1}^{t} \beta_{1,r} \right) \right) \left( \sum_{k=1}^{t} \left( \prod_{r=k+1}^{t} \beta_{1,r} \right) g_{k,u}^2 \right)}{\sqrt{(1-\beta_2)t \sum_{k=1}^{t} \beta_2^{t-k} g_{k,u}^2}}$$

$$\leq \frac{\left( \sum_{k=1}^{t} \beta_1^{t-k} \right) \left( \sum_{k=1}^{t} \beta_1^{t-k} g_{k,u}^2 \right)}{\sqrt{(1-\beta_2)t \sum_{k=1}^{t} \beta_2^{t-k} g_{k,u}^2}}$$

$$\leq \frac{1}{(1-\beta_1)\sqrt{1-\beta_2}} \frac{\sum_{k=1}^{t} \beta_1^{t-k} g_{k,u}^2}{\sqrt{t \sum_{k=1}^{t} \beta_2^{t-k} g_{k,u}^2}}$$

where the second inequality is by Lemma 4, the third inequality is from the fact that $\beta_{1,k} \leq 1$ and $\beta_{1,k} \leq \beta_1$ for all $1 \leq k \leq T$, and the fourth inequality is obtained by applying Lemma 5 to

$\sum_{k=1}^{t} \beta_1^{t-k}$. Therefore,

$$
\frac{m_{t,u}^2}{\sqrt{t\hat{v}_{t,u}}} \leq \frac{1}{(1-\beta_1)\sqrt{1-\beta_2}\sqrt{t}} \frac{\sum_{k=1}^{t} \beta_1^{t-k} g_{k,u}^2}{\sqrt{\sum_{k=1}^{t} \beta_2^{t-k} g_{k,u}^2}}
$$

$$
\leq \frac{1}{(1-\beta_1)\sqrt{1-\beta_2}\sqrt{t}} \sum_{k=1}^{t} \frac{\beta_1^{t-k} g_{k,u}^2}{\sqrt{\beta_2^{t-k} g_{k,u}^2}}
$$

$$
= \frac{1}{(1-\beta_1)\sqrt{1-\beta_2}\sqrt{t}} \sum_{k=1}^{t} \frac{\beta_1^{t-k}}{\sqrt{\beta_2^{t-k}}} |g_{k,u}|
$$

$$
= \frac{1}{(1-\beta_1)\sqrt{1-\beta_2}\sqrt{t}} \sum_{k=1}^{t} \eta^{t-k} |g_{k,u}|
$$

where the second inequality is by Lemma 7 and we define $\eta := \frac{\beta_1}{\sqrt{\beta_2}}$. Therefore,

$$
\sum_{t=1}^{T} \frac{m_{t,u}^2}{\sqrt{t\hat{v}_{t,u}}} = \frac{1}{(1-\beta_1)\sqrt{1-\beta_2}} \sum_{t=1}^{T} \frac{1}{\sqrt{t}} \sum_{k=1}^{t} \eta^{t-k} |g_{k,u}| \tag{10}
$$

It is sufficient to consider $\sum_{t=1}^{T} \frac{1}{\sqrt{t}} \sum_{k=1}^{t} \eta^{t-k} |g_{k,u}|$. Firstly, this can be expanded as:

$$
\sum_{t=1}^{T} \frac{1}{\sqrt{t}} \sum_{k=1}^{t} \eta^{t-k} |g_{k,u}| = \eta^0 |g_{1,u}|
$$

$$
+ \frac{1}{\sqrt{2}} \left[ \eta^1 |g_{1,u}| + \eta^0 |g_{2,u}| \right]
$$

$$
+ \frac{1}{\sqrt{3}} \left[ \eta^2 |g_{1,u}| + \eta^1 |g_{2,u}| + \eta^0 |g_{3,u}| \right]
$$

$$
+ \cdots
$$

$$
+ \frac{1}{\sqrt{T}} \left[ \eta^{T-1} |g_{1,u}| + \eta^{T-2} |g_{2,u}| + \cdots + \eta^0 |g_{T,u}| \right]
$$

Changing the role of $|g_{1,u}|$ as the common factor, we obtain,

$$
\sum_{t=1}^{T} \frac{1}{\sqrt{t}} \sum_{k=1}^{t} \eta^{t-k} |g_{k,u}| = |g_{1,u}| \left( \eta^0 + \frac{1}{\sqrt{2}} \eta^1 + \frac{1}{\sqrt{3}} \eta^2 + \cdots + \frac{1}{\sqrt{T}} \eta^{T-1} \right)
$$

$$
+ |g_{2,u}| \left( \frac{1}{\sqrt{2}} \eta^0 + \frac{1}{\sqrt{3}} \eta^1 + \cdots + \frac{1}{\sqrt{T}} \eta^{T-2} \right)
$$

$$
+ |g_{3,u}| \left( \frac{1}{\sqrt{3}} \eta^0 + \frac{1}{\sqrt{4}} \eta^1 + \cdots + \frac{1}{\sqrt{T}} \eta^{T-3} \right)
$$

$$
+ \cdots
$$

$$
+ |g_{T,u}| \frac{1}{\sqrt{T}} \eta^0
$$

In other words,

$$
\sum_{t=1}^{T} \frac{1}{\sqrt{t}} \sum_{k=1}^{t} \eta^{t-k} |g_{k,u}| = \sum_{t=1}^{T} |g_{t,u}| \sum_{k=t}^{T} \frac{1}{\sqrt{k}} \eta^{k-t}
$$

Moreover, since

$$
\sum_{k=t}^{T} \frac{1}{\sqrt{k}} \eta^{k-t} \leq \sum_{k=t}^{T} \frac{1}{\sqrt{t}} \eta^{k-t} = \frac{1}{\sqrt{t}} \sum_{k=t}^{T} \eta^{k-t} = \frac{1}{\sqrt{t}} \sum_{k=0}^{T-t} \eta^k \leq \frac{1}{\sqrt{t}} \left( \frac{1}{1-\eta} \right)
$$

where the last inequality is by Lemma 5, we obtain

$$\sum_{t=1}^{T} \frac{1}{\sqrt{t}} \sum_{k=1}^{t} \eta^{t-k} |g_{k,u}| \leq \sum_{t=1}^{T} |g_{t,u}| \frac{1}{\sqrt{t}} \left( \frac{1}{1-\eta} \right) = \frac{1}{1-\eta} \sum_{t=1}^{T} \frac{1}{\sqrt{t}} |g_{t,u}|$$

Furthermore, since

$$\sum_{t=1}^{T} \frac{1}{\sqrt{t}} |g_{t,u}| = \sqrt{\left( \sum_{t=1}^{T} \frac{1}{\sqrt{t}} |g_{t,u}| \right)^2} \leq \sqrt{\sum_{t=1}^{T} \frac{1}{t}} \sqrt{\sum_{t=1}^{T} g_{t,u}^2} \leq (\sqrt{1+\ln T}) \|g_{1:T,u}\|$$

where the first inequality is by Lemma 4 and the last inequality is by Lemma 6, we obtain

$$\sum_{t=1}^{T} \frac{1}{\sqrt{t}} \sum_{k=1}^{t} \eta^{t-k} |g_{k,u}| \leq \frac{\sqrt{1+\ln T}}{1-\eta} \|g_{1:T,u}\|$$

Hence, by Eq.(10),

$$\sum_{t=1}^{T} \frac{m_{t,u}^2}{\sqrt{t \hat{v}_{t,u}}} \leq \frac{\sqrt{1+\ln T}}{(1-\beta_1)\sqrt{1-\beta_2}(1-\eta)} \|g_{1:T,u}\|$$

which ends the proof. $\qquad\square$

## B.2    PROOF FOR LEMMA 1

**Lemma 1.**    *Suppose Assumptions 1-2 hold.* `AdamCB` *(Algorithm 1) with a mini-batch of size $K$, which is formed dynamically by distribution $p_t$, achieves the following upper-bound for the cumulative online regret $\mathcal{R}_{online}^{\pi}(T)$ over $T$ iterations,*

$$\mathcal{R}_{online}^{\pi}(T) \leq \rho_1 d \sqrt{T} + \sqrt{d} \rho_2 \sqrt{\frac{1}{n^2 K} \sum_{t=1}^{T} \mathbb{E}_{p_t} \left[ \sum_{j \in J_t} \frac{\|g_{j,t}\|^2}{(p_{j,t})^2} \right]} + \rho_3$$

*where $\rho_1$, $\rho_2$, and $\rho_3$ are defined as follows:*

$$\rho_1 = \frac{D^2 L}{2\alpha\gamma(1-\beta_1)^2}, \quad \rho_2 = \frac{\alpha\sqrt{1+\ln T}}{(1-\beta_1)^2\sqrt{1-\beta_2}(1-\eta)}, \quad \rho_3 = \frac{d\beta_1 D^2 L}{2\alpha\gamma(1-\beta_1)^2(1-\lambda)^2}$$

*Note that $d$ is the dimension of parameter space and the inputs of Algorithm 1 follows these conditions: (a) $\alpha_t = \frac{\alpha}{\sqrt{t}}$, (b) $\beta_1$, $\beta_2 \in [0, 1)$, $\beta_{1,t} := \beta_1 \lambda^{t-1}$ for all $t \in [T]$, $\lambda \in (0, 1)$, (c) $\eta = \beta_1/\sqrt{\beta_2} \leq 1$, and (d) $\gamma \in [0, 1)$.*

*Proof.*    Recall Lemma 3.
Since $f_t : \mathbb{R}^d \to \mathbb{R}$ is convex, we have, $f_t(\theta^*) - f_t(\theta_t) \geq g_t^{\mathrm{T}}(\theta^* - \theta_t)$. This means that

$$f_t(\theta_t) - f_t(\theta^*) \leq g_t^{\mathrm{T}}(\theta_t - \theta^*) = \sum_{u=1}^{d} g_{t,u}(\theta_{t,u} - \theta_{,u}^*)$$

From the parameter update rule presented in Algorithm 1,

$$\theta_{t+1} = \theta_t - \alpha_t m_t / \sqrt{\hat{v}_t}$$
$$= \theta_t - \alpha_t \left( \frac{\beta_{1,t}}{\sqrt{\hat{v}_t}} m_{t-1} + \frac{(1-\beta_{1,t})}{\sqrt{\hat{v}_t}} g_t \right)$$

We focus on the $u$-th dimension of the parameter vector $\theta_t \in \mathbb{R}^d$. Substract the scalar $\theta_{,u}^*$ and square both sides of the above update rule, we have,

$$(\theta_{t+1,u} - \theta_{,u}^*)^2 = (\theta_{t,u} - \theta_{,u}^*)^2 - 2\alpha_t \left( \frac{\beta_{1,t}}{\sqrt{\hat{v}_{t,u}}} m_{t-1,u} + \frac{(1-\beta_{1,t})}{\sqrt{\hat{v}_{t,u}}} g_{t,u} \right) (\theta_{t,u} - \theta_{,u}^*) + \alpha_t^2 \left( \frac{m_{t,u}}{\sqrt{\hat{v}_{t,u}}} \right)^2$$

We can rearrange the above equation

$$g_{t,u}(\theta_{t,u} - \theta_{\cdot,u}^*) = \frac{\sqrt{\hat{v}_{t,u}}}{2\alpha_t(1 - \beta_{1,t})}\left((\theta_{t,u} - \theta_{\cdot,u}^*)^2 - (\theta_{t+1,u} - \theta_{\cdot,u}^*)^2\right)$$
$$+ \frac{\alpha_t}{2(1 - \beta_{1,t})}\frac{m_{t,u}^2}{\sqrt{\hat{v}_{t,u}}} - \frac{\beta_{1,t}}{(1 - \beta_{1,t})}m_{t-1,u}(\theta_{t,u} - \theta_{\cdot,u}^*) \qquad (11)$$

Note that,

$$\mathcal{R}_{\text{online}}^{\pi}(T) = \mathbb{E}\left[\sum_{t=1}^{T} f_t(\theta_t) - \min_{\theta \in \mathbb{R}^d}\sum_{t=1}^{T} f_t(\theta)\right] = \mathbb{E}\left[\sum_{t=1}^{T}[f_t(\theta_t) - f_t(\theta^*)]\right]$$

where $\theta^* \in \arg\min_{\theta \in \mathbb{R}^d}\sum_{t=1}^{T} f_t(\theta)$ is defined as the optimal parameter that minimizes the cumulative loss over given $T$ iterations. Hence,

$$\mathcal{R}_{\text{online}}^{\pi}(T) = \mathbb{E}\left[\sum_{t=1}^{T}[f_t(\theta_t) - f_t(\theta^*)]\right] \le \mathbb{E}\left[\sum_{t=1}^{T} g_t^{\mathrm{T}}(\theta_t - \theta^*)\right] = \mathbb{E}\left[\sum_{t=1}^{T}\sum_{u=1}^{d} g_{t,u}(\theta_{t,u} - \theta_{\cdot,u}^*)\right] \qquad (12)$$

Combining Eq.(11) with Eq.(12), we obtain

$$\mathcal{R}_{\text{online}}^{\pi}(T) \le \mathbb{E}\left[\sum_{u=1}^{d}\sum_{t=1}^{T}\frac{\sqrt{\hat{v}_{t,u}}}{2\alpha_t(1 - \beta_{1,t})}\left((\theta_{t,u} - \theta_{\cdot,u}^*)^2 - (\theta_{t+1,u} - \theta_{\cdot,u}^*)^2\right)\right]$$
$$+ \mathbb{E}\left[\sum_{u=1}^{d}\sum_{t=1}^{T}\frac{\alpha_t}{2(1 - \beta_{1,t})}\frac{m_{t,u}^2}{\sqrt{\hat{v}_{t,u}}}\right] + \mathbb{E}\left[\sum_{u=1}^{d}\sum_{t=2}^{T}\frac{\beta_{1,t}}{(1 - \beta_{1,t})}m_{t-1,u}(\theta_{\cdot,u}^* - \theta_{t,u})\right]$$

On the other hand, for all $t \ge 2$, we have

$$m_{t-1,u}(\theta_{\cdot,u}^* - \theta_{t,u}) = \frac{(\hat{v}_{t-1,u})^{1/4}}{\sqrt{\alpha_{t-1}}}(\theta_{\cdot,u}^* - \theta_{t,u})\sqrt{\alpha_{t-1}}\frac{m_{t-1,u}}{(\hat{v}_{t-1,u})^{1/4}}$$
$$\le \frac{\sqrt{\hat{v}_{t-1,u}}}{2\alpha_{t-1}}(\theta_{\cdot,u}^* - \theta_{t,u})^2 + \alpha_{t-1}\frac{m_{t-1,u}^2}{2\sqrt{\hat{v}_{t-1,u}}}$$

where the inequality is from the fact that $pq \le p^2/2 + q^2/2$ for any $p, q \in \mathbb{R}$. Hence,

$$\mathcal{R}_{\text{online}}^{\pi}(T) \le \mathbb{E}\left[\sum_{u=1}^{d}\sum_{t=1}^{T}\frac{\sqrt{\hat{v}_{t,u}}}{2\alpha_t(1 - \beta_{1,t})}\left((\theta_{t,u} - \theta_{\cdot,u}^*)^2 - (\theta_{t+1,u} - \theta_{\cdot,u}^*)^2\right)\right]$$
$$+ \mathbb{E}\left[\sum_{u=1}^{d}\sum_{t=1}^{T}\frac{\alpha_t}{2(1 - \beta_{1,t})}\frac{m_{t,u}^2}{\sqrt{\hat{v}_{t,u}}}\right]$$
$$+ \mathbb{E}\left[\sum_{u=1}^{d}\sum_{t=2}^{T}\frac{\beta_{1,t}\alpha_{t-1}}{2(1 - \beta_{1,t})}\frac{m_{t-1,u}^2}{\sqrt{\hat{v}_{t-1,u}}}\right]$$
$$+ \mathbb{E}\left[\sum_{u=1}^{d}\sum_{t=2}^{T}\frac{\beta_{1,t}\sqrt{\hat{v}_{t-1,u}}}{2\alpha_{t-1}(1 - \beta_{1,t})}(\theta_{\cdot,u}^* - \theta_{t,u})^2\right]$$

Since $\beta_{1,t} \le \beta_1(1 \le t \le T)$, we obtain

$$\sum_{u=1}^{d}\sum_{t=2}^{T}\frac{\beta_{1,t}\sqrt{\hat{v}_{t-1,u}}}{2\alpha_{t-1}(1 - \beta_{1,t})}(\theta_{\cdot,u}^* - \theta_{t,u})^2 \le \sum_{u=1}^{d}\sum_{t=2}^{T}\frac{\beta_{1,t}\sqrt{\hat{v}_{t-1,u}}}{2\alpha_{t-1}(1 - \beta_1)}(\theta_{\cdot,u}^* - \theta_{t,u})^2$$

Moreover, we have

$$\sum_{u=1}^{d}\sum_{t=2}^{T}\frac{\beta_{1,t}\alpha_{t-1}}{2(1 - \beta_{1,t})}\frac{m_{t-1,u}^2}{\sqrt{\hat{v}_{t-1,u}}} = \sum_{u=1}^{d}\sum_{t=1}^{T-1}\frac{\beta_{1,t+1}\alpha_t}{2(1 - \beta_{1,t+1})}\frac{m_{t,u}^2}{\sqrt{\hat{v}_{t,u}}}$$
$$\le \sum_{u=1}^{d}\sum_{t=1}^{T}\frac{\alpha_t}{2(1 - \beta_{1,t+1})}\frac{m_{t,u}^2}{\sqrt{\hat{v}_{t,u}}}$$
$$\le \sum_{u=1}^{d}\sum_{t=1}^{T}\frac{\alpha_t}{2(1 - \beta_1)}\frac{m_{t,u}^2}{\sqrt{\hat{v}_{t,u}}}$$

where the last inequality is from the assumption that $\beta_{1,t} \leq \beta_1 < 1(1 \leq t \leq T)$. Therefore,

$$\sum_{u=1}^{d}\sum_{t=1}^{T}\frac{\alpha_t}{2(1-\beta_{1,t})}\frac{m_{t,u}^2}{\sqrt{\hat{v}_{t,u}}} + \sum_{u=1}^{d}\sum_{t=2}^{T}\frac{\beta_{1,t}\alpha_{t-1}}{2(1-\beta_{1,t})}\frac{m_{t-1,u}^2}{\sqrt{\hat{v}_{t-1,u}}} \leq \sum_{u=1}^{d}\sum_{t=1}^{T}\frac{\alpha_t}{1-\beta_1}\frac{m_{t,u}^2}{\sqrt{\hat{v}_{t,u}}}$$

and we obtain the bound for $\mathcal{R}_{\text{online}}^{\pi}(T)$ as:

$$\mathcal{R}_{\text{online}}^{\pi}(T) \leq \mathbb{E}\left[\sum_{u=1}^{d}\sum_{t=1}^{T}\frac{\sqrt{\hat{v}_{t,u}}}{2\alpha_t(1-\beta_{1,t})}\left((\theta_{t,u}-\theta_{,u}^*)^2-(\theta_{t+1,u}-\theta_{,u}^*)^2\right)\right] \quad (13)$$

$$+\mathbb{E}\left[\sum_{u=1}^{d}\sum_{t=1}^{T}\frac{\alpha_t}{1-\beta_1}\frac{m_{t,u}^2}{\sqrt{\hat{v}_{t,u}}}\right] \quad (14)$$

$$+\mathbb{E}\left[\sum_{u=1}^{d}\sum_{t=2}^{T}\frac{\beta_{1,t}\sqrt{\hat{v}_{t-1,u}}}{2\alpha_{t-1}(1-\beta_1)}(\theta_{,u}^*-\theta_{t,u})^2\right] \quad (15)$$

Now, we start to bound each term: (13), (14), and (15).

**Bound for the term (13).** Let us rewrite the term (13) as

$$\mathbb{E}\left[\sum_{u=1}^{d}\sum_{t=1}^{T}\frac{\sqrt{\hat{v}_{t,u}}}{2\alpha_t(1-\beta_{1,t})}\left((\theta_{t,u}-\theta_{,u}^*)^2-(\theta_{t+1,u}-\theta_{,u}^*)^2\right)\right]$$

$$=\mathbb{E}\left[\sum_{u=1}^{d}\frac{\sqrt{\hat{v}_{1,u}}}{2\alpha_1(1-\beta_{1,1})}(\theta_{1,u}-\theta_{,u}^*)^2\right] + \mathbb{E}\left[\sum_{u=1}^{d}\sum_{t=2}^{T}\frac{\sqrt{\hat{v}_{t,u}}}{2\alpha_t(1-\beta_{1,t})}(\theta_{t,u}-\theta_{,u}^*)^2\right]$$

$$-\mathbb{E}\left[\sum_{u=1}^{d}\sum_{t=2}^{T}\frac{\sqrt{\hat{v}_{t-1,u}}}{2\alpha_{t-1}(1-\beta_{1,t-1})}(\theta_{t,u}-\theta_{,u}^*)^2\right] - \mathbb{E}\left[\sum_{u=1}^{d}\frac{\sqrt{\hat{v}_{T,u}}}{2\alpha_T(1-\beta_{1,T})}(\theta_{T,u}-\theta_{,u}^*)^2\right]$$

Omitting the last term and replacing $\alpha_t = \alpha/\sqrt{t}(1 \leq t \leq T)$, we obtain

$$\mathbb{E}\left[\sum_{u=1}^{d}\sum_{t=1}^{T}\frac{\sqrt{\hat{v}_{t,u}}}{2\alpha_t(1-\beta_{1,t})}\left((\theta_{t,u}-\theta_{,u}^*)^2-(\theta_{t+1,u}-\theta_{,u}^*)^2\right)\right]$$

$$\leq \mathbb{E}\left[\sum_{u=1}^{d}\frac{\sqrt{\hat{v}_{1,u}}}{2\alpha(1-\beta_{1,1})}(\theta_{1,u}-\theta_{,u}^*)^2\right]$$

$$+\frac{1}{2\alpha}\mathbb{E}\left[\sum_{u=1}^{d}\sum_{t=2}^{T}(\theta_{t,u}-\theta_{,u}^*)^2\left(\frac{\sqrt{t\hat{v}_{t,u}}}{(1-\beta_{1,t})}-\frac{\sqrt{(t-1)\hat{v}_{t-1,u}}}{(1-\beta_{1,t-1})}\right)\right]$$

Recall that by the update rule on $\hat{v}_t$, we have $\hat{v}_{t,u} \leftarrow \max\left\{\frac{(1-\beta_{1,t})^2}{(1-\beta_{1,t-1})^2}\hat{v}_{t-1,u}, v_{t,u}\right\}$. Therefore, $\hat{v}_{t,u} \geq \frac{(1-\beta_{1,t})^2}{(1-\beta_{1,t-1})^2}\hat{v}_{t-1,u}$, and hence

$$\frac{\sqrt{t\hat{v}_{t,u}}}{(1-\beta_{1,t})}-\frac{\sqrt{(t-1)\hat{v}_{t-1,u}}}{(1-\beta_{1,t-1})} \geq \frac{\sqrt{t\frac{(1-\beta_{1,t})^2}{(1-\beta_{1,t-1})^2}\hat{v}_{t-1,u}}}{(1-\beta_{1,t})}-\frac{\sqrt{(t-1)\hat{v}_{t-1,u}}}{(1-\beta_{1,t-1})}$$

$$=\frac{\sqrt{t\hat{v}_{t-1,u}}}{(1-\beta_{1,t-1})}-\frac{\sqrt{(t-1)\hat{v}_{t-1,u}}}{(1-\beta_{1,t-1})}$$

$$> 0$$

Now by the positivity of the essential formula $\frac{\sqrt{t\hat{v}_{t,u}}}{(1-\beta_{1,t})} - \frac{\sqrt{(t-1)\hat{v}_{t-1,u}}}{(1-\beta_{1,t-1})}$, we obtain

$$
\mathbb{E}\left[\sum_{u=1}^{d}\sum_{t=1}^{T}\frac{\sqrt{\hat{v}_{t,u}}}{2\alpha_t(1-\beta_{1,t})}\left((\theta_{t,u}-\theta_{\cdot,u}^*)^2 - (\theta_{t+1,u}-\theta_{\cdot,u}^*)^2\right)\right]
$$

$$
\leq \frac{D^2}{2\alpha}\sum_{u=1}^{d}\frac{\sqrt{\hat{v}_{1,u}}}{(1-\beta_1)} + \frac{D^2}{2\alpha}\mathbb{E}\left[\sum_{u=1}^{d}\sum_{t=2}^{T}\left(\frac{\sqrt{t\hat{v}_{t,u}}}{(1-\beta_{1,t})} - \frac{\sqrt{(t-1)\hat{v}_{t-1,u}}}{(1-\beta_{1,t-1})}\right)\right]
$$

$$
\leq \frac{D^2}{2\alpha}\sum_{u=1}^{d}\frac{\sqrt{T\hat{v}_{T,u}}}{(1-\beta_{1,T})} \leq \frac{dD^2L}{2\alpha\gamma(1-\beta_1)^2}\sqrt{T}
$$

where the last inequality is by Lemma 9.

**Bound for the term (14).**

$$
\mathbb{E}\left[\sum_{u=1}^{d}\sum_{t=1}^{T}\frac{\alpha_t}{1-\beta_1}\frac{m_{t,u}^2}{\sqrt{\hat{v}_{t,u}}}\right] = \frac{\alpha}{1-\beta_1}\mathbb{E}\left[\sum_{u=1}^{d}\sum_{t=1}^{T}\frac{m_{t,u}^2}{\sqrt{t\hat{v}_{t,u}}}\right]
$$

$$
\leq \frac{\alpha}{1-\beta_1}\mathbb{E}\left[\sum_{u=1}^{d}\frac{\sqrt{\ln T + 1}}{(1-\beta_1)\sqrt{1-\beta_2}(1-\eta)}\|g_{1:T,u}\|\right]
$$

$$
= \frac{\alpha\sqrt{\ln T + 1}}{(1-\beta_1)^2\sqrt{1-\beta_2}(1-\eta)}\sum_{u=1}^{d}\mathbb{E}\left[\|g_{1:T,u}\|\right]
$$

where the last inequality is by Lemma 10.

**Bound for the term (15).** By Assumption 2 that $\|\theta_m - \theta_n\| \leq D$ for any $m, n \in [T]$, $\alpha_t = \alpha/\sqrt{t}$, and $\beta_{1,t} = \beta_1\lambda^{t-1} \leq \beta_1 \leq 1$, we obtain

$$
\mathbb{E}\left[\sum_{u=1}^{d}\sum_{t=2}^{T}\frac{\beta_{1,t}\sqrt{\hat{v}_{t-1,u}}}{2\alpha_{t-1}(1-\beta_1)}(\theta_{\cdot,u}^* - \theta_{t,u})^2\right] \leq \frac{D^2}{2\alpha(1-\beta_1)}\mathbb{E}\left[\sum_{u=1}^{d}\sum_{t=2}^{T}\beta_{1,t}\sqrt{(t-1)\hat{v}_{t-1,u}}\right]
$$

Therefore, from Lemma 9, we obtain

$$
\mathbb{E}\left[\sum_{u=1}^{d}\sum_{t=2}^{T}\frac{\beta_{1,t}\sqrt{\hat{v}_{t-1,u}}}{2\alpha_{t-1}(1-\beta_1)}(\theta_{\cdot,u}^* - \theta_{t,u})^2\right] \leq \frac{dD^2L}{2\alpha\gamma(1-\beta_1)^2}\mathbb{E}\left[\sum_{t=2}^{T}\beta_{1,t}\sqrt{(t-1)}\right]
$$

Note that

$$
\sum_{t=2}^{T}\beta_{1,t}\sqrt{(t-1)} = \sum_{t=2}^{T}\beta_1\lambda^{t-1}\sqrt{(t-1)} \leq \sum_{t=2}^{T}\beta_1\sqrt{(t-1)}\lambda^{t-1} \leq \sum_{t=2}^{T}\beta_1 t\lambda^{t-1} \leq \frac{\beta_1}{(1-\lambda)^2}
$$

where the first inequality is from the fact that $\beta_1 \leq 1$, and the last inequality is from Lemma 5. Thus, the bound for the term (15) is

$$
\mathbb{E}\left[\sum_{u=1}^{d}\sum_{t=2}^{T}\frac{\beta_{1,t}\sqrt{\hat{v}_{t-1,u}}}{2\alpha_{t-1}(1-\beta_1)}(\theta_{\cdot,u}^* - \theta_{t,u})^2\right] \leq \frac{d\beta_1 D^2L}{2\alpha\gamma(1-\beta_1)^2(1-\lambda)^2}
$$

We bounded for terms (13), (14), and (15).

$$
\mathcal{R}_{\text{online}}^{\pi}(T) \leq \frac{dD^2L}{2\alpha\gamma(1-\beta_1)^2}\sqrt{T} + \frac{\alpha\sqrt{\ln T + 1}}{(1-\beta_1)^2\sqrt{1-\beta_2}(1-\eta)}\sum_{u=1}^{d}\mathbb{E}\left[\|g_{1:T,u}\|\right]
$$

$$
+ \frac{d\beta_1 D^2L}{2\alpha\gamma(1-\beta_1)^2(1-\lambda)^2}
$$

Hence,

$$
\mathcal{R}_{\text{online}}^{\pi}(T) \leq \rho_1 d\sqrt{T} + \rho_2\sum_{u=1}^{d}\mathbb{E}\left[\|g_{1:T,u}\|\right] + \rho_3 \tag{16}
$$

where $\rho_1, \rho_2$, and $\rho_3$ are defined as the following:

$$\rho_1 = \frac{D^2 L}{2\alpha\gamma(1-\beta_1)^2}, \rho_2 = \frac{\alpha\sqrt{1+\ln T}}{(1-\beta_1)^2\sqrt{1-\beta_2}(1-\eta)}, \rho_3 = \frac{d\beta_1 D^2 L}{2\alpha\gamma(1-\beta_1)^2(1-\lambda)^2}$$

Now, we consider $\sum_{u=1}^{d} \mathbb{E}\left[\|g_{1:T,u}\|\right]$, which is in the right-hand side of Eq.(16).

$$\sum_{u=1}^{d} \mathbb{E}\left[\|g_{1:T,u}\|\right] = d\sum_{u=1}^{d}\frac{1}{d}\mathbb{E}\left[\sqrt{\sum_{t=1}^{T}g_{t,u}^2}\right] \leq d\sqrt{\sum_{u=1}^{d}\frac{1}{d}\mathbb{E}\left[\sum_{t=1}^{T}g_{t,u}^2\right]} = \sqrt{d}\sqrt{\sum_{t=1}^{T}\mathbb{E}\left[\|g_t\|^2\right]}$$

where the first inequality is due to the concavity of square root. Recall that the unbiased gradient estimate is $g_t = \frac{1}{K}\sum_{j\in J_t}\frac{g_{j,t}}{np_{j,t}}$. Hence,

$$\mathcal{R}_{\text{online}}^{\pi}(T) \leq \rho_1 d\sqrt{T} + \rho_2 \sum_{u=1}^{d}\mathbb{E}_{p_t}\left[\|g_{1:T,u}\|\right] + \rho_3$$

$$\leq \rho_1 d\sqrt{T} + \rho_2\sqrt{d}\sqrt{\sum_{t=1}^{T}\mathbb{E}_{p_t}\left[\|g_t\|^2\right]} + \rho_3$$

$$\leq \rho_1 d\sqrt{T} + \rho_2\sqrt{d}\sqrt{\sum_{t=1}^{T}\mathbb{E}_{p_t}\left[\left\|\frac{1}{K}\sum_{j\in J_t}\frac{g_{j,t}}{np_{j,t}}\right\|^2\right]} + \rho_3$$

The last inequality uses Jensen's inequality to the convex function $\|\cdot\|^2$. Therefore,

$$\mathcal{R}_{\text{online}}^{\pi}(T) \leq \rho_1 d\sqrt{T} + \rho_2\sqrt{d}\sqrt{\frac{1}{n^2 K^2}\sum_{t=1}^{T}\mathbb{E}_{p_t}\left[\left\|\sum_{j\in J_t}\frac{g_{j,t}}{p_{j,t}}\right\|^2\right]} + \rho_3$$

$$\leq \rho_1 d\sqrt{T} + \rho_2\sqrt{d}\sqrt{\frac{1}{n^2 K}\sum_{t=1}^{T}\mathbb{E}_{p_t}\left[\sum_{j\in J_t}\frac{\|g_{j,t}\|^2}{(p_{j,t})^2}\right]} + \rho_3$$

where the last inequality is by Lemma 4. This completes the proof of Lemma 1. $\qquad\square$

### B.3 PROOF FOR LEMMA 2

**Lemma 2.** *Suppose Assumptions 1-2 hold. If we set $\gamma = \min\left\{1, \sqrt{\frac{n\ln(n/K)}{(e-1)TK}}\right\}$, the batch selection (Algorithm 2) and the weight update rule (Algorithm 3) following* `AdamCB` *(Algorithm 1) implies*

$$\sum_{t=1}^{T}\mathbb{E}_{p_t}\left[\sum_{j\in J_t}\frac{\|g_{j,t}\|^2}{(p_{j,t})^2}\right] - \min_{p_t}\sum_{t=1}^{T}\mathbb{E}_{p_t}\left[\sum_{j\in J_t}\frac{\|g_{j,t}\|^2}{(p_{j,t})^2}\right] = \mathcal{O}\left(\sqrt{KnT\ln\frac{n}{K}}\right)$$

*Proof.* We set $\ell_{j,t} = \frac{p_{min}^2}{L^2}\left(-\frac{\|g_{j,t}\|^2}{(p_{j,t})^2} + \frac{L^2}{p_{min}^2}\right)$ in Algorithm 3. Since $\|g_{i,t}\| \leq L$ and $p_{i,t} \geq p_{min}$ for all $i \in [n]$ and $t \in [T]$ by Assumption 1, we have $\ell_{i,t} \in [0,1]$.
Let $W_t := \sum_{i=1}^{n} w_t$. Then, for any $t \in [T]$,

$$\frac{W_t}{W_{t-1}} = \sum_{i\in[n]\setminus S_{\text{null},t}}\frac{w_{i,t}}{W_{t-1}} + \sum_{i\in S_{\text{null},t}}\frac{w_{i,t}}{W_{t-1}}$$

$$= \sum_{i\in[n]\setminus S_{\text{null},t}}\frac{w_{i,t-1}}{W_{t-1}}\exp\left(-\frac{K\gamma}{n}\hat{\ell}_{i,t}\right) + \sum_{i\in S_{\text{null},t}}\frac{w_{i,t-1}}{W_{t-1}}$$

The last equality is by the weight update rule in Algorithm 3. From the probability computation in Algorithm 2, we have

$$p_{i,t} = K\left((1-\gamma)\frac{w_{i,t-1}}{\sum_{j=1}^n w_{j,t-1}} + \frac{\gamma}{n}\right) \geq \frac{K\gamma}{n}$$

Thus, we obtain the following bound,

$$0 \leq \frac{K\gamma}{n}\hat{\ell}_{i,t} = \frac{K\gamma\ell_{i,t}}{np_{i,t}} \leq \ell_{i,t} \leq 1$$

By the fact that $e^{-x} \leq 1 - x + (e-2)x^2$ for all $x \in [0,1]$, and considering $\frac{K\gamma}{n}\hat{\ell}_{i,t}$ as $x$, we have

$$\frac{W_t}{W_{t-1}} \leq \sum_{i \in [n]\setminus S_{\text{null},t}} \frac{w_{i,t-1}}{W_{t-1}}\left[1 - \frac{K\gamma}{n}\hat{\ell}_{i,t} + (e-2)\left(\frac{K\gamma}{n}\hat{\ell}_{i,t}\right)^2\right] + \sum_{i \in S_{\text{null},t}} \frac{w_{i,t-1}}{W_{t-1}}$$

$$= 1 + \sum_{i \in [n]\setminus S_{\text{null},t}} \frac{w_{i,t-1}}{W_{t-1}}\left[-\frac{K\gamma}{n}\hat{\ell}_{i,t} + (e-2)\left(\frac{K\gamma}{n}\hat{\ell}_{i,t}\right)^2\right]$$

$$= 1 + \sum_{i \in [n]\setminus S_{\text{null},t}} \frac{\frac{p_{i,t}}{K} - \frac{\gamma}{n}}{1-\gamma}\left[-\frac{K\gamma}{n}\hat{\ell}_{i,t} + (e-2)\left(\frac{K\gamma}{n}\hat{\ell}_{i,t}\right)^2\right]$$

$$\leq 1 - \frac{\gamma}{n(1-\gamma)}\sum_{i \in [n]\setminus S_{\text{null},t}} p_{i,t}\hat{\ell}_{i,t} + \frac{K(e-2)\gamma^2}{n^2(1-\gamma)}\sum_{i \in [n]\setminus S_{\text{null},t}} p_{i,t}(\hat{\ell}_{i,t})^2$$

$$\leq 1 - \frac{\gamma}{n(1-\gamma)}\sum_{i \in J_t\setminus S_{\text{null},t}} \ell_{i,t} + \frac{K(e-2)\gamma^2}{n^2(1-\gamma)}\sum_{i \in [n]} \hat{\ell}_{i,t}$$

The last inequality uses the fact that $p_{i,t}\hat{\ell}_{i,t} = \ell_{i,t} \leq 1$ for $i \in J_t$ and $p_{i,t}\hat{\ell}_{i,t} = 0$ for $i \notin J_t$. Taking logarithms and using the fact that $\ln(1+x) \leq x$ for all $x > -1$ gives

$$\ln\frac{W_t}{W_{t-1}} \leq -\frac{\gamma}{n(1-\gamma)}\sum_{i \in J_t\setminus S_{\text{null},t}} \ell_{i,t} + \frac{K(e-2)\gamma^2}{n^2(1-\gamma)}\sum_{i \in [n]} \hat{\ell}_{i,t}$$

By summing over $t$, we obtain

$$\ln\frac{W_T}{W_1} \leq -\frac{\gamma}{n(1-\gamma)}\sum_{t=1}^T\sum_{i \in J_t\setminus S_{\text{null},t}} \ell_{i,t} + \frac{K(e-2)\gamma^2}{n^2(1-\gamma)}\sum_{t=1}^T\sum_{i \in [n]} \hat{\ell}_{i,t}$$

On the other hand, for the sequence $\{J_t^*\}_{t=1}^T$ of batches with the optimal $\sum_{t=1}^T\sum_{j \in J_t} \ell_{j,t}$ among all subsets $J_t$ containing $K$ elements,

$$\ln\frac{W_T}{W_1} \geq \ln\frac{\sum_{j \in J_t^*} w_{j,T}}{W_1} \geq \frac{\sum_{j \in J_t^*} \ln w_{j,T}}{K} + \ln\frac{K}{n}$$

$$= -\frac{\gamma}{n}\sum_{j \in J_t^*}\sum_{t:j\notin S_{\text{null},t}} \hat{\ell}_{j,t} + \ln\frac{K}{n}$$

The first line above uses the fact that

$$\sum_{j \in J_t^*} w_{j,T} \geq K(\Pi_{j \in J_t^*} w_{j,T})^{1/K}$$

and the second line uses $w_{j,T} = \exp\left(-(K\gamma/n)\sum_{t:j\notin S_{\text{null},t}} \hat{\ell}_{j,t}\right)$.

From combining results,

$$\sum_{j \in J_t^*}\sum_{t:j\notin S_{\text{null},t}} \hat{\ell}_{j,t} + \frac{n}{\gamma}\ln\frac{K}{n} \leq \frac{1}{(1-\gamma)}\sum_{t=1}^T\sum_{i \in J_t\setminus S_{\text{null},t}} \ell_{i,t} + \frac{(e-2)K\gamma}{n(1-\gamma)}\sum_{t=1}^T\sum_{i \in [n]} \hat{\ell}_{i,t}$$

Since $\sum_{j \in J_t^*} \sum_{t:j \in S_{\text{null},t}} \ell_{j,t} \leq \frac{1}{1-\gamma} \sum_{t=1}^{T} \sum_{i \in S_{\text{null},t}} \ell_{i,t}$ trivially holds, we have

$$\sum_{j \in J_t^*} \sum_{t:j \notin S_{\text{null},t}} \hat{\ell}_{j,t} + \sum_{j \in J_t^*} \sum_{t:j \in S_{\text{null},t}} \ell_{j,t} + \frac{n}{\gamma} \ln \frac{K}{n} \leq \frac{1}{(1-\gamma)} \sum_{t=1}^{T} \sum_{i \in J_t} \ell_{i,t} + \frac{(e-2)K\gamma}{n(1-\gamma)} \sum_{t=1}^{T} \sum_{i \in [n]} \hat{\ell}_{i,t}$$

Let $L_{\text{MIN-K}}(T) := \sum_{t=1}^{T} \sum_{j \in J_t^*} \ell_{j,t}$ and $L_{\text{EXP3-K}}(T) := \sum_{t=1}^{T} \sum_{j \in J_t} \ell_{j,t}$. Taking the expectation of both sides and using the properties of $\hat{\ell}_{i,t}$, we obtain,

$$L_{\text{MIN-K}}(T) + \frac{n}{\gamma} \ln \frac{K}{n} \leq \frac{1}{(1-\gamma)} \mathbb{E}[L_{\text{EXP3-K}}(T)] + \frac{(e-2)K\gamma}{n(1-\gamma)} \sum_{t=1}^{T} \sum_{i \in [n]} \ell_{i,t}$$

This is because the expectation of $\hat{\ell}_{j,t}$ is $\ell_{j,t}$ from the fact that $\texttt{DepRound}$ selects $i$-th sample with probability $p_{i,t}$. Since $\sum_{t=1}^{T} \sum_{i=1}^{n} \ell_{i,t} \leq \frac{n L_{\text{MIN-K}}(T)}{K}$, we have the following statement,

$$L_{\text{MIN-K}}(T) - \mathbb{E}[L_{\text{EXP3-K}}(T)] \leq (e-1)\gamma L_{\text{MIN-K}}(T) + \frac{n}{\gamma} \ln \frac{n}{K}$$

Using the fact that $L_{\text{MIN-K}}(T) \leq TK$ and choosing the input parameter as $\gamma = \min\left\{1, \sqrt{\frac{n \ln (n/K)}{(e-1)TK}}\right\}$, we obtain the following,

$$L_{\text{MIN-K}}(T) - \mathbb{E}[L_{\text{EXP3-K}}(T)] \leq 2\sqrt{e-1}\sqrt{KnT \ln \frac{n}{K}} \leq 2.63\sqrt{KnT \ln \frac{n}{K}}$$

Therefore, considering the scaling factor, we have:

$$\sum_{t=1}^{T} \mathbb{E}_{p_t}\left[\sum_{j \in J_t} \frac{\|g_{j,t}\|^2}{(p_{j,t})^2}\right] - \min_{p_t} \sum_{t=1}^{T} \mathbb{E}_{p_t}\left[\sum_{j \in J_t} \frac{\|g_{j,t}\|^2}{(p_{j,t})^2}\right] = \frac{L^2}{p_{min}^2}\left(L_{\text{MIN-K}}(T) - \mathbb{E}[L_{\text{EXP3-K}}(T)]\right)$$

$$\leq \frac{2.63 L^2}{p_{min}^2} \sqrt{KnT \ln \frac{n}{K}}$$

$$= \mathcal{O}\left(\sqrt{KnT \ln \frac{n}{K}}\right)$$

This completes the proof of Lemma 2. $\qquad\qquad\qquad\qquad\qquad\qquad\qquad\qquad\qquad\qquad\quad\square$

### B.4 Proof for Theorem 1 (Regret Bound of AdamCB)

In this section, we present the full proof of Theorem 1. Recall that the online regret only focuses on the minimization over the sequence of mini-batch datasets $\{\mathcal{D}_t\}_{t=1}^T$. Thus, the online regret of the algorithm at the end of $T$ iterations is defined as

$$\mathcal{R}_{\text{online}}^\pi(T) := \mathbb{E}\left[\sum_{t=1}^T f(\theta_t; \mathcal{D}_t) - \min_{\theta \in \mathbb{R}^d} \sum_{t=1}^T f(\theta; \mathcal{D}_t)\right]$$

However, our ultimate goal is to find the optimal selection of the parameter under the full dataset. Consider an online optimization algorithm $\pi$ that computes the sequence of model parameters $\theta_1, \ldots, \theta_T$. Then, we can compare the performance of $\pi$ with the optimal selection of the parameter $\min_{\theta \in \mathbb{R}^d} f(\theta; \mathcal{D})$ under the full dataset. The cumulative regret after $T$ iterations is

$$\mathcal{R}^\pi(T) := \mathbb{E}\left[\sum_{t=1}^T f(\theta_t; \mathcal{D}) - T \cdot \min_{\theta \in \mathbb{R}^d} f(\theta; \mathcal{D})\right]$$

where the expectation is taken with respect to any stochasticity in data sampling and parameter estimation. Before we prove Theorem 1, we first prove the following lemma.

**Lemma 11.** *The cumulative regret $\mathcal{R}^\pi(T)$ can be decomposed into sub-parts which includes the cumulative online regret $\mathcal{R}_{online}^\pi(T)$ and additional terms that are sub-linear in $T$:*

$$\mathcal{R}^\pi(T) = \mathcal{R}_{online}^\pi(T) + \mathcal{O}(\sqrt{T})$$

*Proof.* First, rewrite $\mathcal{R}^\pi(T)$ by expanding the terms inside the expectations. We add and subtract the sum $\sum_{t=1}^T f(\theta_t; \mathcal{D}_t)$ inside the expectation:

$$\mathcal{R}^\pi(T) = \mathbb{E}\left[\sum_{t=1}^T f(\theta_t; \mathcal{D}) - T \cdot \min_{\theta \in \mathbb{R}^d} f(\theta; \mathcal{D})\right]$$

$$= \mathbb{E}\left[\sum_{t=1}^T f(\theta_t; \mathcal{D}) - \sum_{t=1}^T f(\theta_t; \mathcal{D}_t) + \sum_{t=1}^T f(\theta_t; \mathcal{D}_t) - T \cdot \min_{\theta \in \mathbb{R}^d} f(\theta; \mathcal{D})\right]$$

We also add and subtract the term $\min_{\theta \in \mathbb{R}^d} \sum_{t=1}^T f(\theta; \mathcal{D}_t)$ inside the expectation. Then, we have the following,

$$\mathcal{R}^\pi(T) = \mathbb{E}\left[\sum_{t=1}^T f(\theta_t; \mathcal{D}) - \sum_{t=1}^T f(\theta_t; \mathcal{D}_t) + \sum_{t=1}^T f(\theta_t; \mathcal{D}_t) - T \cdot \min_{\theta \in \mathbb{R}^d} f(\theta; \mathcal{D})\right]$$

$$= \mathbb{E}\left[\sum_{t=1}^T f(\theta_t; \mathcal{D}) - \sum_{t=1}^T f(\theta_t; \mathcal{D}_t)\right] + \mathbb{E}\left[\sum_{t=1}^T f(\theta_t; \mathcal{D}_t) - \min_{\theta \in \mathbb{R}^d} \sum_{t=1}^T f(\theta; \mathcal{D}_t)\right]$$

$$+ \mathbb{E}\left[\min_{\theta \in \mathbb{R}^d} \sum_{t=1}^T f(\theta; \mathcal{D}_t) - T \cdot \min_{\theta \in \mathbb{R}^d} f(\theta; \mathcal{D})\right]$$

Since the second term of the right-hand side in above equation is equal the online cumulative regret $\mathcal{R}_{\text{online}}^\pi(T)$, we can rewrite $\mathcal{R}^\pi(T)$ as:

$$\mathcal{R}^\pi(T) = \mathcal{R}_{\text{online}}^\pi(T)$$

$$+ \mathbb{E}\left[\sum_{t=1}^T f(\theta_t; \mathcal{D}) - \sum_{t=1}^T f(\theta_t; \mathcal{D}_t)\right] \tag{17}$$

$$+ \mathbb{E}\left[\min_{\theta \in \mathbb{R}^d} \sum_{t=1}^T f(\theta; \mathcal{D}_t) - T \cdot \min_{\theta \in \mathbb{R}^d} f(\theta; \mathcal{D})\right] \tag{18}$$

Now, let us consider each term in detail.

**Bound for the term (17).** Recall the expression of $f(\theta; \mathcal{D})$ and $f_t := f(\theta; \mathcal{D}_t)$:

$$f(\theta; \mathcal{D}) = \frac{1}{n} \sum_{i=1}^{n} \ell(\theta; x_i, y_i), \quad f(\theta; \mathcal{D}_t) = \frac{1}{K} \sum_{j \in J_t} \frac{\ell(\theta; x_j, y_j)}{n p_{j,t}}$$

where $J_t$ is the set of indices in the subset dataset (mini-batch) at iteration $t$, $\mathcal{D}_t \subseteq \mathcal{D}$. For any $\theta \in \mathbb{R}^d$, we have

$$\mathbb{E}[f(\theta; \mathcal{D}_t)] = \mathbb{E}\left[ \frac{1}{K} \sum_{j \in J_t} \frac{\ell(\theta; x_j, y_j)}{n p_{j,t}} \right] = \frac{1}{K} \sum_{j \in J_t} \mathbb{E}\left[ \frac{\ell(\theta; x_j, y_j)}{n p_{j,t}} \right]$$

$$= \frac{1}{K} \sum_{j \in J_t} \sum_{i=1}^{n} \frac{\ell(\theta; x_i, y_i)}{n p_{i,t}} p_{i,t} = \frac{1}{n} \sum_{i=1}^{n} \ell(\theta; x_i, y_i) = f(\theta; \mathcal{D}).$$

Note that, by linearity of expectation, we can interchange the expectation and the summation. Since $\mathbb{E}[f(\theta; \mathcal{D}_t)] = f(\theta; \mathcal{D})$, we have for the term (17) as:

$$(17) = \mathbb{E}\left[ \sum_{t=1}^{T} f(\theta_t; \mathcal{D}) - \sum_{t=1}^{T} f(\theta_t; \mathcal{D}_t) \right]$$

$$= \mathbb{E}\left[ \sum_{t=1}^{T} [f(\theta_t; \mathcal{D}) - f(\theta_t; \mathcal{D}_t)] \right]$$

$$= \sum_{t=1}^{T} \mathbb{E}[f(\theta_t; \mathcal{D}) - f(\theta_t; \mathcal{D}_t)] = 0$$

**Bound for the term (18).** Let $\theta^*$ be the parameter that minimizes the cumulative loss over the full dataset $\mathcal{D}$, i.e, $\theta^* \in \arg\min_{\theta \in \mathbb{R}^d} f(\theta; \mathcal{D})$. Since $\theta^*$ is optimal for the full dataset, we have:

$$\min_{\theta \in \mathbb{R}^d} f(\theta; \mathcal{D}) = f(\theta^*; \mathcal{D})$$

Similarly, denote the optimal parameter for the cumulative regret for mini-batch datasets by $\theta_t^* := \arg\min_{\theta \in \mathbb{R}^d} \sum_{t=1}^{T} f(\theta; \mathcal{D}_t)$. Given these notations, we can write the term (18) as:

$$(18) = \mathbb{E}\left[ \min_{\theta \in \mathbb{R}^d} \sum_{t=1}^{T} f(\theta; \mathcal{D}_t) - T \cdot \min_{\theta \in \mathbb{R}^d} f(\theta; \mathcal{D}) \right] = \mathbb{E}\left[ \sum_{t=1}^{T} f(\theta_t^*; \mathcal{D}_t) - T \cdot f(\theta^*; \mathcal{D}) \right]$$

We can add and subtract the term $\sum_{t=1}^{T} f(\theta^*; \mathcal{D}_t)$ inside the expectation.

$$\mathbb{E}\left[ \sum_{t=1}^{T} f(\theta_t^*; \mathcal{D}_t) - T \cdot f(\theta^*; \mathcal{D}) \right] = \mathbb{E}\left[ \sum_{t=1}^{T} f(\theta_t^*; \mathcal{D}_t) - \sum_{t=1}^{T} f(\theta^*; \mathcal{D}_t) \right]$$

$$+ \mathbb{E}\left[ \sum_{t=1}^{T} f(\theta^*; \mathcal{D}_t) - T \cdot f(\theta^*; \mathcal{D}) \right]$$

Note that $\mathbb{E}[f(\theta^*; \mathcal{D}_t)] = f(\theta^*; \mathcal{D})$ holds as we have shown when bounding the term (17). By the linearity of expectation, we have

$$\mathbb{E}\left[ \sum_{t=1}^{T} f(\theta^*; \mathcal{D}_t) \right] = \sum_{t=1}^{T} \mathbb{E}[f(\theta^*; \mathcal{D}_t)] = T \cdot f(\theta^*; \mathcal{D})$$

Since $\mathbb{E}\left[ \sum_{t=1}^{T} f(\theta^*; \mathcal{D}_t) - T \cdot f(\theta^*; \mathcal{D}) \right] = 0$ holds, the term (18) reduces to

$$(18) = \mathbb{E}\left[ \sum_{t=1}^{T} (f(\theta_t^*; \mathcal{D}_t) - f(\theta^*; \mathcal{D}_t)) \right]$$

$$= \mathbb{E}\left[ \sum_{t=1}^{T} (f_t(\theta_t^*) - f_t(\theta^*)) \right]$$

By the convexity of $f_t$, we have:

$$f_t(\theta_t^*) - f_t(\theta^*) \le g_t^T(\theta_t^* - \theta^*)$$

Therefore,

$$\mathbb{E}\left[\sum_{t=1}^{T}(f_t(\theta_t^*) - f_t(\theta^*))\right] \le \mathbb{E}\left[\sum_{t=1}^{T} g_t^T(\theta_t^* - \theta^*)\right]$$

Using bounded gradients assumption (Assumption 1), i.e, $\|g_t\| \le L/\gamma$ (Proof in Lemma 9), and Cauchy-Schwarz inequality (Lemma 4), we have

$$(18) \le \mathbb{E}\left[\sum_{t=1}^{T} g_t^T(\theta_t^* - \theta^*)\right] \le \sum_{t=1}^{T}\mathbb{E}[\|g_t\|\|\theta_t^* - \theta^*\|] \le (L/\gamma)\sum_{t=1}^{T}\mathbb{E}[\|\theta_t^* - \theta^*\|]$$

Recall the parameter update rule, $\theta_{t+1} \leftarrow \theta_t - \alpha_t m_t/(\sqrt{\hat{v}_t} + \epsilon)$. Then

$$\|\theta_{t+1}^* - \theta^*\| \le \|\theta_t^* - \theta^*\| + \alpha_t\left\|m_t/(\sqrt{\hat{v}_t} + \epsilon)\right\| \tag{19}$$

Now, we claim that $\|m_t\|$ is bounded. The update rule for the first moment estimate:

$$m_t \leftarrow \beta_{1,t}m_{t-1} + (1 - \beta_{1,t})g_t$$

Then, the expression for $m_t$ is:

$$m_t = \sum_{k=1}^{t}(1 - \beta_{1,k})\left(\prod_{r=k+1}^{t}\beta_{1,r}\right)g_k$$

where $\beta_{1,t} = \beta_1\lambda^{t-1}$ with $\beta_1 < 1$ and $\lambda < 1$. Note that $\|g_k\|$ is bounded by $L/\gamma$ for all $k$. This implies that:

$$\|m_t\| \le \sum_{k=1}^{t}|1 - \beta_{1,k}|\left|\prod_{r=k+1}^{t}\beta_{1,r}\right|\|g_k\|$$

$$\le (L/\gamma)\sum_{k=1}^{t}|1 - \beta_1\lambda^{k-1}|\left|\prod_{r=k+1}^{t}\beta_1\lambda^{r-1}\right|$$

$$\le (L/\gamma)\sum_{k=1}^{t}\beta_1^{t-k}\lambda^{\frac{t(t-1)-k(k-1)}{2}}$$

$$\le (L/\gamma)\sum_{k=1}^{t}\beta_1^{t-k}$$

$$\le \frac{L}{\gamma(1 - \beta_1)}$$

The last inequality is due to Lemma 5. Therefore, the step size in Eq.(19) is bounded by:

$$\frac{\alpha_t\|m_t\|}{\sqrt{\hat{v}_t} + \epsilon} \le \frac{\alpha_t L}{\epsilon\gamma(1 - \beta_1)} = \frac{\alpha L}{\sqrt{t}\epsilon\gamma(1 - \beta_1)}$$

We use the fact that $\alpha_t = \alpha/\sqrt{t}$. By summing over $T$ iterations, we obtain

$$\sum_{t=1}^{T}\mathbb{E}[\|\theta_t^* - \theta^*\|] \le \frac{\alpha L}{\epsilon\gamma(1 - \beta_1)}\sum_{t=1}^{T}\frac{1}{\sqrt{t}} \le \frac{2\alpha L\sqrt{T}}{\epsilon\gamma(1 - \beta_1)}$$

The last inequality is by Lemma 6. Finally, we get

$$(18) \le (L/\gamma)\sum_{t=1}^{T}\mathbb{E}[\|\theta_t^* - \theta^*\|] \le \frac{2\alpha L^2\sqrt{T}}{\epsilon\gamma^2(1 - \beta_1)} = \mathcal{O}(\sqrt{T})$$

In summary, the cumulative regret $\mathcal{R}^\pi(T)$ is decomposed by the following:

$$\mathcal{R}^\pi(T) = \mathcal{R}_{\text{online}}^\pi(T) + (17) + (18)$$

where $(17) = 0$ and $(18) = \mathcal{O}(\sqrt{T})$. Thus, this completes the proof of Lemma 11, saying

$$\mathcal{R}^\pi(T) = \mathcal{R}_{\text{online}}^\pi(T) + \mathcal{O}(\sqrt{T})$$

$\square$

Now, we prove the main Theorem 1.

*Proof.* From Lemma 11, we have shown that the cumulative regret $\mathcal{R}^\pi(T)$ can be decomposed into the online regret $\mathcal{R}^\pi_{\text{online}}(T)$ with the additional sub-linear terms. Hence, we are left to bound the cumulative online regret $\mathcal{R}^\pi_{\text{online}}(T)$. Recall the first key lemma (Lemma 1):

$$\mathcal{R}^\pi_{\text{online}}(T) \le \rho_1 d\sqrt{T} + \sqrt{d}\rho_2 \sqrt{\frac{1}{n^2 K} \sum_{t=1}^{T} \mathbb{E}_{p_t}\left[\sum_{j \in J_t} \frac{\|g_{j,t}\|^2}{(p_{j,t})^2}\right]} + \rho_3$$

Recall also the second key lemma (Lemma 2):

$$\sum_{t=1}^{T} \mathbb{E}_{p_t}\left[\sum_{j \in J_t} \frac{\|g_{j,t}\|^2}{(p_{j,t})^2}\right] - \min_{p_t} \sum_{t=1}^{T} \mathbb{E}_{p_t}\left[\sum_{j \in J_t} \frac{\|g_{j,t}\|^2}{(p_{j,t})^2}\right] = \mathcal{O}\left(\sqrt{KnT \ln \frac{n}{K}}\right)$$

Let we denote $M := \min_{p_t} \sum_{t=1}^{T} \mathbb{E}_{p_t}\left[\sum_{j \in J_t} \frac{\|g_{j,t}\|^2}{(p_{j,t})^2}\right]$. Then by Lemma 2, we have

$$\sum_{t=1}^{T} \mathbb{E}_{p_t}\left[\sum_{j \in J_t} \frac{\|g_{j,t}\|^2}{(p_{j,t})^2}\right] = M + C\sqrt{KnT \ln \frac{n}{K}}$$

where $C > 0$ is a constant. By plugging above equation to Lemma 1, we obtain

$$\mathcal{R}^\pi_{\text{online}}(T) \le \rho_1 d\sqrt{T} + \rho_2 \frac{\sqrt{d}}{n\sqrt{K}} \sqrt{M + C\sqrt{KnT \ln \frac{n}{K}}} + \rho_3$$

$$\le \rho_1 d\sqrt{T} + \rho_2 \frac{\sqrt{d}}{n\sqrt{K}} \sqrt{M} + \rho_2 \frac{\sqrt{d}}{n\sqrt{K}} \sqrt{C\sqrt{KnT \ln \frac{n}{K}}} + \rho_3$$

$$= \rho_1 d\sqrt{T} + \frac{\rho_2 \sqrt{d}}{n\sqrt{K}} \sqrt{M} + \frac{\rho_4 \sqrt{d}}{n} \left(\frac{nT}{K} \ln \frac{n}{K}\right)^{1/4} + \rho_3$$

We use the fact that $\sqrt{a+b} \le \sqrt{a} + \sqrt{b}$ in the second inequality and we define $\rho_4 := \rho_2 \sqrt{C}$.

Now, we should consider $M$. Using the tower property, we can express $M$ as,

$$M = \min_{p_t} \sum_{t=1}^{T} \mathbb{E}_{p_t}\left[\sum_{j \in J_t} \frac{\|g_{j,t}\|^2}{(p_{j,t})^2}\right]$$

$$= \min_{p_t} \sum_{t=1}^{T} \mathbb{E}_{p_t}\left[\mathbb{E}_{p_t}\left[\sum_{j \in J_t} \frac{\|g_{j,t}\|^2}{(p_{j,t})^2} \mid p_t\right]\right]$$

$$= \min_{p_t} \sum_{t=1}^{T} \mathbb{E}_{p_t}\left[\sum_{i=1}^{n}\left[\sum_{j \in J_t} \frac{\|g_{i,t}\|^2}{(p_{i,t})^2} p_{i,t}\right]\right]$$

$$= \min_{p_t} \sum_{t=1}^{T} \mathbb{E}_{p_t}\left[\sum_{j \in J_t}\left[\sum_{i=1}^{n} \frac{\|g_{i,t}\|^2}{p_{i,t}}\right]\right]$$

$$= K \min_{p_t} \sum_{t=1}^{T} \mathbb{E}_{p_t}\left[\sum_{i=1}^{n} \frac{\|g_{i,t}\|^2}{p_{i,t}}\right]$$

For this minimization problem, it can be shown that for every iteration $t$, the optimal distribution $p_t^*$ is proportional to the gradient norm of individual example. Formally speaking, for any $t$, the optimal solution $p_t^*$ to the problem $\arg\min_{p_t} \sum_{t=1}^{T} \mathbb{E}_{p_t}\left[\sum_{i=1}^{n} \frac{\|g_{i,t}\|^2}{p_{i,t}}\right]$ is $(p_{j,t})^* = \frac{\|g_{j,t}\|}{\sum_{i=1}^{n} \|g_{i,t}\|}$ for all $j \in [n]$. By plugging this solution,

$$M = K \sum_{t=1}^{T} \left(\sum_{i=1}^{n} \|g_{i,t}\|\right)^2$$

By plugging $M$ to the online regret bound expression,

$$\mathcal{R}^{\pi}_{\text{online}}(T) \leq \rho_1 d\sqrt{T} + \rho_2 \frac{\sqrt{d}}{n\sqrt{K}} \sqrt{M} + \rho_4 \frac{\sqrt{d}}{n} \left(\frac{nT}{K} \ln \frac{n}{K}\right)^{1/4} + \rho_3$$

$$= \rho_1 d\sqrt{T} + \rho_2 \frac{\sqrt{d}}{n\sqrt{K}} \sqrt{K \sum_{t=1}^{T} \left(\sum_{i=1}^{n} \|g_{i,t}\|\right)^2} + \rho_4 \frac{\sqrt{d}}{n} \left(\frac{nT}{K} \ln \frac{n}{K}\right)^{1/4} + \rho_3$$

$$= \rho_1 d\sqrt{T} + \sqrt{d}\rho_2 \sqrt{\frac{1}{n^2} \sum_{t=1}^{T} \left(\sum_{i=1}^{n} \|g_{i,t}\|\right)^2} + \rho_4 \frac{\sqrt{d}}{n} \left(\frac{nT}{K} \ln \frac{n}{K}\right)^{1/4} + \rho_3$$

By Assumption 1, $\|g_{i,t}\| \leq L$ for $i \in [n]$ and $t \in [T]$. Then, the second term in the right-hand side of above inequality is bounded by $L\rho_2\sqrt{dT}$, which diminishes by the first term that have order of $\mathcal{O}(d\sqrt{T})$. Hence, the online regret $\mathcal{R}^{\pi}_{\text{online}}(T)$ after $T$ iterations is,

$$\mathcal{R}^{\pi}_{\text{online}}(T) \leq \mathcal{O}(d\sqrt{T}) + \mathcal{O}\left(\frac{\sqrt{d}}{n} \left(\frac{nT}{K} \ln \frac{n}{K}\right)^{1/4}\right)$$

Finally, by Lemma 11, we can bound the cumulative regret using the bound of the online regret as

$$\mathcal{R}^{\pi}(T) = \mathcal{R}^{\pi}_{\text{online}}(T) + \mathcal{O}(\sqrt{T}) \leq \mathcal{O}(d\sqrt{T}) + \mathcal{O}\left(\frac{\sqrt{d}}{n} \left(\frac{nT}{K} \ln \frac{n}{K}\right)^{1/4}\right) + \mathcal{O}(\sqrt{T})$$

$$= \mathcal{O}\left(d\sqrt{T} + \frac{\sqrt{d}}{n^{3/4}} \left(\frac{T}{K} \ln \frac{n}{K}\right)^{\frac{1}{4}}\right)$$

This completes the proof of Theorem 1. $\qquad\square$

## C  Proof for Convergence Rate when using Uniform Sampling

To compare the convergence rate between using uniform sampling and bandit sampling, we will now prove the following Theorem 2. It is important to note that Theorem 2 includes an additional condition—Assumption 3—which was not present in Theorem 1. This assumption plays a key role in distinguishing the results between these two theorems.

**Theorem 2.** *Suppose Assumptions 1,2, and 3 hold. The convergence rate for* `AdamX` *(variant of* `Adam`*) using uniform sampling is given by:*

$$
\mathcal{O}\left(d\sqrt{T} + \frac{\sqrt{d}}{n^{1/2}}\sqrt{T}\right)
$$

*Proof.* We start the proof from the first key lemma (Lemma 1):

**Lemma 1.** *Suppose Assumptions 1-2 hold.* `AdamCB` *(Algorithm 1) with a mini-batch of size $K$, which is formed dynamically by distribution $p_t$, achieves the following upper-bound for the cumulative online regret $\mathcal{R}_{online}^{\pi}(T)$ over $T$ iterations,*

$$
\mathcal{R}_{online}^{\pi}(T) \le \rho_1 d\sqrt{T} + \sqrt{d}\rho_2\sqrt{\frac{1}{n^2 K}\sum_{t=1}^{T}\mathbb{E}_{p_t}\left[\sum_{j \in J_t}\frac{\|g_{j,t}\|^2}{(p_{j,t})^2}\right]} + \rho_3 \tag{20}
$$

*where $\rho_1$, $\rho_2$, and $\rho_3$ are defined as follows:*

$$
\rho_1 = \frac{D^2 L}{2\alpha\gamma(1-\beta_1)^2}, \quad \rho_2 = \frac{\alpha\sqrt{1+\ln T}}{(1-\beta_1)^2\sqrt{1-\beta_2}(1-\eta)}, \quad \rho_3 = \frac{d\beta_1 D^2 L}{2\alpha\gamma(1-\beta_1)^2(1-\lambda)^2}
$$

*Note that $d$ is the dimension of parameter space and the inputs of Algorithm 1 follows these conditions: (a) $\alpha_t = \frac{\alpha}{\sqrt{t}}$, (b) $\beta_1$, $\beta_2 \in [0,1)$, $\beta_{1,t} := \beta_1\lambda^{t-1}$ for all $t \in [T]$, $\lambda \in (0,1)$, (c) $\eta = \beta_1/\sqrt{\beta_2} \le 1$, and (d) $\gamma \in [0,1)$.*

Consider the second term in the right-hand side of Eq.(20),

$$
\begin{aligned}
\frac{1}{n^2 K}\sum_{t=1}^{T}\mathbb{E}_{p_t}\left[\sum_{j \in J_t}\frac{\|g_{j,t}\|^2}{(p_{j,t})^2}\right] &= \frac{1}{n^2 K}\sum_{t=1}^{T}\mathbb{E}_{p_t}\left[\mathbb{E}_{p_t}\left[\sum_{j \in J_t}\frac{\|g_{j,t}\|^2}{(p_{j,t})^2} \mid p_t\right]\right] \\
&= \frac{1}{n^2 K}\sum_{t=1}^{T}\mathbb{E}_{p_t}\left[\sum_{i=1}^{n}\left[\sum_{j \in J_t}\frac{\|g_{i,t}\|^2}{(p_{i,t})^2}p_{i,t}\right]\right] \\
&= \frac{1}{n^2 K}\sum_{t=1}^{T}\mathbb{E}_{p_t}\left[\sum_{j \in J_t}\left[\sum_{i=1}^{n}\frac{\|g_{i,t}\|^2}{p_{i,t}}\right]\right] \\
&= \frac{1}{n^2}\sum_{t=1}^{T}\mathbb{E}_{p_t}\left[\sum_{i=1}^{n}\frac{\|g_{i,t}\|^2}{p_{i,t}}\right]
\end{aligned}
$$

The tower property is used in the first equality. Since $\sum_{i=1}^{n}\frac{\|g_{i,t}\|^2}{p_{i,t}}$ is independent to $j \in J_t$, the mini-batch size $K$ is multiplied in the last equality. Therefore, we can express the cumulative online regret $\mathcal{R}_{online}^{\pi}(T)$ as:

$$
\mathcal{R}_{online}^{\pi}(T) \le \rho_1 d\sqrt{T} + \sqrt{d}\rho_2\sqrt{\frac{1}{n^2}\sum_{t=1}^{T}\mathbb{E}_{p_t}\left[\sum_{i=1}^{n}\frac{\|g_{i,t}\|^2}{p_{i,t}}\right]} + \rho_3
$$

In the case when we select samples uniformly, we can set the probability distribution $p_t$ to satisfy $p_{i,t} = 1/n$ for all $t \in [T]$ and $i \in [n]$. By plugging it, we obtain

$$
\mathcal{R}_{online}^{\pi}(T) \le \rho_1 d\sqrt{T} + \sqrt{d}\rho_2\sqrt{\frac{1}{n}\sum_{t=1}^{T}\left[\sum_{i=1}^{n}\|g_{i,t}\|^2\right]} + \rho_3
$$

Now, recall Assumption 3:

**Assumption 3.** *There exists $\sigma > 0$ such that $\mathrm{Var}(\|g_{i,t}\|) \leq \sigma^2$ for all $i \in [n]$ and $t \in [T]$*

$$\frac{1}{n}\left[\sum_{i=1}^{n}\|g_{i,t}\|^2\right] \leq \left(\frac{1}{n}\sum_{i=1}^{n}\|g_{i,t}\|\right)^2 + \frac{\sigma^2}{n}$$

Therefore, the online regret bound $\mathcal{R}_{\mathrm{online}}^{\pi}(T)$ for uniform sampling is,

$$\mathcal{R}_{\mathrm{online}}^{\pi}(T) = \mathcal{O}(d\sqrt{T}) + \mathcal{O}\left(\sqrt{d}\sqrt{\frac{1}{n^2}\sum_{t=1}^{T}\left(\sum_{i=1}^{n}\|g_{i,t}\|\right)^2 + \frac{\sigma^2}{n}T}\right)$$

Applying the fact that $\sqrt{a + b} \leq \sqrt{a} + \sqrt{b}$, we obtain,

$$\mathcal{R}_{\mathrm{online}}^{\pi}(T) = \mathcal{O}(d\sqrt{T}) + \mathcal{O}\left(\sqrt{d}\sqrt{\frac{1}{n^2}\sum_{t=1}^{T}\left(\sum_{i=1}^{n}\|g_{i,t}\|\right)^2}\right) + \mathcal{O}\left(\sqrt{d}\sqrt{\frac{T}{n}}\right)$$

By Assumption 1, $\|g_{i,t}\| \leq L$ for $i \in [n]$ and $t \in [T]$. Then, the second term in the right-hand side of above inequality is bounded by $\mathcal{O}(\sqrt{dT})$, which diminishes by the first term that have order of $\mathcal{O}(d\sqrt{T})$. Hence, the online regret $\mathcal{R}_{\mathrm{online}}^{\pi}(T)$ after $T$ iterations is given by

$$\mathcal{R}_{\mathrm{online}}^{\pi}(T) = \mathcal{O}(d\sqrt{T}) + \mathcal{O}\left(\frac{\sqrt{d}}{n^{1/2}}\sqrt{T}\right)$$

Finally, by Lemma 11, we can bound the cumulative regret using the online regret, which completes the regret analysis for uniform sampling.

$$\mathcal{R}^{\pi}(T) = \mathcal{R}_{\mathrm{online}}^{\pi}(T) + \mathcal{O}(\sqrt{T}) = \mathcal{O}(d\sqrt{T}) + \mathcal{O}\left(\frac{\sqrt{d}}{n^{1/2}}\sqrt{T}\right) + \mathcal{O}(\sqrt{T})$$

$$= \mathcal{O}\left(d\sqrt{T} + \frac{\sqrt{d}}{n^{1/2}}\sqrt{T}\right)$$

$\square$

# D    CORRECTION OF ADAMBS (LIU ET AL., 2020)

This section introduces the corrected analysis for `AdamBS` (Liu et al., 2020). We use Algorithm 4 and Algorithm 5 for modified `AdamBS`.

---

**Algorithm 4:** (Corrected) Adam with Bandit Sampling (`AdamBS`)

---

**Input:** learning rate $\{\alpha_t\}_{t=1}^T$, decay rates $\{\beta_{1,t}\}_{t=1}^T$, $\beta_2$, batch size $K$, exploration parameter $\gamma \in [0,1)$

**Initialize:** model parameters $\theta_0$; first moment estimate $m_0 \leftarrow 0$; second moment estimate $v_0 \leftarrow 0, \hat{v}_0 \leftarrow 0$; sample weights $w_0^i \leftarrow 1$ for all $i \in [n]$

1 **for** $t = 1$ **to** $T$ **do**
2     Compute sample distribution $p_t$ for all $j \in [n]$
3

$$p_{j,t} = (1 - \gamma)\frac{w_{j,t-1}}{\sum_{i=1}^n w_{i,t-1}} + \frac{\gamma}{n}$$

    Select a mini-batch $\mathcal{D}_t := \{(x_j, y_j)\}_{j \in J_t}$ by sampling *with replacement* from $p_t$
4     Compute unbiased gradient estimate $g_t$ with respect to the mini-batch $\mathcal{D}_t$ using Eq.(7)
5     $m_t \leftarrow \beta_{1,t} m_{t-1} + (1 - \beta_{1,t})g_t$
6     $v_t \leftarrow \beta_2 v_{t-1} + (1 - \beta_2)g_t^2$
7     $\hat{v}_1 \leftarrow v_1, \hat{v}_t \leftarrow \max\left\{\frac{(1-\beta_{1,t})^2}{(1-\beta_{1,t-1})^2}\hat{v}_{t-1}, v_t\right\}$ if $t \geq 2$
8     $\theta_{t+1} \leftarrow \theta_t - \alpha_t m_t/(\sqrt{\hat{v}_t} + \epsilon)$
9     $w_t \leftarrow$ `Weight-Update`$(w_{t-1}, p_t, J_t, \{g_{j,t}\}_{j \in J_t}, \gamma)$ (Algorithm 5)

---

**Algorithm 5:** (Corrected) `Weight-Update` for `AdamBS`

---

**Input:** $w_{t-1}, p_t, J_t, \{g_{j,t}\}_{j \in J_t}$, and $\gamma \in [0,1)$

1 **for** $j = 1$ **to** $n$ **do**
2     Compute loss $\ell_{j,t} = \frac{p_{\min}^2}{L^2}\left(-\frac{\|g_{j,t}\|^2}{(p_{j,t})^2} + \frac{L^2}{p_{\min}^2}\right)$ if $j \in J_t$, otherwise, $\ell_{j,t} = 0$
3     Compute unbiased gradient estimate $\hat{\ell}_{j,t} = \frac{\ell_{j,t} \sum_{k=1}^K \mathbb{I}(j = J_t^k)}{K p_{j,t}}$
4     Update sample weights $w_{j,t} \leftarrow w_{j,t-1} \exp\left(-\gamma\hat{\ell}_{j,t}/n\right)$
5 **return** $w_t$

---

At iteration $t \in [T]$, `AdamBS` chooses a mini-batch $\mathcal{D}_t = \{(x_j, y_j)\}_{j \in J_t}$ of size $K$ according to probability distribution $p_t$ with replacement. We denote $J_t$ as the set of indices for the mini-batch $\mathcal{D}_t$. Then, the algorithm receives the loss, regarding losses from all chosen samples in the mini-batch $\mathcal{D}$ as one loss, is $\frac{1}{K}\sum_{j \in J_t} \ell_{j,t}$, denote as $\ell_{j,t} \in [0,1]$. The unbiased estimate of the loss $\hat{\ell}_{j,t}$ is,

$$\hat{\ell}_{j,t} = \frac{\ell_{j,t} \sum_{k=1}^K \mathbb{I}(j = J_t^k)}{K p_{j,t}}$$

We have a following key lemma concerning the rate of convergence of `AdamBS`.

**Lemma 12** (Corrected version of Lemma 1 in Liu et al. (2020))**.** *Suppose Assumptions 1-2 hold. If we set $\gamma = \min\left\{1, \sqrt{\frac{n \ln n}{(e-1)T}}\right\}$, the weight update rule (Algorithm 5) following* `AdamBS` *(Algorithm 4) implies*

$$\sum_{t=1}^T \mathbb{E}_{p_t}\left[\sum_{j \in J_t} \frac{\|g_{j,t}\|^2}{(p_{j,t})^2}\right] - \min_{p_t} \sum_{t=1}^T \mathbb{E}_{p_t}\left[\sum_{j \in J_t} \frac{\|g_{j,t}\|^2}{(p_{j,t})^2}\right] = \mathcal{O}(K\sqrt{nT \ln n})$$

*Proof.* We set $\ell_{j,t} = \frac{p_{min}^2}{L^2}\left(-\frac{\|g_{j,t}\|^2}{(p_{j,t})^2} + \frac{L^2}{p_{min}^2}\right)$ in Algorithm 5. Since, $\|g_{i,t}\|_2 \leq L$ and $p_{i,t} \geq p_{min}$ for all $t \in [T]$, $i \in [n]$ by Assumption 1, we have $\ell_{i,t} \in [0,1]$.

We use the following simple facts, which are immediately derived from the definitions,

$$\sum_{i=1}^{n} p_{i,t} \hat{\ell}_{i,t} = \frac{1}{K} \sum_{j \in J_t} \ell_{j,t} := \ell_t^{J_t} \tag{21}$$

$$\sum_{i=1}^{n} p_{i,t} (\hat{\ell}_{i,t})^2 = \sum_{i=1}^{n} p_{i,t} \left( \frac{\ell_{i,t} \sum_{k=1}^{K} \mathbb{I}(i = J_t^k)}{K p_{i,t}} \right) \hat{\ell}_{i,t} = \sum_{i=1}^{n} \ell_{i,t} \frac{\sum_{k=1}^{K} \mathbb{I}(i = J_t^k)}{K} \hat{\ell}_{i,t} \leq \sum_{i=1}^{n} \hat{\ell}_{i,t} \tag{22}$$

Let $W_t := \sum_{i=1}^{n} w_t$. Then, for any $t \in [T]$,

$$\frac{W_t}{W_{t-1}} = \sum_{i=1}^{n} \frac{w_{i,t}}{W_{t-1}}$$

$$= \sum_{i=1}^{n} \frac{w_{i,t-1}}{W_{t-1}} \exp\left( -\frac{\gamma}{n} \hat{\ell}_{i,t} \right)$$

The last equality is by the weight update rule in Algorithm 5. From the probability computation in Algorithm 4, we have

$$p_{i,t} = (1 - \gamma) \frac{w_{i,t-1}}{\sum_{j=1}^{n} w_{j,t-1}} + \frac{\gamma}{n} \geq \frac{\gamma}{n}$$

Thus, we obtain the following bound,

$$0 \leq \frac{\gamma}{n} \hat{\ell}_{i,t} = \frac{\gamma}{n} \left( \frac{\ell_{i,t} \sum_{k=1}^{K} \mathbb{I}(i = J_t^k)}{K p_{i,t}} \right) \leq \ell_{i,t} \leq 1$$

By the fact that $e^{-x} \leq 1 - x + (e - 2)x^2$ for all $x \in [0, 1]$, and considering $\frac{\gamma}{n} \hat{\ell}_{i,t}$ as $x$, we have

$$\frac{W_t}{W_{t-1}} \leq \sum_{i=1}^{n} \frac{w_{i,t-1}}{W_{t-1}} \left[ 1 - \frac{\gamma}{n} \hat{\ell}_{i,t} + (e - 2)\left(\frac{\gamma}{n} \hat{\ell}_{i,t}\right)^2 \right]$$

$$= \sum_{i=1}^{n} \frac{p_{i,t} - \gamma/n}{1 - \gamma} \left[ 1 - \frac{\gamma}{n} \hat{\ell}_{i,t} + (e - 2)\left(\frac{\gamma}{n} \hat{\ell}_{i,t}\right)^2 \right]$$

$$\leq 1 - \frac{\gamma/n}{1 - \gamma} \sum_{i=1}^{n} p_{i,t} \hat{\ell}_{i,t} + \frac{(e - 2)(\gamma/n)^2}{1 - \gamma} \sum_{i=1}^{n} p_{i,t} (\hat{\ell}_{i,t})^2$$

$$\leq 1 - \frac{\gamma/n}{1 - \gamma} \ell_t^{J_t} + \frac{(e - 2)(\gamma/n)^2}{1 - \gamma} \sum_{i=1}^{n} \hat{\ell}_{i,t}$$

The last inequality uses Eq.(21) and Eq.(22). Taking logarithms and using the fact that $\ln(1 + x) \leq x$ for all $x > -1$ gives

$$\ln \frac{W_t}{W_{t-1}} \leq -\frac{\gamma/n}{1 - \gamma} \ell_t^{J_t} + \frac{(e - 2)(\gamma/n)^2}{1 - \gamma} \sum_{i=1}^{n} \hat{\ell}_{i,t}$$

By summing over $t$, we obtain

$$\ln \frac{W_T}{W_1} \leq -\frac{\gamma/n}{1 - \gamma} \sum_{t=1}^{T} \ell_t^{J_t} + \frac{(e - 2)(\gamma/n)^2}{1 - \gamma} \sum_{t=1}^{T} \sum_{i=1}^{n} \hat{\ell}_{i,t}$$

On the other hand, for any action $j$,

$$\ln \frac{W_T}{W_1} \geq \ln \frac{w_{j,T}}{W_1} = -\frac{\gamma}{n} \sum_{t=1}^{T} \hat{\ell}_{j,t} - \ln n$$

From combining results,

$$\sum_{t=1}^{T} \ell_t^{J_t} \geq (1 - \gamma) \sum_{t=1}^{T} \hat{\ell}_{j,t} - \frac{n \ln n}{\gamma} - (e - 2)\frac{\gamma}{n} \sum_{t=1}^{T} \sum_{i=1}^{n} \hat{\ell}_{i,t}$$

We next take the expectation of both sides with respect to probability distribution $p_t$ and since $\mathbb{E}_{p_t}[\hat{\ell}_{j,t}] = \ell_{j,t}$, we have

$$\mathbb{E}_{p_t}[\sum_{t=1}^{T} \ell_t^{J_t}] \geq (1-\gamma)\sum_{t=1}^{T} \ell_{j,t} - \frac{n\ln n}{\gamma} - (e-2)\frac{\gamma}{n}\sum_{i=1}^{n}\sum_{t=1}^{T} \ell_{i,t}$$

Since $j \in J_t$ were chosen arbitrarily, we can choose the best $J_t^*$ for every iteration $t$. Let $L_{\text{MIN}}(T) := \sum_{t=1}^{T}\sum_{j \in J_t^*} \ell_{j,t}$ and $L_{\text{EXP3}}(T) := \sum_{t=1}^{T}\sum_{j \in J_t} \ell_{j,t}$. Summing over $j \in J_t^*$, and using the fact that $\sum_{t=1}^{T}\sum_{i=1}^{n} \ell_{i,t} \leq \frac{nL_{\text{MIN}}(T)}{K}$, we have the following statement,

$$\mathbb{E}[L_{\text{EXP3}}(T)] \geq (1-\gamma)L_{\text{MIN}}(T) - \frac{nK\ln n}{\gamma} - (e-2)\gamma L_{\text{MIN}}(T)$$

Then, we get the following,

$$L_{\text{MIN}}(T) - \mathbb{E}[L_{\text{EXP3}}(T)] \leq (e-1)\gamma L_{\text{MIN}}(T) + \frac{nK\ln n}{\gamma}$$

Using the fact that $L_{\text{MIN}}(T) \leq TK$ and choosing the input parameter as $\gamma = \min\left\{1, \sqrt{\frac{n\ln n}{(e-1)T}}\right\}$, we obtain the following,

$$L_{\text{MIN}}(T) - \mathbb{E}[L_{\text{EXP3}}(T)] \leq 2\sqrt{e-1}K\sqrt{nT\ln n} \leq 2.63K\sqrt{nT\ln n}$$

Therefore, considering the scaling factor, we have:

$$\sum_{t=1}^{T}\mathbb{E}_{p_t}\left[\sum_{j \in J_t}\frac{\|g_{j,t}\|^2}{(p_{j,t})^2}\right] - \min_{p_t}\sum_{t=1}^{T}\mathbb{E}_{p_t}\left[\sum_{j \in J_t}\frac{\|g_{j,t}\|^2}{(p_{j,t})^2}\right] = \frac{L^2}{p_{min}^2}(L_{\text{MIN}}(T) - \mathbb{E}[L_{\text{EXP3}}(T)])$$

$$\leq \frac{2.63L^2}{p_{min}^2}K\sqrt{nT\ln n}$$

$$= \mathcal{O}\left(K\sqrt{nT\ln n}\right)$$

$\square$

**Theorem 3** (Corrected version of Theorem 4 in Liu et al. (2020)). *Suppose Assumptions 1-2 hold. The convergence rate for (corrected)* `AdamBS` *using bandit sampling is given by:*

$$\mathcal{O}\left(d\sqrt{T} + \frac{\sqrt{d}}{n^{3/4}}(T\ln n)^{1/4}\right)$$

*Proof.* From Lemma 11, we have shown that the cumulative regret $\mathcal{R}^{\pi}(T)$ can be decomposed into the online regret $\mathcal{R}^{\pi}_{\text{online}}(T)$ with the additional sub-linear terms. Hence, we are left to bound the cumulative online regret $\mathcal{R}^{\pi}_{\text{online}}(T)$. Recall the first key lemma (Lemma 1):

$$\mathcal{R}^{\pi}_{\text{online}}(T) \leq \rho_1 d\sqrt{T} + \sqrt{d}\rho_2\sqrt{\frac{1}{n^2K}\sum_{t=1}^{T}\mathbb{E}_{p_t}\left[\sum_{j \in J_t}\frac{\|g_{j,t}\|^2}{(p_{j,t})^2}\right]} + \rho_3$$

We can apply Lemma 1 to `AdamBS` as `AdamCB`, since both `AdamBS` and `AdamCB` follow the same model parameter update rule. However, we use the corrected lemma (Lemma 12) for `AdamBS`, rather than applying the key lemma (Lemma 2) used for `AdamCB`. Recall Lemma 12:

$$\sum_{t=1}^{T}\mathbb{E}_{p_t}\left[\sum_{j \in J_t}\frac{\|g_{j,t}\|^2}{(p_{j,t})^2}\right] - \min_{p_t}\sum_{t=1}^{T}\mathbb{E}_{p_t}\left[\sum_{j \in J_t}\frac{\|g_{j,t}\|^2}{(p_{j,t})^2}\right] = \mathcal{O}(K\sqrt{nT\ln n})$$

Let we denote $M := \min_{p_t} \sum_{t=1}^{T} \mathbb{E}_{p_t} \left[ \sum_{j \in J_t} \frac{\|g_{j,t}\|^2}{(p_{j,t})^2} \right]$. Then by Lemma 12, we have

$$\sum_{t=1}^{T} \mathbb{E}_{p_t} \left[ \sum_{j \in J_t} \frac{\|g_{j,t}\|^2}{(p_{j,t})^2} \right] = M + C'K\sqrt{nT \ln n}$$

where $C' > 0$ is a constant. By plugging above equation to Lemma 1, we obtain

$$\mathcal{R}_{\text{online}}^{\pi}(T) \leq \rho_1 d\sqrt{T} + \rho_2 \frac{\sqrt{d}}{n\sqrt{K}} \sqrt{M + C'K\sqrt{nT \ln n}} + \rho_3$$

$$\leq \rho_1 d\sqrt{T} + \rho_2 \frac{\sqrt{d}}{n\sqrt{K}} \sqrt{M} + \rho_2 \frac{\sqrt{d}}{n\sqrt{K}} \sqrt{C'K\sqrt{nT \ln n}} + \rho_3$$

$$= \rho_1 d\sqrt{T} + \frac{\rho_2 \sqrt{d}}{n\sqrt{K}} \sqrt{M} + \frac{\rho_5 \sqrt{d}}{n} (nT \ln n)^{1/4} + \rho_3$$

We use the fact that $\sqrt{a+b} \leq \sqrt{a} + \sqrt{b}$ in the second inequality and we define $\rho_5 := \rho_2 \sqrt{C'}$. Now, we should consider $M$. Using the tower property and applying the optimal solution for $p_t$ at each iteration, we can express $M$ as,

$$M = K \sum_{t=1}^{T} \left( \sum_{i=1}^{n} \|g_{i,t}\| \right)^2$$

This follows the same argument as in the proof of Theorem 1 (see Appendix B.4). Then, by plugging $M$ to the online regret bound expression,

$$\mathcal{R}_{\text{online}}^{\pi}(T) \leq \rho_1 d\sqrt{T} + \rho_2 \frac{\sqrt{d}}{n\sqrt{K}} \sqrt{M} + \rho_5 \frac{\sqrt{d}}{n} (nT \ln n)^{1/4} + \rho_3$$

$$= \rho_1 d\sqrt{T} + \rho_2 \frac{\sqrt{d}}{n\sqrt{K}} \sqrt{K \sum_{t=1}^{T} \left( \sum_{i=1}^{n} \|g_{i,t}\| \right)^2} + \rho_5 \frac{\sqrt{d}}{n} (nT \ln n)^{1/4} + \rho_3$$

$$= \rho_1 d\sqrt{T} + \sqrt{d}\rho_2 \sqrt{\frac{1}{n^2} \sum_{t=1}^{T} \left( \sum_{i=1}^{n} \|g_{i,t}\| \right)^2} + \rho_5 \frac{\sqrt{d}}{n} (nT \ln n)^{1/4} + \rho_3$$

By Assumption 1, $\|g_{i,t}\| \leq L$ for $i \in [n]$ and $t \in [T]$. Then, the second term in the right-hand side of above inequality is bounded by $L\rho_2\sqrt{dT}$, which diminishes by the first term that have order of $\mathcal{O}(d\sqrt{T})$. Hence, the online regret $\mathcal{R}_{\text{online}}^{\pi}(T)$ after $T$ iterations is,

$$\mathcal{R}_{\text{online}}^{\pi}(T) = \mathcal{O}(d\sqrt{T}) + \mathcal{O}\left( \frac{\sqrt{d}}{n} (nT \ln n)^{1/4} \right)$$

Finally, by Lemma 11, we can bound the cumulative regret using the bound of the online regret as

$$\mathcal{R}^{\pi}(T) = \mathcal{R}_{\text{online}}^{\pi}(T) + \mathcal{O}(\sqrt{T}) = \mathcal{O}(d\sqrt{T}) + \mathcal{O}\left( \frac{\sqrt{d}}{n} (nT \ln n)^{1/4} \right) + \mathcal{O}(\sqrt{T})$$

$$= \mathcal{O}\left( d\sqrt{T} + \frac{\sqrt{d}}{n^{3/4}} (T \ln n)^{1/4} \right)$$

This completes the proof of Theorem 3. □

# E Additional Algorithm

## E.1 DepRound Algorithm

---

**Algorithm 6:** DepRound

---

**Input:** Natural number $K(< n)$, sample distribution $\mathbf{p} := (p^1, p^2, \ldots, p^n)$ with $\sum_{i=1}^{n} p^i = K$

**Output:** Subset of $[n]$ with distinct $K$ elements

1 **while** there is an $i$ with $0 < p^i < 1$ **do**

2      Choose distinct $i, j$ with $0 < p^i < 1$ and $0 < p^j < 1$

3      Set $\alpha = \min\{1 - p^i, p^j\}$ and $\beta = \min\{p^i, 1 - p^j\}$

4      Update $p^i$ and $p^j$ as:

$$(p^i, p^j) = \begin{cases} (p^i + \alpha, p^j - \alpha) & \text{with probability} \frac{\beta}{\alpha+\beta} \\ (p^i - \beta, p^j + \beta) & \text{with probability} \frac{\alpha}{\alpha+\beta} \end{cases}$$

5 **return** $\{i : p^i = 1, 1 \le i \le n\}$

---

The DepRound (Gandhi et al., 2006) (Dependent Rounding) algorithm is used to select a subset of elements from a set while maintaining certain probabilistic properties. It ensures that the sum of probabilities is preserved and elements are chosen with the correct marginal probabilities.

# F More on Numerical Experiments

## F.1 Details on Experimental Setup

We compared our method, AdamCB, with Adam, AdamX, and corrected AdamBS. The experiments measured training loss and test loss, averaged over five runs with different random seeds, and included 1-sigma error bars for reliability. Throughout the entire experiments, identical hyper-parameters are used with any tuning as shown in Table 2.

Table 2: Hyper-parameters used for experiments

| Hyper-parameter | Value |
|---|---|
| Learning rate $\alpha_t$ | 0.001 |
| Exponential decay rates for momentum $\beta_{1,1}, \beta_2$ | 0.9, 0.999 |
| Decay rate for $\beta_{1,1}$ for convergence guarantee $\lambda$ | 1-1e-8 |
| $\epsilon$ for non-zero division | 1e-8 |
| Loss Function | Cross-Entropy |
| Batch Size $K$ | 128 |
| exploration parameter $\gamma$ | 0.4 |
| Number of epochs | 10 |

We trained MLP models on the MNIST, Fashion MNIST, and CIFAR-10 datasets. The detailed architectures of the MLP models for each dataset are provided in Table 3.

Table 3: MLP Architecture for MNIST/Fashion MNIST (left) and CIFAR10 (right)

| Layer Type | Input | Output | Layer Type | Input | Output |
|---|---|---|---|---|---|
| Flatten | (N, 28281) | (N, 28281) | Flatten | (N, 32323) | (N, 32323) |
| Dense + ReLU | (N, 28281) | (N, 512) | Dense + ReLU | (N, 32323) | (N, 512) |
| Dense + ReLU | (N, 512) | (N, 256) | Dense + ReLU | (N, 512) | (N, 256) |
| Dense | (N, 256) | (N, 10) | Dense | (N, 256) | (N, 10) |

We also trained CNN models on the same datasets. The detailed architectures of the CNN models for each dataset are presented in Table 4.

Table 4: CNN Architecture for MNIST/Fashion MNIST (left) and CIFAR10 (right)

| Layer Type | Input | Output |
| --- | --- | --- |
| Conv + ReLU | (N, 1, 28, 28) | (N, 32, 28, 28) |
| MaxPool | (N, 32, 28, 28) | (N, 32, 14, 14) |
| Conv + ReLU | (N, 32, 14, 14) | (N, 64, 14, 14) |
| MaxPool | (N, 64, 14, 14) | (N, 64, 7, 7) |
| Flatten | (N, 64, 7, 7) | (N, 3136) |
| Dense | (N, 3136) | (N, 128) |
| Dense + Softmax | (N, 128) | (N, 10) |

| Layer Type | Input | Output |
| --- | --- | --- |
| Conv + ReLU | (N, 3, 32, 32) | (N, 64, 32, 32) |
| MaxPool | (N, 64, 32, 32) | (N, 64, 16, 16) |
| Conv + ReLU | (N, 64, 16, 16) | (N, 128, 16, 16) |
| MaxPool | (N, 128, 16, 16) | (N, 128, 8, 8) |
| Conv + ReLU | (N, 128, 8, 8) | (N, 256, 8, 8) |
| MaxPool | (N, 256, 8, 8) | (N, 256, 4, 4) |
| Flatten | (N, 256, 4, 4) | (N, 25644) |
| Dense | (N, 25644) | (N, 512) |
| Dense + Softmax | (N, 512) | (N, 10) |

Table 5: VGG Architecture for MNIST/Fashion MNIST (left) and CIFAR10 (right)

| Layer Type | Input | Output | Layer Type | Input | Output |
| --- | --- | --- | --- | --- | --- |
| Conv + ReLU | (N, 1, 28, 28) | (N, 64, 28, 28) | Conv + ReLU | (N, 3, 32, 32) | (N, 64, 32, 32) |
| Conv + ReLU | (N, 64, 28, 28) | (N, 64, 28, 28) | Conv + ReLU | (N, 64, 32, 32) | (N, 64, 32, 32) |
| MaxPool | (N, 64, 28, 28) | (N, 64, 14, 14) | MaxPool | (N, 64, 32, 32) | (N, 64, 16, 16) |
| Conv + ReLU | (N, 64, 14, 14) | (N, 128, 14, 14) | Conv + ReLU | (N, 64, 16, 16) | (N, 128, 16, 16) |
| Conv + ReLU | (N, 128, 14, 14) | (N, 128, 14, 14) | Conv + ReLU | (N, 128, 16, 16) | (N, 128, 16, 16) |
| MaxPool | (N, 128, 14, 14) | (N, 128, 7, 7) | MaxPool | (N, 128, 16, 16) | (N, 128, 8, 8) |
| Conv + ReLU | (N, 128, 7, 7) | (N, 256, 7, 7) | Conv + ReLU | (N, 128, 8, 8) | (N, 256, 8, 8) |
| Conv + ReLU | (N, 256, 7, 7) | (N, 256, 7, 7) | Conv + ReLU | (N, 256, 8, 8) | (N, 256, 8, 8) |
| Conv + ReLU | (N, 256, 7, 7) | (N, 256, 7, 7) | Conv + ReLU | (N, 256, 8, 8) | (N, 256, 8, 8) |
| MaxPool | (N, 256, 7, 7) | (N, 256, 3, 3) | MaxPool | (N, 256, 8, 8) | (N, 256, 4, 4) |
| Flatten | (N, 256, 3, 3) | (N, 2304) | Flatten | (N, 256, 4, 4) | (N, 4096) |
| Dense | (N, 2304) | (N, 512) | Dense | (N, 4096) | (N, 512) |
| Dense | (N, 512) | (N, 512) | Dense | (N, 512) | (N, 512) |
| Dense | (N, 512) | (N, 10) | Dense | (N, 512) | (N, 10) |

We also evaluated the `AMSGrad` optimizer and the corrected `AdamBS` algorithm (Algorithm 4) on the CIFAR-10 dataset using both MLP and CNN models. The results are presented in Figures 3 and 4. From these plots, it is evident that our `AdamCB` algorithm outperforms the other `Adam`-based algorithms. To further assess performance, we conducted experiments using the VGG model, which is a larger architecture compared to the MLP and CNN models. The detailed structure of the VGG architecture is provided in Table 5, and the results are shown in Figure 5.

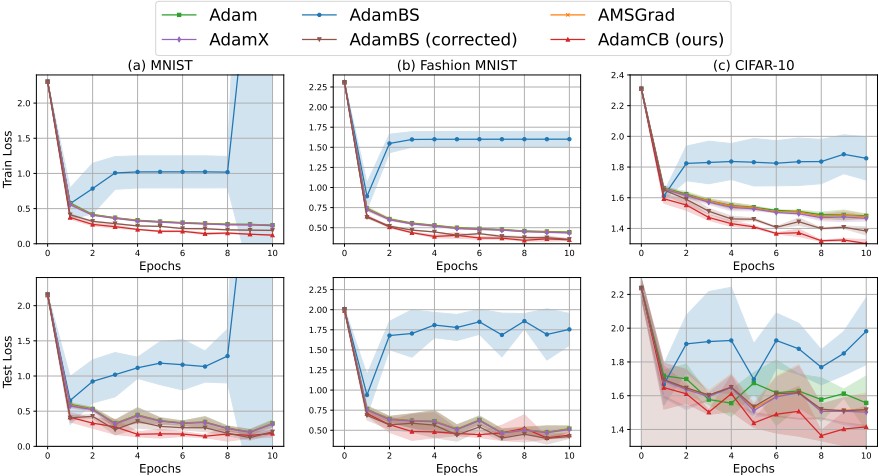

Figure 3: Comparison of Adam-based optimizations on MLP model

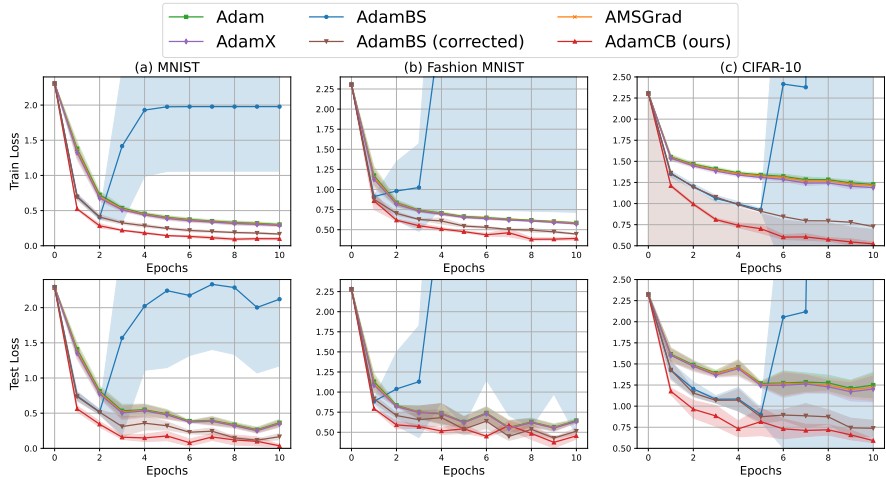

Figure 4: Comparison of Adam-based optimizations on CNN model

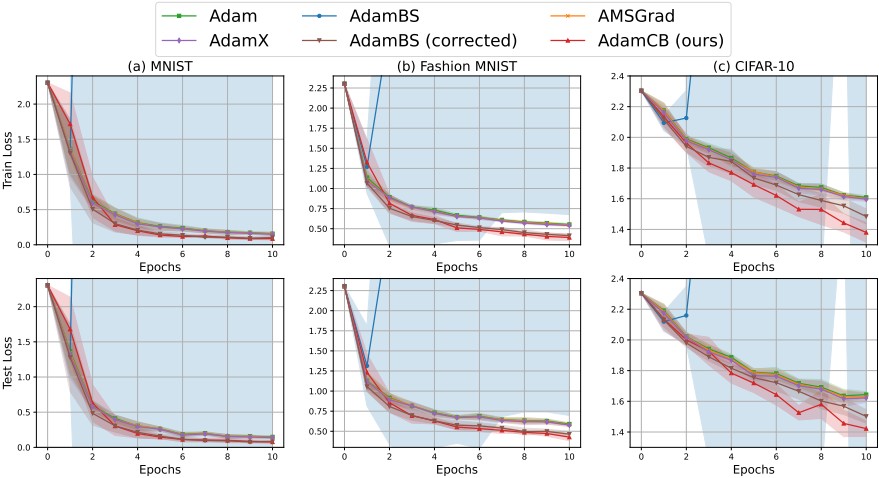

Figure 5: Comparison of Adam-based optimizations on VGG model

## F.2 Additional Experiments

To further evaluate the effectiveness of our proposed method, we conducted additional experiments using logistic regression, ResNet-18 (He et al., 2016), ConvNeXt-Base (Liu et al., 2022), and

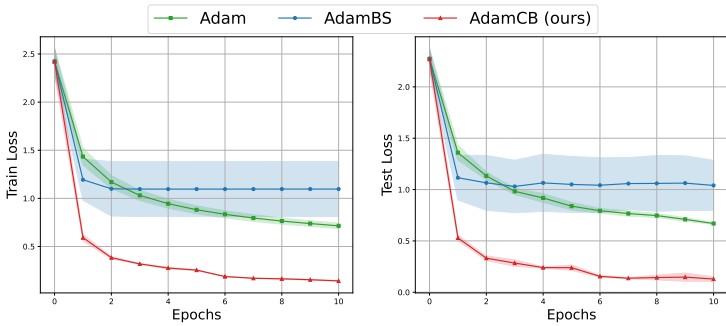

Figure 6: Comparison of Adam-based optimizations on the logistic regression model (MNIST)

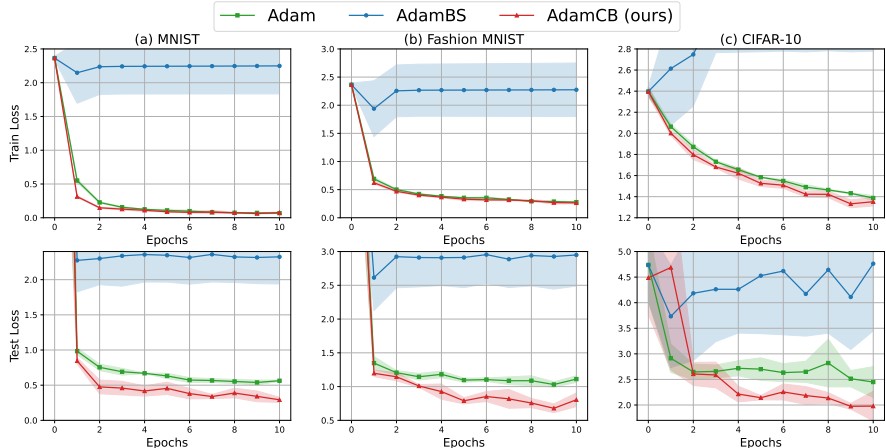

Figure 7: Comparison of Adam-based optimizations on ResNet-18 model

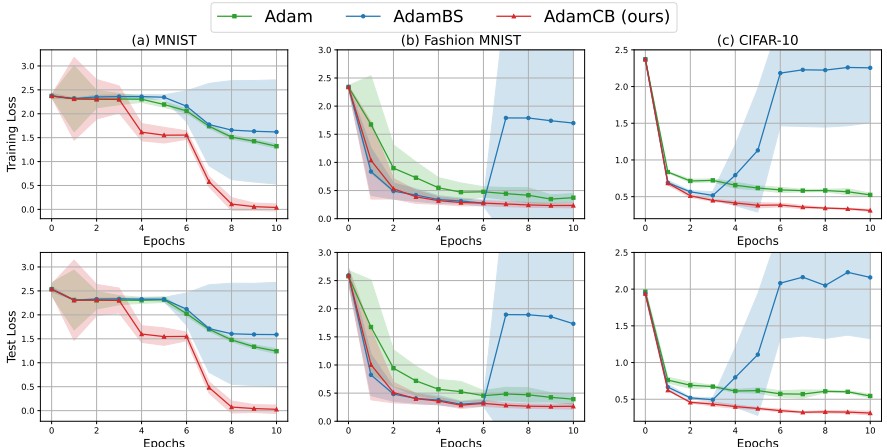

Figure 8: Comparison of Adam-based optimizations on ConvNext-base model

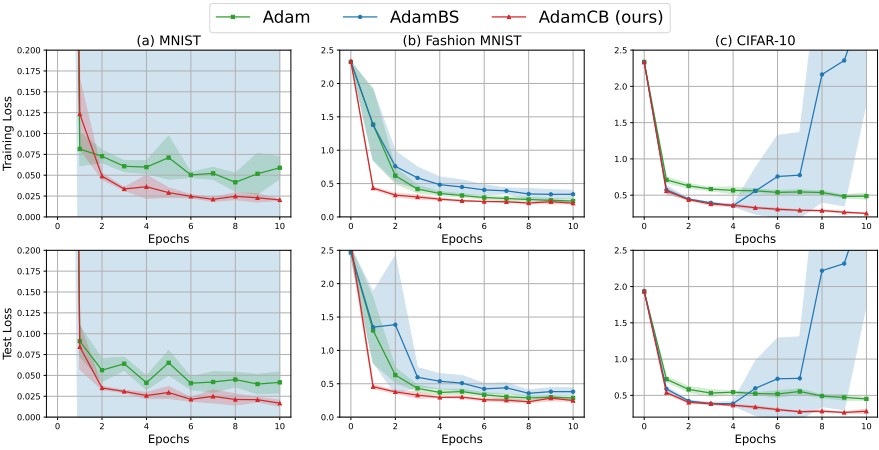

Figure 9: Comparison of Adam-based optimizations on ConvNext-large model

ConvNeXt-Large (Liu et al., 2022) networks. The archecture of The logistic regression model was employed to assess the performance of our algorithm in convex optimization settings.

For general non-convex optimization, we tested our method on the ResNet-18, ConvNeXt-Base, and ConvNeXt-Large models. Notably, ResNet-18 (11.4 million parameters), ConvNeXt-Base (89 million parameters), and ConvNeXt-Large (198 million parameters) are substantially larger architectures compared to the simple MLP and CNN models evaluated in the previous section. These experiments demonstrate the scalability and efficiency of our algorithm on larger, more complex models.

In all experiments, our proposed algorithm, AdamCB, consistently outperformed existing methods, reaffirming its effectiveness across both convex and non-convex optimization tasks and on models of varying complexity.

## G    WHEN $L$ IS NOT KNOWN

---

**Algorithm 7:** `Weight-Update` (with unknown $L$)

---
**Input:** $w_{t-1}, p_t, J_t, \{g_{j,t}\}_{j \in J_t}, S_{\text{null},t}, \gamma \in [0,1), L_{t-1}$

1  Set $L_t \leftarrow \max(L_{t-1}, \max_{j \in J_t} \|g_{j,t}\|)$
2  **for** $j = 1$ **to** $n$ **do**
3  $\quad$ Compute loss $\ell_{j,t} = \frac{p_{\min}^2}{L_t^2} \left( -\frac{\|g_{j,t}\|^2}{(p_{j,t})^2} + \frac{L_t^2}{p_{\min}^2} \right)$ if $j \in J_t$; otherwise $\ell_{j,t} = 0$
4  $\quad$ **if** $j \notin S_{\text{null},t}$ **then**
5  $\quad\quad$ $w_{j,t} \leftarrow w_{j,t-1} \exp\left( -K\gamma \ell_{j,t}/n \right)$

6  **return** $w_t, L_t$

---

**Lemma 13.** *(Lemma 9 when $L$ is unknown) For all $t \geq 1$, we have*

$$\sqrt{\hat{v}_t} \leq \frac{L_t}{\gamma(1 - \beta_1)}$$

*where $\hat{v}_t$ is in `AdamCB` (Algorithm 1).*

*Proof.* The argument follows the same reasoning as presented in Lemma 9, with the modification that $L$ is replaced by $L_t$, reflecting the condition that $\|g_{i,t}\| \leq L_t$ for all $i \in [n]$ at any $t$. $\qquad\square$

**Lemma 14.** *(Lemma 1 when $L$ is unknown) Suppose Assumptions 1-2 hold. `AdamCB` (Algorithm 1) with a mini-batch of size $K$, which is formed dynamically by distribution $p_t$, achieves the following upper-bound for the cumulative online regret $\mathcal{R}^\pi_{online}(T)$ over $T$ iterations,*

$$\mathcal{R}^\pi_{online}(T) \leq \rho_1' d\sqrt{T} + \sqrt{d}\rho_2' \sqrt{\frac{1}{n^2 K} \sum_{t=1}^{T} \mathbb{E}_{p_t} \left[ \sum_{j \in J_t} \frac{\|g_{j,t}\|^2}{(p_{j,t})^2} \right]} + \rho_3'$$

*where $\rho_1'$, $\rho_2'$, and $\rho_3'$ are defined as follows:*

$$\rho_1' = \frac{D^2 L_T}{2\alpha\gamma(1-\beta_1)^2}, \quad \rho_2' = \frac{\alpha\sqrt{1 + \ln T}}{(1-\beta_1)^2 \sqrt{1-\beta_2}(1-\eta)}, \quad \rho_3' = \frac{d\beta_1 D^2 L_T}{2\alpha\gamma(1-\beta_1)^2(1-\lambda)^2}$$

*Note that $d$ is the dimension of parameter space and the inputs of Algorithm 1 follows these conditions: (a) $\alpha_t = \frac{\alpha}{\sqrt{t}}$, (b) $\beta_1, \beta_2 \in [0,1)$, $\beta_{1,t} := \beta_1 \lambda^{t-1}$ for all $t \in [T]$, $\lambda \in (0,1)$, (c) $\eta = \beta_1/\sqrt{\beta_2} \leq 1$, and (d) $\gamma \in [0,1)$.*

*Proof.* The proof is the same as Lemma 1 until bounding the terms (13), (14), and (15).

**Bound for the term (13).**    Following the same reasoning as Lemma 1, we have

$$\mathbb{E}\left[ \sum_{u=1}^{d} \sum_{t=1}^{T} \frac{\sqrt{\hat{v}_{t,u}}}{2\alpha_t(1-\beta_{1,t})} \left( (\theta_{t,u} - \theta_{,u}^*)^2 - (\theta_{t+1,u} - \theta_{,u}^*)^2 \right) \right] \leq \frac{D^2}{2\alpha} \sum_{u=1}^{d} \frac{\sqrt{T\hat{v}_{T,u}}}{(1-\beta_{1,T})} \leq \frac{dD^2 L_t}{2\alpha\gamma(1-\beta_1)^2} \sqrt{T}$$

where the last inequality is by Lemma 13.

**Bound for the term (14).**    Nothing changes here.

**Bound for the term (15).**    Following the same reasoning as Lemma 1, we obtain

$$\mathbb{E}\left[ \sum_{u=1}^{d} \sum_{t=2}^{T} \frac{\beta_{1,t}\sqrt{\hat{v}_{t-1,u}}}{2\alpha_{t-1}(1-\beta_1)} (\theta_{,u}^* - \theta_{t,u})^2 \right] \leq \frac{D^2}{2\alpha(1-\beta_1)} \mathbb{E}\left[ \sum_{u=1}^{d} \sum_{t=2}^{T} \beta_{1,t}\sqrt{(t-1)\hat{v}_{t-1,u}} \right]$$

Therefore, from Lemma 13, we obtain

$$\mathbb{E}\left[ \sum_{u=1}^{d} \sum_{t=2}^{T} \frac{\beta_{1,t}\sqrt{\hat{v}_{t-1,u}}}{2\alpha_{t-1}(1-\beta_1)} (\theta_{,u}^* - \theta_{t,u})^2 \right] \leq \frac{dD^2}{2\alpha\gamma(1-\beta_1)^2} \mathbb{E}\left[ \sum_{t=2}^{T} \beta_{1,t} L_t \sqrt{(t-1)} \right] \qquad (23)$$

Since $L_t$ is a running max, $\{L_t\}_{t=1}^T$ is a non-decreasing sequence, i.e., $L_1 \leq L_2 \leq \cdots \leq L_T$. Thus, the inequality (23) becomes

$$\mathbb{E}\left[\sum_{u=1}^d \sum_{t=2}^T \frac{\beta_{1,t}\sqrt{\hat{v}_{t-1,u}}}{2\alpha_{t-1}(1-\beta_1)}(\theta_{,u}^* - \theta_{t,u})^2\right] \leq \frac{dD^2 L_T}{2\alpha\gamma(1-\beta_1)^2}\mathbb{E}\left[\sum_{t=2}^T \beta_{1,t}\sqrt{(t-1)}\right]$$

Note that

$$\sum_{t=2}^T \beta_{1,t}\sqrt{(t-1)} = \sum_{t=2}^T \beta_1 \lambda^{t-1}\sqrt{(t-1)} \leq \sum_{t=2}^T \beta_1 \sqrt{(t-1)}\lambda^{t-1} \leq \sum_{t=2}^T \beta_1 t \lambda^{t-1} \leq \frac{\beta_1}{(1-\lambda)^2}$$

where the first inequality is from the fact that $\beta_1 \leq 1$, and the last inequality is from Lemma 5. Thus, the bound for the term (15) is

$$\mathbb{E}\left[\sum_{u=1}^d \sum_{t=2}^T \frac{\beta_{1,t}\sqrt{\hat{v}_{t-1,u}}}{2\alpha_{t-1}(1-\beta_1)}(\theta_{,u}^* - \theta_{t,u})^2\right] \leq \frac{d\beta_1 D^2 L_T}{2\alpha\gamma(1-\beta_1)^2(1-\lambda)^2}$$

We now bounded three terms: (13), (14), and (15). Hence,

$$\mathcal{R}_{\text{online}}^\pi(T) \leq \frac{dD^2 L_T}{2\alpha\gamma(1-\beta_1)^2}\sqrt{T} + \frac{\alpha\sqrt{\ln T + 1}}{(1-\beta_1)^2\sqrt{1-\beta_2}(1-\eta)}\sum_{u=1}^d \mathbb{E}\left[\|g_{1:T,u}\|\right]$$
$$+ \frac{d\beta_1 D^2 L_T}{2\alpha\gamma(1-\beta_1)^2(1-\lambda)^2}$$

Thus, we can express $\mathcal{R}_{\text{online}}^\pi(T)$ as

$$\mathcal{R}_{\text{online}}^\pi(T) \leq \rho_1' d\sqrt{T} + \rho_2' \sum_{u=1}^d \mathbb{E}\left[\|g_{1:T,u}\|\right] + \rho_3'$$

where $\rho_1'$, $\rho_2'$, and $\rho_3'$ are defined as the following:

$$\rho_1' = \frac{D^2 L_T}{2\alpha\gamma(1-\beta_1)^2}, \quad \rho_2' = \frac{\alpha\sqrt{1+\ln T}}{(1-\beta_1)^2\sqrt{1-\beta_2}(1-\eta)}, \quad \rho_3' = \frac{d\beta_1 D^2 L_T}{2\alpha\gamma(1-\beta_1)^2(1-\lambda)^2}$$

The subsequent proof process is same as Lemma 1. $\qquad\square$

Note that, by Assumption 1, $L_T$ is always less than or equal to the theoretical upper bound of the maximum gradient norm across all iterations ($L$). Hence, we have $\rho_1' \leq \rho_1$, $\rho_2' = \rho_2$, and $\rho_3' = \rho_3$. This implies that Lemma 1 holds even when $L$ is not known.

**Lemma 15.** *(Lemma 2 when $L$ is unknown) Suppose Assumptions 1-2 hold. If we set $\gamma = \min\left\{1, \sqrt{\frac{n\ln(n/K)}{(e-1)TK}}\right\}$, the batch selection (Algorithm 2) and the weight update rule (Algorithm 7) following* `AdamCB` *(Algorithm 1) implies*

$$\sum_{t=1}^T \mathbb{E}_{p_t}\left[\sum_{j\in J_t} \frac{\|g_{j,t}\|^2}{(p_{j,t})^2}\right] - \min_{p_t}\sum_{t=1}^T \mathbb{E}_{p_t}\left[\sum_{j\in J_t} \frac{\|g_{j,t}\|^2}{(p_{j,t})^2}\right] = \mathcal{O}\left(\sqrt{KnT\ln\frac{n}{K}}\right)$$

*Proof.* The proof is the same as Lemma 2. However, at the last part, where we scale,

$$\sum_{t=1}^T \mathbb{E}_{p_t}\left[\sum_{j\in J_t} \frac{L_t^2}{p_{min}^2}\frac{\|g_{j,t}\|^2}{(p_{j,t})^2}\right] - \min_{p_t}\sum_{t=1}^T \mathbb{E}_{p_t}\left[\sum_{j\in J_t} \frac{L_t^2}{p_{min}^2}\frac{\|g_{j,t}\|^2}{(p_{j,t})^2}\right] = L_{\text{MIN-K}}(T) - \mathbb{E}[L_{\text{EXP3-K}}(T)] \quad (24)$$

Since $L_t$ is a running max, $\{L_t\}_{t=1}^T$ is a non-decreasing sequence, i.e., $L_1 \leq L_2 \leq \cdots \leq L_T$. Hence, Eq.(24) becomes

$$L_{\text{MIN-K}}(T) - \mathbb{E}[L_{\text{EXP3-K}}(T)] \leq \frac{2.63 L_T^2}{p_{min}^2}\sqrt{KnT\ln\frac{n}{K}}$$

By Assumption 1, $L_T$ is always less than or equal to $L$, which implies $L_T = \mathcal{O}(1)$. This completes the proof of Lemma 15. $\qquad\square$

Lemma 15 implies that Lemma 2 holds even when $L$ is not known.

**Theorem 4.** *(Regret bound of* `AdamCB` *(Theorem 1) when $L$ is unknown) Suppose Assumptions 1-2 hold, and we run* `AdamCB` *for a total $T$ iterations with $\alpha_t = \frac{\alpha}{\sqrt{t}}$ and with $\beta_{1,t} := \beta_1 \lambda^{t-1}$, $\lambda \in (0,1)$. Then, the cumulative regret of* `AdamCB` *(Algorithm 1) with batch size $K$ is upper-bounded by*

$$\mathcal{O}\left( d\sqrt{T} + \frac{\sqrt{d}}{n^{3/4}} \left( \frac{T}{K} \ln \frac{n}{K} \right)^{1/4} \right). \tag{25}$$

*Proof.* The overall proof is similar to the proof of Theorem 1 (when $L$ is known) detailed in Appendix B.4. The part that is different is when bounding the term Eq.(18) in Lemma 11.

$$(18) \leq (L_T/\gamma) \sum_{t=1}^{T} \mathbb{E}[\|\theta_t^* - \theta^*\|] \leq \frac{2\alpha L_T^2 \sqrt{T}}{\epsilon \gamma^2 (1 - \beta_1)}$$

By Assumption 1, $L_T$ is always less than or equal to the upper bound of the maximum gradient norm across all iterations ($L$), which implies $L_T = \mathcal{O}(1)$. Therefore, we have $(18) = \mathcal{O}(\sqrt{T})$. This implies that Lemma 11 still holds. Since both Lemma 1 and Lemma 2 hold even when $L$ is not known according to Lemma 14 and Lemma 15, we complete the proof of Theorem 4 by following the same proof process as Theorem 1.

$\square$

