# OpenReview forum: "ADAM Optimization with Adaptive Batch Selection"
_ICLR.cc/2025/Conference — ICLR 2025 Poster_

### Official Review · Reviewer_fG67 · 2024-10-31

**Soundness:** 3
**Presentation:** 3
**Contribution:** 3
**Rating:** 6
**Confidence:** 3

**Summary:**

This paper proposes an extension of the Adam optimizer by integrating adaptive batch selection. It also identifies a flaw in the proof presented in previous papers. Based on extensive theoretical analysis and some experiments, the proposed method demonstrates better convergence and improved performance.

**Strengths:**

1. This paper clearly points out the flaw in the proof of previous papers.
2. This paper provides a new convergence rate of Adam/AdamBS in the new perspective.
3. The results on some simple datasets are better than baselines.

**Weaknesses:**

I appreciate the theoretical contribution, and my major concern is the applicability of this method in real-world applications. Since the optimizer is one of the most fundamental components of the entire machine learning process, I wonder if this method can be directly integrated into existing practices. Additionally, I question whether the performance remains robust given the extra effort required to assign sampling bias towards certain samples during batch construction. Please see the question part for more details.

**Questions:**

1. **Semi-supervised learning**: Semi-supervised learning is a popular machine learning task where the testing dataset is given without labels. In this context, I wonder if the adaptive sampling method can still assign weights to both training and unlabeled testing samples.

2. **Extra regularizations**: Many regularization techniques are sample-agnostic, such as weight decay and dropout. Sometimes, regularization is only related to the output instead of the input, for example, using a total-variation loss to encourage smoothness in image generation. How does the adaptive sampling method work in these cases?

3. **Adversarial samples/noises**: The paper mentions that "samples with a low gradient norm are assigned a low weight, whereas samples with larger gradient norms are more likely to be chosen in future iterations." What if there is noise in the dataset, which is common in real-world datasets? Is it beneficial for the model to learn more from noisy samples that are difficult to fit?

4. **Data augmentation**. Does this method compatible with data augmentation? The augmented data will share strong similarity among samples. What should be considered when using data augmentation? Will the training be biased in an unexpected manner?

5. **Comparison with other methods for data sampling**: Given the popularity of large language models (LLMs), it is important to fine-tune/pretrain LLMs with a wise combination of data. Is there any potential to demonstrate this method in such a setting? How does this method compare with currently adopted reinforcement learning-based methods for sampling, such as "DoReMi: Optimizing Data Mixtures Speeds Up Language Model Pretraining, NeurIPS 2023"?

---

> ### Author Response · Authors · 2024-11-22
>
> Thank you for taking the time to review our paper and for your thoughtful and valuable feedback. We appreciate your positive recognition of our work and the constructive comments you have provided. Below, we address each of your comments and questions in detail:
>
> ---
>
> **[W] Applicability of the Method in Real-World Applications**
>
> We would like to clarify that our paper primarily focuses on **optimization with adaptive batch selection using a bandit method**, and our contributions are centered on this specific optimization framework. While the reviewer raises important questions regarding the type of task, model dependence (e.g., regularization techniques like weight decay and dropout), and data-related concerns (e.g., noise and augmentation), we emphasize that the proposed method is **task-agnostic and model-independent**, as long as the loss function can be computed.
>
> Below, we specifically address each aspect while reiterating the broader applicability of our method:
>
> 1. **Type of Task:** The method is not restricted to any particular task or domain. It dynamically adapts to the task at hand by leveraging the computed loss and gradient norms, making it applicable to supervised learning, semi-supervised learning, and even self-supervised learning setups.
> 2. **Model Dependence (Regularization Techniques):** Our approach does not interfere with standard or task-specific regularization techniques such as weight decay, dropout, or total variation loss. These regularizations function independently of how batches are constructed. Our method focuses solely on selecting the most informative samples for efficient optimization, complementing these regularization strategies.
> 3. **Data Samples (Noise and Augmentation):** Regarding data augmentation, the approach is fully compatible and can incorporate augmented samples effectively without biasing the training process, provided the augmentation strategy is balanced.
>
> In summary, while the reviewer raises interesting questions, the scope of our paper is the general **optimization method** rather than the intricacies of specific tasks, models, or data manipulations. We hope this clarification resolves any questions and underscores the broad applicability of our combinatorial bandit-based (adaptive batch selection) optimization method.
>
>
> ---
>
> ### **Answers to Questions**
>
> **[Q1] Semi-Supervised learning**
>
> Once a specific loss function is defined, the feedback is derived from the gradient of that loss function. Therefore, even in semi-supervised learning settings, **as long as a loss can be computed,** our proposed method can be applied. For unlabeled data, **pseudo-labeling** techniques could be used to assign temporary labels, **allowing the computation of a loss**. This enables our adaptive sampling approach to function effectively, regardless of whether the setting is fully supervised or semi-supervised.
>
> ---
>
> **[Q2] Extra regularizations**
>
> Our method does not conflict with sample-agnostic regularization techniques such as weight decay or dropout, as these **operate independently of batch construction**. For regularizations related to the output, such as total variation loss, the adaptive sampling method would indirectly align by prioritizing samples that lead to higher gradient norms, which often correlate with areas requiring regularization (e.g., regions with high variance). The flexibility of our approach ensures compatibility with a wide range of regularization techniques.
>
> ---
> **[Q3] Adversarial Samples/Noises**
>
> Our method can be uOur method does not inherently prioritize noisy samples unless they consistently produce high gradient norms over multiple iterations. In practice, gradient norms for noisy samples often decrease once the model learns to ignore the noise, reducing their selection probability. However, for datasets with substantial noise, adding noise-robust loss functions or noise-detection mechanisms can complement our method to prevent overfitting to such samples.
>
> ---
> **[Q4] Data augmentation**
>
> Data augmentation can indeed affect the adaptive sampling process, as augmented samples often exhibit strong similarity to original samples. Our method could be adapted to treat augmented samples as lower-priority if they exhibit lower gradient relevance, thereby ensuring that the model focuses on diverse samples. Alternatively, adaptive sampling could prioritize augmented samples when they introduce new informative patterns.
>
> ---
> **[Q5] Comparison with Other Sampling Methods (e.g., DoReMi)**
>
> Our method leverages a combinatorial bandit framework focused on adaptive batch selection, which is theoretically grounded in gradient-based relevance. In contrast, reinforcement learning-based methods like DoReMi optimize data mixtures with LLMs as the specific use case. There is a clear difference between the two methods. We are confident that its ability to prioritize informative samples enhances efficiency in many large-scale applications.

---

> > ### Comment · Reviewer_fG67 · 2024-11-25
> >
> > Thanks for the detailed response! Thus, I maintain my score.

---

### Official Review · Reviewer_7DWV · 2024-11-03

**Soundness:** 3
**Presentation:** 3
**Contribution:** 2
**Rating:** 6
**Confidence:** 3

**Summary:**

This paper proposes a variant of the Adam optimizer with a combinatorial bandit approach (AdamCB) for adaptively selecting batches to train on.  Using a combinatorial bandit to select a batch of examples addresses the limitation of a prior approach called Adam with Bandit Sampling (AdamBS) which failed to improve performance with larger batch size due to myopic approach of selecting a single sample at a time with replacement.  Another core contribution of the paper is new theoretical analysis of Adam, AdamBS, and the proposed AdamCB that relaxes prior assumptions as well as fixes errors in the prior proofs.  These convergence guarantees show AdamCB to have faster convergence than Adam and AdamBS and faster convergence with increasing batch size.  Experimental studies show AdamCB to outperform Adam and AdamBS on small scale MLP and CNN tasks.

**Strengths:**

The primary strengths of this paper are as follows:
- The writing is clear and easy to understand.
- The assumptions required for the convergence analysis is more general than previous work and the theory also covers Adam and AdamBS.
- The paper identifies and addresses incorrect assumptions made in the analysis of AdamBS.

**Weaknesses:**

- The experiments are fairly limited in scale and not very reflective of practical settings that we are in these days with large models and large datasets.
- The benefits of adaptive selection will likely be limited in settings with large model dimensionality since then the $d\sqrt{T}$ term will dominate the $\sqrt{d}/n^{3/4}T^{3/4}$ term controlled by adaptive selection.  The potentially marginal improvement of adaptive selection is not discussed as far as I can tell in the main paper (it is discussed in the AdamBS paper).
- It is unclear how useful bandit selection will be in settings where we see the entire dataset just a few times as with LLM pretraining.
- It seems like the theoretical analysis has a lot of overlap with Tran et al. 2019 but this is mainly mentioned in the Appendix and not stated in the main text.  It also detracts from the novelty of the theoretical analysis.

**Questions:**

The convergence rate of Adam, Adam with Bandit Selection, and the proposed Adam with Combinatorial Bandits are all dominated by the $d\sqrt{T}$ term.  In practice, what is the expected speedup of using combinatorial bandit in training settings with >100 million parameters?

In the LLM setting, not only are models over 100 billion parameters in some cases, we also rarely loop through the full dataset if at all during pretraining.  What benefits if any do you expect adaptive batch selection to provide in this setting?

---

> ### Author Response · Authors · 2024-11-22
>
> Thank you for taking the time to review our paper and for your thoughtful and valuable feedback. We appreciate your positive recognition of our work and the constructive comments you have provided. Below, we address each of your comments and questions in detail:
>
> ---
> **[W1] Experiments and practical relevance**
>
> We would like to emphasize that the primary focus of our work is to propose a **provably efficient and practical optimization algorithm** that addresses longstanding inefficiencies—and even incorrectness—in Adam-based methods. AdamCB is a general optimization algorithm with convergence guarantees, making it applicable not only to large models but also to a wide range of models and optimization tasks. We sincerely hope that this fundamental focus is appropriately considered, as our algorithm is not solely designed for large models such as LLMs.
>
> That said, in response to the reviewer's feedback, we have performed and included additional experiments with larger-scale models such as ResNet-18 (11.4 million parameters), ConvNeXt-Base (89 million parameters), and ConvNeXt-Large (198 million parameters), in addition to the previously included experiments with MLPs, CNNs, and the VGG network (Figure 5). These new results are presented in the supplementary material (in Appendix G) of the updated manuscript, with some snapshots shown in the tables below (all results are test errors on CIFAR-10). These experiments clearly demonstrate that AdamCB consistently outperforms baselines such as Adam and AdamBS, even for models with larger parameter counts and higher complexity.
>
>
> **ResNet-18** (11.4 million parameters)
>
> | Epoch | 1 | 2 | 3 | 4 | 5 | 6 | 7 | 8 | 9 | 10 |
> | --- | --- | --- | --- | --- | --- | --- | --- | --- | --- | --- |
> | Adam | 2.913$\pm$ 0.277  | 2.646$\pm$ 0.151 | 2.657$\pm$ 0.126 | 2.718$\pm$ 0.146 | 2.701$\pm$ 0.221 | 2.633$\pm$ 0.175 | 2.649$\pm$ 0.207 | 2.820$\pm$ 0.465 | 2.514$\pm$ 0.159 | 2.449$\pm$ 0.302 |
> | AdamBS | 3.733$\pm$ 1.000 | 4.184$\pm$ 1.316 | 4.262$\pm$ 1.023 | 4.261$\pm$0.855  | 4.528$\pm$1.134  | 4.619$\pm$1.246  | 4.172$\pm$0.825  | 4.643$\pm$1.239  | 4.111$\pm$ 1.026  | 4.763$\pm$1.316  |
> | AdamCB | 4.688$\pm$2.003  | 2.607$\pm$0.225  | 2.587$\pm$0.251  | 2.214$\pm$0.153  | 2.143$\pm$ 0.042 | 2.255$\pm$0.156  | 2.185$\pm$ 0.175 | 2.135$\pm$0.107  | 1.975$\pm$ 0.057 | 1.978$\pm$0.289  |
>
>
> **ConvNext-base** (89 million parameters)
>
> | Epoch | 1 | 2 | 3 | 4 | 5 | 6 | 7 | 8 | 9 | 10 |
> | --- | --- | --- | --- | --- | --- | --- | --- | --- | --- | --- |
> | Adam | 0.761 $\pm$ 0.040 | 0.691$\pm$ 0.043 | 0.673$\pm$ 0.017 | 0.612$\pm$0.022 | 0.618$\pm$0.037 | 0.573$\pm$0.044 | 0.570$\pm$ 0.057 | 0.608$\pm$0.016 | 0.601$\pm$ 0.012 | 0.544$\pm$0.027 |
> | AdamBS | 0.665$\pm$0.038 | 0.520$\pm$ 0.010 | 0.493$\pm$ 0.029 | 0.798$\pm$ 0.406 | 1.108$\pm$ 0.829 | 2.082$\pm$ 0.752 | 2.163$\pm$0.802 | 2.050$\pm$0.722 | 2.229$\pm$0.856 | 2.160$\pm$ 0.834 |
> | AdamCB | 0.624$\pm$0.005 | 0.458$\pm$0.005 | 0.434$\pm$0.028 | 0.400$\pm$0.025 | 0.374$\pm$0.013 | 0.345$\pm$0.019 | 0.323$\pm$0.010 | 0.328$\pm$0.017 | 0.325$\pm$0.019 | 0.312$\pm$0.026 |
>
>
> **ConvNext-large** (198 million parameters)
>
> | Epoch | 1 | 2 | 3 | 4 | 5 | 6 | 7 | 8 | 9 | 10 |
> | --- | --- | --- | --- | --- | --- | --- | --- | --- | --- | --- |
> | Adam | 0.722$\pm$0.038 | 0.582$\pm$ 0.041 | 0.532$\pm$0.049 | 0.544$\pm$0.018 | 0.525$\pm$0.011 | 0.519$\pm$0.045 | 0.553$\pm$0.037 | 0.490$\pm$0.021 | 0.471$\pm$0.039 | 0.451$\pm$0.020 |
> | AdamBS | 0.589$\pm$0.030 | 0.423$\pm$0.010 | 0.386$\pm$0.020 | 0.386$\pm$0.029 | 0.597$\pm$0.374 | 0.729$\pm$0.561 | 0.734$\pm$0.573 | 2.218$\pm$1.873 | 2.317$\pm$2.017 | 3.227$\pm$1.475 |
> | AdamCB | 0.538$\pm$0.023 | 0.403$\pm$0.013 | 0.386$\pm$0.010 | 0.364$\pm$0.009 | 0.338$\pm$0.018 | 0.304$\pm$0.013 | 0.274$\pm$0.009 | 0.281$\pm$0.015 | 0.264$\pm$0.006 | 0.281$\pm$0.031 |
>
> The inclusion of new experiments on these larger models complements the MLP, CNN, VGG network results already presented in the main paper, bridging the gap between simpler architectures and larger models. This expanded evaluation further substantiates the effectiveness and scalability of AdamCB across a diverse range of architectures. For more details, please refer to Appendix G in the updated manuscript.
>
> With these new results, alongside the findings already presented both in theory and experiments, we strongly believe that the efficacy of our method is well supported. Furthermore, our core contribution—the provable efficiency of AdamCB—is further strengthened by this comprehensive evaluation. We are also more than willing to conduct additional evaluations, time permitting, to further strengthen our results. We sincerely and respectfully request the reviewer to recognize the potential impact of our work in light of our main contributions and the additional evidence provided.

---

> ### Author Response · Authors · 2024-11-22
>
> **[W2] In high dimension**
>
> We appreciate the reviewer’s comment and are happy to clarify any potential misunderstandings. The convergence guarantee shown in our paper is a regret comparison between the true optimal solution and the algorithm's performance, where the first term, $\mathcal{O}(d \sqrt{T})$, represents the leading term. But, that is about the difference between the performance of an optimization algorithm and the optimality.
>
> However, if the reviewer’s point pertains to the "marginal improvement of adaptive selection" (as it appears from the comment), this requires examining the difference between adaptive and non-adaptive (uniform) sampling strategies.
>
> As illustrated in Table 1, AdamCB achieves a convergence rate of
> $\mathcal{O}\left(d\sqrt{T} + \frac{\sqrt{d}}{n^{3/4}}\left(\frac{T}{K}\ln{\frac{n}{K}}\right)^{1/4}\right)$,
> while corrected Adam (with uniform sampling) achieves a convergence rate of
> $\mathcal{O}\left(d\sqrt{T} + \frac{\sqrt{d}}{n^{1/2}}\sqrt{T}\right)$.
> As shown in the regret analysis, the constant factor in the leading term $\mathcal{O}(d \sqrt{T})$ is the same for both AdamCB and corrected Adam. Therefore, the regret gap arises from the difference in the second terms:
> $\mathcal{O}\left(\frac{\sqrt{d}}{n^{1/2}}\sqrt{T}\right) - \mathcal{O}\left(\frac{\sqrt{d}}{n^{3/4}}\left(\frac{T}{K}\ln{\frac{n}{K}}\right)^{1/4}\right)$.
> This gap becomes larger as $d$ or $T$ increases, highlighting the greater comparative improvement provided by AdamCB in high-dimensional settings. This behavior is both theoretically demonstrated and empirically validated in our experiments with neural network models, as presented in the experiments of Appendix G.
>
> We sincerely thank the reviewer for raising this question, as it provides an opportunity to clarify this point in detail. However, we respectfully disagree with the notion (if the implication was intended) that there is a diminishing marginal improvement of adaptive selection in high dimensions. The theoretical results in our paper clearly demonstrate that adaptive selection retains its advantages in both low- and high-dimensional settings.
> Additionally, we have reviewed the discussion in the AdamBS paper mentioned by the reviewer. We believe that the discussion there does not necessarily align with the theoretical results in our work as well as the comparison mentioned above, and as shown in our analysis, the results in AdamBS themselves are invalid. We would be more than happy to include this discussion in the revised manuscript to further highlight our contributions.
>
> In conclusion, adaptive selection consistently provides a distinct advantage in both low- and high-dimensional settings, as demonstrated by our theoretical analysis and empirical results.
>
> ---
> **[W3, Q2] LLM pretraining**
>
> We appreciate the reviewer’s point regarding the applicability of our method to settings such as LLM pretraining, where the dataset may only be swept a few times. However, our method remains valuable in such scenarios. Each gradient update during LLM pretraining is computationally expensive, and any unnecessary gradient computation becomes significantly more wasteful compared to smaller models.
> Even in cases where the dataset is seen only a limited number of times, the adaptive batch sampling introduced by our method can still provide meaningful efficiency improvements by prioritizing the most informative gradients. Exploring the full potential of this adaptive strategy in LLM pretraining contexts is indeed an intriguing direction for future work. That said, we respectfully believe this aspect should not be considered a weakness of our work as a general optimization method.
>
> ---
> **[W4] On Tran et al. 2019**
>
> We would like to clarify that while some techniques from Tran et al. 2019 are utilized to address the technical error in the original Adam analysis, we have explicitly acknowledged and properly credited Tran et al. 2019 in our work.
>
> It is important to emphasize that the primary goal of our research is not to fix errors in the analysis of Adam (or AMSGrad). Rather, the main theoretical contribution of our work lies in **designing and proving the provable efficiency of adaptive batch sampling** (leveraging a combinatorial bandit approach) for Adam-based optimization. This contribution is independent of Tran et al. 2019.
>
> Our theoretical improvement in convergence efficiency, achieved through the rigorous integration of a combinatorial bandit framework into Adam optimization, is a core novelty of our work. This innovation is not present in Tran et al. 2019 and represents a significant advancement in the field.
>
> Therefore, the novelty of our work is not detracted in any sense, as it addresses a distinct problem and provides contributions that go beyond the scope of Tran et al. 2019. We are happy to clarify this.

---

> > ### Author Response · Authors · 2024-11-22
> >
> > ### **Answers to Questions**
> > ---
> > **[Q1]** As detailed in our response to the [W2] comment, we again emphasize that the comparative advantage of AdamCB arises from the second-order terms, Importantly, as demonstrated in our theoretical analysis and experiments, this comparative advantage **does not diminish**, even in high-dimensional settings with a large number of parameters (e.g., > 100 million).
> >
> > In practice, the speedup achieved by AdamCB extends beyond theoretical guarantees. In our newly included experiments with large-scale models—89 million parameters (ConvNext-base) and 198 million parameters (ConvNext-large)—AdamCB consistently outperforms both Adam and Adam with Bandit Selection (AdamBS) in terms of convergence efficiency. This improvement is attributed to AdamCB’s adaptive batch selection mechanism, which prioritizes informative gradients, accelerating convergence compared to the uniform sampling used in Adam or the single-arm selection strategy in AdamBS.
> > These experimental results underscore that AdamCB effectively leverages adaptive selection to enhance training efficiency. This aligns with our theoretical results.
> >
> > In conclusion, the advantages of AdamCB, both theoretical and empirical, remain significant irrespective of model size or dimensionality. The additional experiments as well as the already presented results provide compelling evidence of its effectiveness across both small- and large-scale settings.
> >
> > **[Q2]**  Answered along with the response to [W3]

---

> > > ### Comment · Reviewer_7DWV · 2024-11-26
> > > **Post Author Response**
> > >
> > > I appreciate the responses to my questions but I will maintain my score of 6.
> > >
> > > For large networks, it's clear the $d\sqrt{T}$ term dominates.  Take for example the ConvNext-base model with 89 million parameters and cifar 10 dataset with 50k training examples.  For Adam, the ratio of the second term to the first term is 4.7e-7, and even reducing this to 0 may have a negligible impact.

---

> ### Author Response · Authors · 2024-11-26
>
> We thank the reviewer for their continued discussion and for assigning a positive score.
>
> However, there appears to be a misunderstanding in the reviewer's comment regarding the performance gap between AdamCB and Adam, as our key results indicate the opposite of what the reviewer's comment suggests. We sincerely hope the reviewer recognizes our genuine intention to ensure that the evaluations are based on what our results truly represent. To this end, we are happy to provide further clarification on this point.
>
> As explained in our previous response (see the section "In high dimension"), **the performance gap between AdamCB and Adam does NOT diminish as $d$ increases**. On the contrary, **the performance gap increases with $d$** with $O(\sqrt{d})$ -- the gap is given by:
>
> $$
> \mathcal{O}\left(\frac{\sqrt{d}}{n^{1/2}}\sqrt{T}\right) - \mathcal{O}\left(\frac{\sqrt{d}}{n^{3/4}}\left(\frac{T}{K}\ln{\frac{n}{K}}\right)^{1/4}\right).
> $$
>
> This reflects the fact that AdamCB retains its comparative advantage even in high-dimensional settings (or for any $d$, both in low- and high-dimension). We had clearly stated this above and the reviewer may refer to our response for details (e.g., the distinction between regret compared to the optimality and comparative advantage of AdamCB over Adam).
>
> To further illustrate, consider the example you provided. The relative difference in regret between Adam and AdamCB (**the larger this value, the greater the advantage of AdamCB**) is shown below:
>
> - **89 million parameters, training examples $n = 50,000$, iterations $T = 1000$, batch size $K = 128$**
> $$
> C \cdot \left( \sqrt{\frac{89,000,000}{50,000}}\sqrt{1000} - \frac{\sqrt{89,000,000}}{50,000^{3/4}}\left(\frac{1000}{128} \ln{\frac{50,000}{128}}\right)^{1/4} \right) \approx \mathbf{1328} \cdot C
> $$
>
> - **89 million parameters, training examples $n = 50,000$, iterations $T = 10,000$, batch size $K = 128$**
> $$
> C \cdot \left( \sqrt{\frac{89,000,000}{50,000}}\sqrt{10,000} - \frac{\sqrt{89,000,000}}{50,000^{3/4}}\left(\frac{10,000}{128} \ln{\frac{50,000}{128}}\right)^{1/4} \right) \approx \mathbf{4208} \cdot C
> $$
>
> - **198 million parameters, training examples $n = 50,000$, iterations $T = 10,000$, batch size $K = 128$**
> $$
> C \cdot \left( \sqrt{\frac{198,000,000}{50,000}}\sqrt{10,000} - \frac{\sqrt{198,000,000}}{50,000^{3/4}}\left(\frac{10,000}{128} \ln{\frac{50,000}{128}}\right)^{1/4} \right) \approx \mathbf{6276} \cdot C
> $$
> where $C > 1$ is some constant. (By the way, this gap is in cumulative sense as the regret was defined in cumulation.) These calculations demonstrate that **AdamCB maintains a significant comparative advantage over Adam** in all cases.
>
> Furthermore, **this advantage is observed not only in theory (as shown in the regret bound) but also in experiments**. Please see **Figure 8** and **Figure 9** in Appendix G for experimental results using ConvNext models, where AdamCB consistently maintains its comparative advantage over Adam.
>
> We sincerely hope this clears up any misunderstanding, and we would be happy to provide further clarifications if needed. In light of this clarification (and given our earnest effort to address your feedback comprehensively), we kindly and respectfully ask the reviewer to reconsider the rating.

---

### Official Review · Reviewer_797D · 2024-11-03

**Soundness:** 3
**Presentation:** 3
**Contribution:** 3
**Rating:** 8
**Confidence:** 3

**Summary:**

This paper introduces the AdamCB algorithm, aiming to address inefficiencies in the Adam optimizer by improving data sampling. AdamCB integrates combinatorial bandit techniques, enabling adaptive batch selection to focus on informative samples. The authors claim enhanced theoretical guarantees, providing a rigorously analyzed regret bound that surpasses those of standard Adam and its bandit-based variant, AdamBS, which are identified here as having flawed guarantees. Experimental results across diverse datasets demonstrate the practical benefits of AdamCB, indicating faster convergence and greater efficiency.

**Strengths:**

1. This paper rigorously addresses and corrects the theoretical flaws in convergence guarantees for both Adam and AdamBS, presenting refined proofs that offer independent value to the community.
2. The proposed AdamCB algorithm is shown to be both theoretically robust and empirically effective, with rigorous analysis and extensive experimental validation.
3. The paper presents the method and supporting claims clearly, facilitating reader comprehension and enhancing accessibility of the technical content.
4. Through a fair and thorough comparison, this paper evaluates AdamCB alongside Adam and AdamBS, incorporating corrected theoretical guarantees to provide a balanced assessment.

**Weaknesses:**

1. The paper's motivation could benefit from further clarification and depth. In the abstract and introduction, the authors state that uniform sampling leads to inefficiencies in Adam, but they should specify the type of inefficiency (e.g., memory, computational, time, or convergence efficiency; presumably the latter). Additionally, the authors should provide evidence or discussion showing that alternative sampling methods indeed improve Adam's efficiency, strengthening the case for the proposed approach.
2. Algorithm 3 requires prior knowledge of $L$ for the weight update rule, which may be limiting in practical applications. It would be valuable to discuss potential ways to relax this requirement or clarify how the authors manage this constraint in practice.

**Questions:**

1. In the weight adjustment section, it is unclear why AdamCB requires the sum of probabilities to equal $K$ instead of 1. This choice appears to necessitate an additional operation in the sampling strategy, specifically the introduction of a threshold $\tau$. Clarification on the rationale behind this requirement would be beneficial.
2. The paper could further explain why increasing the sample size $n$ leads to faster convergence in AdamCB. A theoretical or intuitive justification for this relationship would strengthen the understanding of the algorithm’s convergence behavior.

---

> ### Author Response · Authors · 2024-11-22
>
> Thank you for taking the time to review our paper and for your thoughtful and valuable feedback. We deeply appreciate your recognition of our work and the constructive comments you have provided. Below, we address each of your comments and questions in detail:
>
> ---
>
> **[W1] Clarification Regarding Inefficiency**
>
> We thank the reviewer for the opportunity to clarify this point. The inefficiencies of Adam (with uniform sampling) can be categorized into two main issues:
>
> 1. **Convergence Inefficiency**: As explained in Section 2.4.2, the analysis of Adam reveals a technical error in the original framework, preventing it from providing convergence guarantees. This issue implies that Adam can potentially diverge under certain conditions.
> 2. **Algorithmic Limitation with Uniform Sampling**: Adam's uniform sampling nature limits its ability to fully leverage the feedback provided by multiple samples in each mini-batch. This constraint leads to slower convergence (even with corrections) compared to our proposed algorithm, AdamCB, which utilizes a combinatorial bandit sampling mechanism to address these inefficiencies (see Table 1).
>
> Our proposed AdamCB algorithm overcomes these challenges by dynamically adapting the sampling distribution to prioritize informative samples, resulting in faster convergence. This improvement is supported by rigorous theoretical guarantees, as outlined in Theorem 1. The provable efficiency of our method is verified through comparisons with existing methods, as shown in Table 1.
>
> AdamCB achieves provable regret bounds and demonstrates superior convergence performance compared to both the original Adam and its bandit-based variant, AdamBS. Furthermore, our experiments highlight that AdamCB consistently outperforms its counterparts across various models and datasets. We will refine the paper to ensure these clarifications are more explicit and accessible. Thank you for the opportunity to highlight these points.
>
> ---
> **[W2] Knowledge of $L$ is NOT needed.**
>
> We sincerely thank the reviewer for the comment regarding the knowledge of $L$ in Algorithm 3. We are happy to clarify that the presentation of the algorithm requiring knowledge of $L$ is actually **without loss of generality**, meaning that prior knowledge of $L$ is **not necessary**.
>
> In fact, we can relax this requirement and adopt a dynamic approach where the upper bound \(L\) is replaced with a running maximum based on the **gradient norms** observed during training. Specifically:
>
> - **Dynamic update of $L$:** In practice, we can replace $L$ with $L_{t}=\max_{t' \leq t} \max_{i \in [n]} \|g_{i,t'}\|$, where $L_{t}$ maintain the running maximum gradient norm observed during training up to iteration $t$. This ensures $L_{t}$ is non-decreasing and provides a sufficient upper bound for the gradient norms.
> - **Implementation in Algorithm 3:** The weight update rule is modified to incorporate $L_t$, which is updated at every iteration: $L_t \leftarrow \max(L_{t-1}, \max_{i \in [n]} \| g_{i,t} \|).$This ensures that $L_t$ remains a valid upper bound for all gradients observed during training.
> - **Impact on Theoretical Guarantees:** The use of $L_t$ still preserves the theoretical analysis of the algorithm, as the bounded gradient assumption is satisfied by construction. The convergence guarantees and performance remain unaffected.
>
> This modification enhances the practicality of our method while retaining the rigor of our theoretical results. We sincerely thank the reviewer again for their valuable feedback and for providing us the opportunity to improve the practicality of our approach.

---

> ### Author Response · Authors · 2024-11-22
>
> ### **Answers to Questions**
> ---
>
> **[Q1] Clarification on Setting $\sum_{i=1}^{n} p_{i,t}=K$.**
>
> - **Why $\sum_{i=1}^{n} p_{i,t}=K$ instead of $\sum_{i=1}^{n} p_{i,t}=1$?**
>
> We thank the reviewer for providing us with the opportunity to clarify this point. AdamCB uses a **combinatorial bandit framework** to sample multiple arms (samples) without replacement in each mini-batch. Unlike single-arm selection bandit algorithms like AdamBS, where$\sum_{i=1}^{n}p_{i,t}=1$ because only one arm is selected at a time, AdamCB must select $K$  simultaneously for a mini-batch. Therefore, it is natural to scale the sum of the probabilities to $K$, reflecting the expected number of samples selected in each round.
>
> If the sum of probabilities were constrained to 1, the algorithm would need to perform additional rescaling or sampling adjustments to ensure $K$ samples are drawn, which would unnecessarily complicate the sampling process. Instead, directly setting $\sum_{i=1}^{n}p_{i,t}=K$ aligns the probability distribution with the batch-level selection requirements. By setting $\sum_{i=1}^{n} p_{i,t}=K$, AdamCB **simplifies the sampling process** and ensures compatibility with mini-batch training.
>
> - **Rationale for the Threshold $\tau$**
>
> Allowing the sum of probabilities to equal $K$ can lead to individual probabilities $p_{i,t}$ exceeding 1, especially when certain samples are assigned significantly higher weights due to their importance or gradient magnitude. To ensure valid probabilities and prevent any sample from being overrepresented, AdamCB introduces a threshold $\tau$. If a sample's probability $p_{i,t}$ exceeds $\tau$:
>
> 1. Its index is added to a null set $S_{null,t}$, effectively removing it from active consideration for selection.
> 2. The probabilities of the remaining samples are adjusted to redistribute the excess weight while ensuring the sum of probabilities remains $K$.
>
> This adjustment ensures that no single sample dominates the mini-batch while maintaining the proportional relationship between the weights $w_{i,t}$ and the probabilities $p_{i,t}$. We again thank the reviewer for the chance to clarify this key aspect.
>
> ---
> **[Q2] Relationship Between Sample Size $n$  and Convergence in AdamCB**
>
> We thank the reviewer for their thoughtful feedback, which provides us with an opportunity to enhance the understanding of our algorithm’s convergence behavior by further explaining the relationship between the sample size $n$ and the faster convergence of AdamCB.
>
> **For theory**, the second term of the regret bound of AdamCB (Theorem 1) is given as, $(\sqrt{d} / n^{3/4}) \left( (T/K) \ln{(n/K)}\right)^{1/4}$. From this term, it is evident that the regret decreases as $n$ increases. This implies that larger $n$ leads to smaller regret, improving the algorithm’s convergence rate.
>
> **For intuitions,** increasing the sample size $n$ expands the pool of data samples from which mini-batches are drawn. This broader pool allows AdamCB (as well as other algorithms albeit slower convergence rate) to access a wider range of informative samples during each update step. As a result, the selected mini-batches are more representative of the overall data distribution, **reducing variance in the gradient estimates** and leading to more accurate updates. This relationship underscores the advantage of AdamCB in leveraging large datasets to achieve faster convergence, making it particularly effective in modern machine learning tasks with abundant data.

---

> > ### Comment · Reviewer_797D · 2024-11-27
> >
> > I would like to thank the authors for their comprehensive explanations. My concerns have been well addressed, and I sincerely hope that these discussions will be included in the revised paper. So, I decided to raise my score.

---

> > > ### Author Response · Authors · 2024-11-30
> > >
> > > Thank you for your support and for recognizing the contributions of our work. Your feedback has been invaluable in helping us improve our paper.

---

### Meta-Review · Area_Chair_Vheo · 2024-12-23

**Metareview:**

The paper introduces AdamCB, an extension of the Adam optimizer using combinatorial bandit-based adaptive batch selection. It rigorously analyzes convergence guarantees for Adam, AdamBS, and AdamCB, correcting errors in prior proofs. The reviewers highlighted the paper's rigorous theoretical contributions, addressing fundamental issues in Adam's convergence and clear writing. Key criticisms included limited experiments on large-scale models and real-world scenarios, and that there is a lot of overlap in some of the formal analysis with Tran et al, which is only mentioned in the appendix. To me, the biggest criticism is that the paper’s experimental evaluation is minimal: no discussion of (decoupled) weight decay (or extension to a „AdamWCB“), only convolutional networks, no language models, no diffusion, and only using a single fixed hyperparameter setting for all methods (where different methods sometimes require different default hyperparameters; especially AdamBS often diverges, and I assume it would require a lower learning rate). As such, this is a strong theory paper, but its practical relevance will have to be determined in the future.

**Additional Comments On Reviewer Discussion:**

Reviewer 797D's concerns around prior knowledge and uniform sampling were resolved, leading to a score increase.
Some of the other two reviewers' concerns were addressed, but they both remained skeptical about real-world applicability / large-scale applications.

---

### Decision · Program_Chairs · 2025-01-22

Accept (Poster)